# A Generalized Weighted Optimization Method for Computational Learning and Inversion

**Björn Engquist**
The University of Texas at Austin
Austin, TX 78712, USA
engquist@oden.utexas.edu

**Kui Ren**
Columbia University
New York, NY 10027, USA
kr2002@columbia.edu

**Yunan Yang**
ETH Zürich
Zürich, Switzerland
yyn0410@gmail.com

## Abstract

The generalization capacity of various machine learning models exhibits different phenomena in the under- and over-parameterized regimes. In this paper, we focus on regression models such as feature regression and kernel regression and analyze a generalized weighted least-squares optimization method for computational learning and inversion with noisy data. The highlight of the proposed framework is that we allow weighting in both the parameter space and the data space. The weighting scheme encodes both a priori knowledge on the object to be learned and a strategy to weight the contribution of different data points in the loss function. Here, we characterize the impact of the weighting scheme on the generalization error of the learning method, where we derive explicit generalization errors for the random Fourier feature model in both the under- and over-parameterized regimes. For more general feature maps, error bounds are provided based on the singular values of the feature matrix. We demonstrate that appropriate weighting from prior knowledge can improve the generalization capability of the learned model.

## 1 Introduction

Given $N$ data pairs $\{x_j, y_j\}_{j=1}^N$, where $x_j \in \mathbb{R}$, $y_j \in \mathbb{C}$, $j = 1, \ldots, N$, we are interested in learning a random Fourier feature (RFF) model (Rahimi & Recht, 2008; Liao et al., 2020; Xie et al., 2020)

$$f_{\boldsymbol{\theta}}(x) = \sum_{k=0}^{P-1} \theta_k e^{\mathrm{i}kx}, \quad x \in [0, 2\pi], \tag{1}$$

where $P \in \mathbb{N}$ is a given positive integer and we used the short-hand notation $\boldsymbol{\theta} := (\theta_0, \cdots, \theta_{P-1})^\mathfrak{T}$ with the superscript $^\mathfrak{T}$ denoting the transpose operation.

This exact model as well as its generalization to more complicated setups have been extensively studied; see for instance Liao & Couillet (2018); Shahrampour & Kolouri (2019); d'Ascoli et al. (2020); Li et al. (2020); Özcelikkale (2020); Liu et al. (2020; 2021) and references therein. While this model may seem to be overly simplified from a practical perspective for many real-world applications, it serves as a prototype for theoretical understandings of different phenomena in machine learning models (Sriperumbudur & Szabo, 2015; Belkin et al., 2020; Li et al., 2021a).

A common way to computationally solve this learning problem is to reformulate it as an optimization problem where we find $\boldsymbol{\theta}$ by minimizing the model and data mismatch for a given dataset. In this paper, we assume that the training data are collected on a uniform grid of $x$ over the domain $[0, 2\pi]$. That is, $\{x_j = \frac{2\pi j}{N}\}_{j=0}^{N-1}$. Let $\omega_N = \exp(\frac{2\pi \mathrm{i}}{N})$ where $\mathrm{i}$ is the imaginary unit. We introduce $\Psi \in \mathbb{C}^{N \times P}$ to be the feature matrix with elements

$$(\Psi)_{jk} = (\omega_N)^{jk}, \quad 0 \le j \le N-1, \ 0 \le k \le P-1.$$

Based on the form of $f_{\boldsymbol{\theta}}(x)$ in (1), we can then write the 2-norm based data mismatch into the form $\sum_{j=0}^{N-1} |f_{\boldsymbol{\theta}}(x_j) - y_j|^2 = \|\Psi\boldsymbol{\theta} - \mathbf{y}\|_2^2$ where the column data vector $\mathbf{y} = (y_0, \cdots, y_{N-1})^\mathfrak{T}$. The learning problem is therefore recast as a least-squares optimization problem of the form

$$\widehat{\boldsymbol{\theta}} = \arg\min_{\boldsymbol{\theta}} \|\Psi\boldsymbol{\theta} - \mathbf{y}\|_2^2, \tag{2}$$

assuming that a minimizer does exist, especially when we restrict $\boldsymbol{\theta}$ to an appropriate space.

In a general feature regression problem, the Fourier feature $\{e^{ikx}\}_{k=0}^{P-1}$ is then replaced with a different feature model $\{\varphi_k(x)\}_{k=0}^{P-1}$, while the least-squares form (2) remains unchanged except that the entries of the matrix $\Psi$ is now $\Psi_{jk} = \varphi_k(x_j)$. We emphasize that this type of generalization will be discussed in Section 5. Moreover, we remark that this least-squares optimization formulation is a classical computational inversion tool in solving the general linear inverse problems of the form $\Psi\boldsymbol{\theta} = y$; see for instance Engl et al. (1996); Tarantola (2005) and references therein.

**Previous work on weighted optimization for feature and kernel learning.** Xie et al. (2020) studied the fitting problem for this model under the assumption that the coefficient vector $\boldsymbol{\theta}$ is sampled from a distribution with the property that $\gamma$ is a positive constant,

$$\mathbb{E}_{\boldsymbol{\theta}}[\boldsymbol{\theta}] = \mathbf{0}, \ \ \mathbb{E}_{\boldsymbol{\theta}}[\boldsymbol{\theta}\boldsymbol{\theta}^*] = c_\gamma \boldsymbol{\Lambda}_{[P]}^{-2\gamma}, \tag{3}$$

where the superscript $*$ denotes the Hermitian transpose and the diagonal matrix $\boldsymbol{\Lambda}_{[P]}$ has diagonal elements $(\boldsymbol{\Lambda}_{[P]})_{kk} = t_k = 1 + k$, $k \geq 0$. That is,

$$\boldsymbol{\Lambda}_{[P]} = \mathrm{diag}\{t_0, \ t_1, \ t_2, \ \ldots, \ t_k, \ \ldots, \ t_{P-1}\}, \ \ \ t_k := 1 + k. \tag{4}$$

The subscript $[P]$ indicates that $\boldsymbol{\Lambda}_{[P]}$ is a diagonal submatrix of $\boldsymbol{\Lambda}$ that contains its element indexed in the set $[P] := \{0, \ 1, \ \cdots, \ P - 1\}$. The normalization constant $c_\gamma = 1/(\sum_{k=0}^{P-1}(1 + k)^{-2\gamma})$ is only selected so that $\mathbb{E}_{\boldsymbol{\theta}}[\|\boldsymbol{\theta}\|^2] = 1$. It does not play a significant role in the rest of the paper.

The main assumption in (3) says that statistically, the signal to be recovered has algebraically decaying Fourier coefficients. This is simply saying that the target function we are learning is relatively smooth, which is certainly the case for many functions as physical models in practical applications.

It was shown in Xie et al. (2020) that, to learn a model with $p \leq P$ features, it is advantageous to use the following weighted least-squares formulation

$$\widehat{\boldsymbol{\theta}}_p = \boldsymbol{\Lambda}_{[p]}^{-\beta}\widehat{\mathbf{w}}, \ \ \text{with} \ \ \widehat{\mathbf{w}} = \arg\min_{\boldsymbol{\theta}} \|\Psi_{[N \times p]}\boldsymbol{\Lambda}_{[p]}^{-\beta}\mathbf{w} - \mathbf{y}\|_2^2, \tag{5}$$

when the learning problem is overparameterized, i.e., $p > N$. Here, $\Psi_{[N \times p]} \in \mathbb{C}^{N \times p}$ is the matrix containing the first $p$ columns of $\Psi$, and $\beta > 0$ is some pre-selected exponent that can be different from the $\gamma$ in (3). To be more precise, we define the the generalization error of the learning problem

$$\mathcal{E}_\beta(P, p, N) := \mathbb{E}_{\boldsymbol{\theta}}\left[\|f_{\boldsymbol{\theta}}(x) - f_{\widehat{\boldsymbol{\theta}}_p}(x)\|_{L^2([0,2\pi])}^2\right] = \mathbb{E}_{\boldsymbol{\theta}}\left[\|\widehat{\boldsymbol{\theta}}_p - \boldsymbol{\theta}\|_2^2\right], \tag{6}$$

where the equality comes from the Parseval's identity, and $\widehat{\boldsymbol{\theta}}_p$ is understood as the vector $(\boldsymbol{\theta}_p^{\mathfrak{T}}, 0, \cdots, 0)^{\mathfrak{T}}$ so that $\boldsymbol{\theta}$ and $\widehat{\boldsymbol{\theta}}_p$ are of the same length $P$. The subscript $\boldsymbol{\theta}$ in $\mathbb{E}_{\boldsymbol{\theta}}$ indicates that the expectation is taken with respect to the distribution of the random variable $\boldsymbol{\theta}$. It was shown in Xie et al. (2020) that the lowest generalization error achieved from the weighted least-squares approach (5) in the overparameterized regime ($p > N$) is strictly less than the lowest possible generalization error in the underparameterized regime ($p \leq N$). This, together with the analysis and numerical evidence in previous studies such as those in Belkin et al. (2019; 2020), leads to the understanding that smoother approximations (i.e., solutions that are dominated by lower Fourier modes) give better generalization in learning with the RFF model (1).

**Main contributions of this work.** In this work, we analyze a generalized version of (5) for general feature regression from noisy data. Following the same notations as before, we introduce the following weighted least-squares formulation for feature regression:

$$\widehat{\boldsymbol{\theta}}_p^\delta = \boldsymbol{\Lambda}_{[p]}^{-\beta}\widehat{\mathbf{w}}, \ \ \text{with} \ \ \widehat{\mathbf{w}} = \arg\min_{\mathbf{w}} \|\boldsymbol{\Lambda}_{[N]}^{-\alpha}\left(\Psi_{[N \times p]}\boldsymbol{\Lambda}_{[p]}^{-\beta}\mathbf{w} - \mathbf{y}^\delta\right)\|_2^2, \tag{7}$$

where the superscript $\delta$ on $\mathbf{y}$ and $\widehat{\boldsymbol{\theta}}_p$ denotes the fact that the training data contain random noise of level $\delta$ (which will be specified later). The exponent $\alpha$ is pre-selected and can be different from $\beta$. While sharing similar roles with the weight matrix $\boldsymbol{\Lambda}_{[P]}^{-\beta}$, the weight matrix $\boldsymbol{\Lambda}_{[N]}^{-\alpha}$ provides us the additional ability to deal with noise in the training data. Moreover, as we will see later, the weight matrix $\boldsymbol{\Lambda}_{[N]}^{-\alpha}$ does not have to be either diagonal or in the same form as the matrix $\boldsymbol{\Lambda}_{[p]}^{-\beta}$; the

current form is to simplify the calculations for the RFF model. It can be chosen based on the *a priori* information we have on the operator $\Psi$ as well as the noise distribution of the training data.

The highlight and also one of the main contributions of our work is that we introduce a new weight matrix $\mathbf{\Lambda}_{[N]}^{-\alpha}$ that emphasizes the data mismatch in terms of its various modes, in addition to $\mathbf{\Lambda}_{[p]}^{-\beta}$, the weight matrix imposed on the unknown feature coefficient vector $\boldsymbol{\theta}$. This type of generalization has appeared in different forms in many computational approaches for solving inverse and learning problems where the standard 2-norm (or $\ell^2$ in the infinite-dimensional setting) is replaced with a weighted norm that is either weaker or stronger than the unweighted 2-norm.

In this paper, we characterize the impact of the new weighted optimization framework (7) on the generalization capability of various feature regression and kernel regression models. The new contributions of this work are threefold. First, we discuss in detail the generalized weighted least-squares framework (7) in Section 2 and summarize the main results for training with *noise-free* data in Section 3 for the RFF model in both the overparameterized and the underparameterized regimes. This is the setup considered in Xie et al. (2020), but our analysis is based on the proposed weighted model (7) instead of (5) as in their work. Second, we provide the generalization error in both two regimes for the case of training with *noisy* data; see Section 4. This setup was not considered in Xie et al. (2020), but we demonstrate here that it is a significant advantage of the weighted optimization when data contains noise since the weighting could effectively minimize the influence of the noise and thus improve the stability of feature regression. Third, we extend the same type of results to more general models in feature regression and kernel regression that are beyond the RFF model, given that the operator $\Psi$ satisfies certain properties. In the general setup presented in Section 5, we derive error bounds in the asymptotic limit when $P$, $N$, and $p$ all become very large. Our analysis provides some guidelines on selecting weighting schemes through either the parameter domain weighting or the data domain weighting, or both, to emphasize the features of the unknowns to be learned based on a priori knowledge.

## 2 GENERALIZED WEIGHTED LEAST-SQUARES FORMULATION

There are four essential elements in the least-squares formulation of the learning problem: (i) the parameter to be learned ($\boldsymbol{\theta}$), (ii) the dataset used in the training process ($\mathbf{y}$), (iii) the feature matrix ($\Psi$), and (iv) the metric chosen to measure the data mismatch between $\Psi\boldsymbol{\theta}$ and $\mathbf{y}$.

Element (i) of the problem is determined not only by the data but also by *a priori* information we have. The information encoded in (3) reveals that the size (i.e., the variance) of the Fourier modes in the RFF model decays as fast as $(1 + k)^{-2\gamma}$. Therefore, the low-frequency modes in (1) dominate high-frequency modes, which implies that in the learning process, we should search for the solution vectors that have more low-frequency components than the high-frequency components. The motivation behind introducing the weight matrix $\mathbf{\Lambda}_{[p]}^{-\beta}$ in (5) is exactly to force the optimization algorithm to focus on admissible solutions that are consistent with the *a priori* knowledge given in (3), which is to seek $\boldsymbol{\theta}$ whose components $|\theta_k|^2$ statistically decay like $(1 + k)^{-2\beta}$.

When the problem is formally determined (i.e., $p = N$), the operator $\Psi$ is invertible, and the training data are noise-free, similar to the weight matrix $\mathbf{\Lambda}_{[p]}^{-\beta}$, the weight matrix $\mathbf{\Lambda}_{[N]}^{-\alpha}$ does not change the solution of the learning problem. However, as we will see later, these two weight matrices do impact the solutions in various ways under the practical setups that we are interested in, for instance, when the problem is over-parameterized or when the training data contain random noise.

The weight matrix $\mathbf{\Lambda}_{[N]}^{-\alpha}$ is introduced to handle elements (ii)-(iv) of the learning problem. First, since $\mathbf{\Lambda}_{[N]}^{-\alpha}$ is directly applied to the data $\mathbf{y}^\delta$, it allows us to suppress (when $\alpha > 0$) or promote (when $\alpha < 0$) high-frequency components in the data during the training process. In particular, when transformed back to the physical space, the weight matrix $\mathbf{\Lambda}_{[N]}^{-\alpha}$ with $\alpha > 0$ corresponds to a smoothing convolutional operator whose kernel has Fourier coefficients decaying at the rate $k^{-\alpha}$. This operator suppresses high-frequency information in the data. Second, $\mathbf{\Lambda}_{[N]}^{-\alpha}$ is also directly applied to $\Psi\boldsymbol{\theta}$. This allows us to precondition the learning problem by making $\mathbf{\Lambda}_{[N]}^{-\alpha}\Psi$ a better-conditioned operator (in an appropriate sense) than $\Psi$, for some applications where the feature matrix $\Psi$ has certain undesired properties. Finally, since $\mathbf{\Lambda}_{[N]}^{-\alpha}$ is applied to the residual $\Psi\boldsymbol{\theta} - \mathbf{y}$, we can

regard the new weighted optimization formulation (7) as the generalization of the classic least-squares formulation with a new loss function (a weighted norm) measuring the data mismatch.

Weighting optimization schemes such as (7) have been studied, implicitly or explicitly, in different settings (Needell et al., 2014; Byrd & Lipton, 2019; Engquist et al., 2020; Li, 2021; Yang et al., 2021). For instance, if we take $\beta = 0$, then we have a case where we rescale the classical least-squares loss function with the weight $\mathbf{\Lambda}_{[N]}^{-\alpha}$. If we take $\alpha = 1$, then this least-squares functional is equivalent to the loss function based on the $\mathcal{H}^{-1}$ norm, instead of the usual $L^2$ norm, of the mismatch between the target function $f_{\boldsymbol{\theta}}(x)$ and the learned model $f_{\widehat{\boldsymbol{\theta}}}(x)$. Based on the asymptotic equivalence between the quadratic Wasserstein metric and the $\mathcal{H}^{-1}$ semi-norm (on an appropriate functional space), this training problem is asymptotically equivalent to the same training problem based on a quadratic Wasserstein loss function; see for instance Engquist et al. (2020) for more detailed illustration on the connection. In the classical statistical inversion setting, $\mathbf{\Lambda}^{2\alpha}$ plays the role of the covariance matrix of the additive Gaussian random noise in the data (Kaipio & Somersalo, 2005). When the noise is sampled from mean-zero Gaussian distribution with covariance matrix $\mathbf{\Lambda}^{2\alpha}$, a standard maximum likelihood estimator (MLE) is often constructed as the minimizer of $(\Psi\boldsymbol{\theta} - \mathbf{y})^*\mathbf{\Lambda}_{[N]}^{-2\alpha}(\Psi\boldsymbol{\theta} - \mathbf{y}) = \|\mathbf{\Lambda}_{[N]}^{-\alpha}(\Psi\boldsymbol{\theta} - \mathbf{y})\|_2^2$.

The exact solution to (7), with $\mathbf{X}^+$ denoting the Moore–Penrose inverse of operator $\mathbf{X}$, is given by

$$\widehat{\boldsymbol{\theta}}_p^\delta = \mathbf{\Lambda}_{[p]}^{-\beta}\left(\mathbf{\Lambda}_{[N]}^{-\alpha}\Psi_{[N\times p]}\mathbf{\Lambda}_{[p]}^{-\beta}\right)^+\mathbf{\Lambda}_{[N]}^{-\alpha}\mathbf{y}^\delta. \tag{8}$$

In the rest of this paper, we analyze this training result and highlight the impact of the weight matrices $\mathbf{\Lambda}_{[N]}^{-\alpha}$ and $\mathbf{\Lambda}_{[N]}^{-\beta}$ in different regimes of the learning problem. We reproduce the classical bias-variance trade-off analysis in the weighted optimization framework. For that purpose, we utilize the linearity of the problem to decompose $\widehat{\boldsymbol{\theta}}_p^\delta$ as

$$\widehat{\boldsymbol{\theta}}_p^\delta = \mathbf{\Lambda}_{[p]}^{-\beta}(\mathbf{\Lambda}_{[N]}^{-\alpha}\Psi_{[N\times p]}\mathbf{\Lambda}_{[p]}^{-\beta})^+\mathbf{\Lambda}_{[N]}^{-\alpha}\mathbf{y} + \mathbf{\Lambda}_{[p]}^{-\beta}(\mathbf{\Lambda}_{[N]}^{-\alpha}\Psi_{[N\times p]}\mathbf{\Lambda}_{[p]}^{-\beta})^+\mathbf{\Lambda}_{[N]}^{-\alpha}(\mathbf{y}^\delta - \mathbf{y}), \tag{9}$$

where the first part is simply $\widehat{\boldsymbol{\theta}}_p$, the result of learning with noise-free data, while the second part is the contribution from the additive noise. We define the generalization error in this case as

$$\mathcal{E}_{\alpha,\beta}^\delta(P, p, N) = \mathbb{E}_{\boldsymbol{\theta},\delta}\left[\|f_{\boldsymbol{\theta}}(x) - f_{\widehat{\boldsymbol{\theta}}_p^\delta}(x)\|_{L^2([0,2\pi])}^2\right] = \mathbb{E}_{\boldsymbol{\theta},\delta}\left[\|\widehat{\boldsymbol{\theta}}_p^\delta - \widehat{\boldsymbol{\theta}}_p + \widehat{\boldsymbol{\theta}}_p - \boldsymbol{\theta}\|_2^2\right], \tag{10}$$

where the expectation is taken over the joint distribution of $\boldsymbol{\theta}$ and the random noise $\boldsymbol{\delta}$. By the standard triangle inequality, this generalization error is bounded by sum of the generalization error from training with noise-free data and the error caused by the noise. We will use this simple observation to bound the generalization errors when no exact formulas can be derived. We also look at the variance of the generalization error with respect to the random noise, which is

$$\mathrm{Var}_\delta(\mathbb{E}_{\boldsymbol{\theta}}[\|\widehat{\boldsymbol{\theta}}^\delta - \boldsymbol{\theta}\|_2^2]) := \mathbb{E}_{\boldsymbol{\delta}}[(\mathbb{E}_{\boldsymbol{\theta}}[\|\widehat{\boldsymbol{\theta}}^\delta - \boldsymbol{\theta}\|_2^2] - \mathbb{E}_{\boldsymbol{\theta},\boldsymbol{\delta}}[\|\widehat{\boldsymbol{\theta}}^\delta - \boldsymbol{\theta}\|_2^2])^2]. \tag{11}$$

In the rest of the work, we consider two parameter regimes of learning:
(i) In the *overparameterized regime*, we have the following setup of the parameters:

$$N < p \leq P, \quad \text{and,} \quad P = \mu N, \quad p = \nu N \quad \text{for some } \mu, \nu \in \mathbb{N} \text{ s.t. } \mu \geq \nu \gg 1. \tag{12}$$

(ii) In the *underparameterized regime*, we have the following scaling relations:

$$p \leq N \leq P, \quad \text{and,} \quad P = \mu N \quad \text{for some } \mu \in \mathbb{N}. \tag{13}$$

The formally-determined case of $p = N \leq P$ is included in both the overparameterized and the underparameterized regimes. We make the following assumptions throughout the work:

(**A-I**) The random noise $\boldsymbol{\delta}$ in the training data is additive in the sense that $\mathbf{y}^\delta = \mathbf{y} + \boldsymbol{\delta}$.

(**A-II**) The random vectors $\boldsymbol{\delta}$ and $\boldsymbol{\theta}$ are independent.

(**A-III**) The random noise $\boldsymbol{\delta} \sim \mathcal{N}(\mathbf{0}, \sigma\mathbf{I}_{[P]})$ for some constant $\sigma > 0$.

While assumptions (**A-I**) and (**A-II**) are essential, assumption (**A-III**) is only needed to simplify the calculations. Most of the results we obtain in this paper can be reproduced straightforwardly for the random noise $\boldsymbol{\delta}$ with any well-defined covariance matrix.

## 3 GENERALIZATION ERROR FOR TRAINING WITH NOISE-FREE DATA

We start with the problem of training with noise-free data. In this case, we utilize tools developed in Belkin et al. (2020) and Xie et al. (2020) to derive exact generalization errors. Our main objective is to compare the difference and similarity of the roles of the weight matrices $\mathbf{\Lambda}_{[N]}^{-\alpha}$ and $\mathbf{\Lambda}_{[p]}^{-\beta}$.

We have the following results on the generalization error. The proof is in Appendix A.1 and A.2.

**Theorem 3.1** (Training with noise-free data)**.** *Let $\boldsymbol{\delta} = \mathbf{0}$, and $\boldsymbol{\theta}$ be sampled with the properties in (3). Then the generalization error in the overparameterized regime (12) is:*

$$\mathcal{E}_{\alpha,\beta}^0(P,p,N) = 1 - 2c_\gamma \sum_{k=0}^{N-1} \frac{\sum_{\eta=0}^{\nu-1} t_{k+N\eta}^{-2\beta-2\gamma}}{\sum_{\eta=0}^{\nu-1} t_{k+N\eta}^{-2\beta}} + c_\gamma \sum_{k=0}^{N-1} \frac{(\sum_{\eta=0}^{\nu-1} t_{k+N\eta}^{-4\beta})(\sum_{\eta=0}^{\mu-1} t_{k+N\eta}^{-2\gamma})}{(\sum_{\eta=0}^{\nu-1} t_{k+N\eta}^{-2\beta})^2}, \quad (14)$$

*and the generalization error in the underparameterized regime (13) is:*

$$\mathcal{E}_{\alpha,\beta}^0(P,p,N) = c_\gamma \sum_{j=N}^{P-1} t_j^{-2\gamma} + c_\gamma \sum_{k=0}^{p-1}\sum_{\eta=1}^{\mu-1} t_{k+N\eta}^{-2\gamma} - c_\gamma \sum_{k=p}^{N-1}\sum_{\eta=1}^{\mu-1} t_{k+N\eta}^{-2\gamma} + N \sum_{i,j=0}^{N-p-1} \frac{\widetilde{e}_{ij}^{(N)}\widehat{e}_{ji}^{(N)}}{\Sigma_{ii}\Sigma_{jj}}, \tag{15}$$

*where $\{t_j\}_{j=0}^{P-1}$ and $c_\gamma$ are those introduced in (3) and (4), while*

$$\widetilde{e}_{ij}^{(N)} = \sum_{k=0}^{N-1} t_k^{2\alpha}\overline{U}_{ki}U_{kj}, \quad \widehat{e}_{ij}^{(N)} = \sum_{k'=0}^{N-p-1}(c_\gamma t_{p+k'}^{-2\gamma} + \chi_{p+k'})\overline{V}_{ik'}V_{jk'}, \quad 0 \le i,j \le N-p-1,$$

*with $\mathbf{U\Sigma V}^*$ being the singular value decomposition of $\mathbf{\Lambda}_{[N]}^\alpha \Psi_{[N]\backslash[p]}$ and $\{\chi_m\}_{m=0}^{N-1}$ defined as*

$$\chi_m = \sum_{k=0}^{N-1}\left(\sum_{\eta=1}^{\mu-1} t_{k+N\eta}^{-2\gamma}\right)\left(\frac{1}{N}\sum_{j=0}^{N-1}\omega_N^{(m-k)j}\right), \quad 0 \le m \le N-1.$$

We want to emphasize that the generalization errors we obtained in Theorem 3.1 are for the weighted optimization formulation (7) where we have an additional weight matrix $\mathbf{\Lambda}_{[N]}^{-\alpha}$ compared to the formulation in Xie et al. (2020), even though our results look similar to the previous results of Belkin et al. (2020) and Xie et al. (2020). Moreover, we kept the weight matrix $\mathbf{\Lambda}_{[p]}^{-\beta}$ in the underparameterized regime, which is different from the setup in Xie et al. (2020) where the same weight matrix was removed in this regime. The reason for keeping $\mathbf{\Lambda}_{[p]}^{-\beta}$ in the underparameterized regime will become more obvious in the case of training with noisy data, as we will see in the next section.

Here are some key observations from the above results, which, we emphasize again, are obtained in the setting where the optimization problems are solved exactly, and the data involved contain no random noise. The conclusion will differ when data contain random noise or when optimization problems cannot be solved exactly.

First, the the weight matrix $\mathbf{\Lambda}_{[p]}^{-\beta}$ only matters in the overparameterized regime while $\mathbf{\Lambda}_{[N]}^{-\alpha}$ only matters in the underparameterized regime. In the overparameterized regime, the weight $\mathbf{\Lambda}_{[p]}^{-\beta}$ forces the inversion procedure to focus on solutions that are biased toward the low-frequency modes. In the underparameterized regime, the matrix $\mathbf{\Lambda}_{[N]}^{-\alpha}$ re-weights the frequency content of the "residual" (data mismatch) before it is backprojected into the learned parameter $\widehat{\boldsymbol{\theta}}$. Using the scaling $P = \mu N$ and $p = \nu N$ in the overparameterized regime, and the definition of $c_\gamma$, we can verify that when $\alpha = \beta = 0$, the generalization error reduces to

$$\mathcal{E}_{0,0}^0(P,p,N) = 1 - \frac{N}{p} + \frac{2N}{p}c_\gamma \sum_{j=p}^{P-1} t_j^{-2\gamma} = 1 + \frac{N}{p} - \frac{2N}{p}c_\gamma \sum_{j=0}^{p-1} t_j^{-2\gamma}. \tag{16}$$

This is given in Xie et al. (2020, Theorem 1).

Second, when the learning problem is formally determined, i.e., when $p = N$, neither weight matrices play a role when the training data are generated from the true model with no additional random noise and the minimizer can be found exactly. The generalization error simplifies to

$$\mathcal{E}_{\alpha,\beta}^0(P,p,N) = 2c_\gamma \sum_{j=p}^{P-1} t_j^{-2\gamma}. \tag{17}$$

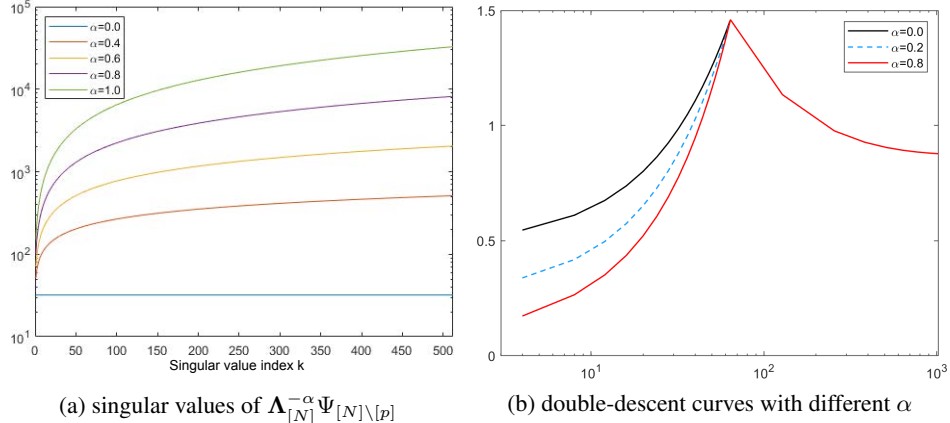

(a) singular values of $\mathbf{\Lambda}_{[N]}^{-\alpha}\Psi_{[N]\setminus[p]}$      (b) double-descent curves with different $\alpha$

Figure 1: Left: singular values of $\mathbf{\Lambda}_{[N]}^{-\alpha}\Psi_{[N]\setminus[p]}$ for a system with $N = 1024$ and $p = 512$. Shown are singular values for $\alpha = 0$, $\alpha = 0.4$, $\alpha = 0.6$, $\alpha = 0.8$, and $\alpha = 1.0$; Right: double-descent curves for the generalization error $\mathcal{E}_{\alpha,\beta}^0$ for the cases of $N = 64$, $\gamma = 0.3$, $\beta = 0.3$, and $\alpha = 0,\ 0.3,\ 0.8$.

This is not surprising as, in this case, $\Psi$ is invertible (because it is a unitary matrix scaled by the constant $N$). The true solution to $\Psi\boldsymbol{\theta} = \mathbf{y}$ is simply $\boldsymbol{\theta} = \Psi^{-1}\mathbf{y}$. The weight matrices in the optimization problem are invertible and therefore do not change the true solution of the problem. The generalization error, in this case, is therefore only due to the Fourier modes that are not learned from the training data, i.e., modes $p$ to $P - 1$.

While it is obvious from the formulas for the generalization error that the weight matrix $\mathbf{\Lambda}_{[N]}^{-\alpha}$ indeed plays a role in the underparameterized regime, we show in Figure 1a the numerical calculation of the singular value decomposition of the matrix $\mathbf{\Lambda}_{[N]}^{\alpha}\Psi_{[N]\setminus[p]}$ for the case of $(N, p) = (1024, 512)$. The impact of $\alpha$ can be seen by comparing the singular values to their correspondence in the $\alpha = 0$ case (where all the singular values are the same and equal to $\sqrt{N}$). When the system size is large (in this case $N = 1024$), even a small $\alpha$ (e.g., $\alpha = 0.4$) can significantly impact the result. In Figure 1b, we plot the theoretical prediction of $\mathcal{E}_{\alpha,\beta}^0$ in Theorem 3.1 to demonstrate the double-descent phenomenon observed in the literature on the RFF model. We emphasize again that in this particular noise-free setup with perfectly solved minimization problem by the pseudoinverse, $\mathbf{\Lambda}_{[N]}^{-\alpha}$ only plays a role in the underparameterized regime as can be seen from the double-descent curves.

**Selecting $p$ to minimize generalization error.** It is clear (and also expected) from (16) and (17) that, in the cases of $p = N$ or $\alpha = \beta = \gamma = 0$, the generalization error decreases monotonically with respect to the number of modes learned in the training process. One should then learn as many Fourier coefficients as possible. When $p \neq N$ or $\alpha,\ \beta \neq 0$, $\mathcal{E}_{\alpha,\beta}^0(P, p, N)$, for fixed $P$ and $N$, does not change monotonically with respect to $p$ anymore. In such a situation, we can choose the $p$ values that minimize the generalization error and perform learning with these $p$ values.

## 4    Error bounds for training with noisy data

In this section, we study the more realistic setting of training with noisy data. The common practice is that when we solve the minimization problem in such a case, we should avoid overfitting the model to the data by stopping the optimization algorithm early at an appropriate level of values depending on the noise level in the data, but see Bartlett et al. (2020); Li et al. (2021b) for some analysis in the direction of "benign overfitting". When training with noisy data, the impact of the weight matrices $\mathbf{\Lambda}_{[N]}^{-\alpha}$ and $\mathbf{\Lambda}_{[p]}^{-\beta}$ on the generalization error becomes more obvious. In fact, both weight matrices play non-negligible roles in the overparameterized and the underparameterized regimes, respectively.

We start with the circumstance where we still match the data perfectly for each realization of the noise in the data. The result is summarized as follows.

**Lemma 4.1** (Training with noisy data: exact error). *Under the assumptions* (**A-I**)-(**A-III**), *the generalization error* $\mathcal{E}^{\delta}_{\alpha,\beta}(P,p,N)$ *is given as*

$$\mathcal{E}^{\delta}_{\alpha,\beta}(P,p,N) = \mathcal{E}^{0}_{\alpha,\beta}(P,p,N) + \mathcal{E}_{\text{noise}}(P,p,N)\,,$$

*where* $\mathcal{E}^{0}_{\alpha,\beta}(P,p,N)$ *is the generalization error from training with noise-free data given in Theorem 3.1 and* $\mathcal{E}_{\text{noise}}(P,p,N)$ *is the error due to noise. The error due to noise and the variance of the generalization error with respect to noise are respectively*

$$\mathcal{E}_{\text{noise}}(P,p,N) = \sigma^2 \sum_{k=0}^{N-1} \frac{\sum_{\eta=0}^{\nu-1} t_{k+N\eta}^{-4\beta}}{\left[\sum_{\eta=0}^{\nu-1} t_{k+N\eta}^{-2\beta}\right]^2}, \ \text{Var}_{\delta}\Big(\mathbb{E}_{\boldsymbol{\theta}}[\|\boldsymbol{\theta}^{\delta}-\boldsymbol{\theta}\|_2^2]\Big) = \frac{2\sigma^4}{N^2} \sum_{k=0}^{N-1} \frac{\left[\sum_{\eta=0}^{\nu-1} t_{k+N\eta}^{-4\beta}\right]^2}{\left[\sum_{\eta=0}^{\nu-1} t_{k+N\eta}^{-2\beta}\right]^4}.$$

(18)

*in the overparameterized regime* (12)*, and*

$$
\begin{aligned}
\mathcal{E}_{\text{noise}}(P,p,N) &= \sigma^2 \Big(\frac{2p-N}{N} + \sum_{j=0}^{N-p-1} \frac{\widetilde{e}_{jj}^{(N)}}{\Sigma_{jj}^2}\Big), \ p > N/2, \\
\text{Var}_{\delta}\Big(\mathbb{E}_{\boldsymbol{\theta}}[\|\boldsymbol{\theta}^{\delta}-\boldsymbol{\theta}\|_2^2]\Big) &= 2\sigma^4 \Big(\frac{2p-N}{N^2} + \sum_{i,j=0}^{N-p-1} \frac{\widetilde{e}_{ij}^{(N)}\widetilde{e}_{ji}^{(N)}}{\Sigma_{ii}^2 \Sigma_{jj}^2}\Big),
\end{aligned}
$$

(19)

*in the underparameterized regime* (13)*.*

The proof of this lemma is documented in Appendix A.3. Note that due to the assumption that the noise $\boldsymbol{\delta}$ and the coefficient $\boldsymbol{\theta}$ are independent, the noise-averaged generalization errors are split exactly into two separate parts: the part due to $\boldsymbol{\theta}$ and the part due to $\boldsymbol{\delta}$. The coupling between them disappears. This also leads to the fact that the variance of the generalization error with respect to noise only depends on the noise distribution (instead of both the noise variance and the $\boldsymbol{\theta}$ variance).

For the impact of noise on the generalization error, we see again that the impact of $\boldsymbol{\Lambda}_{[P]}^{-\beta}$ is only seen in the overparameterized regime while that of $\boldsymbol{\Lambda}_{[N]}^{-\alpha}$ is only seen in the underparameterized regime. This happens because we assume that we can solve the optimization problem exactly for each realization of the noisy data. In the overparameterized regime, when $\beta = 0$, we have that $\mathcal{E}_{\text{noise}} = N\sigma^2/p$ as expected. In such a case, the variance reduces to $\text{Var}_{\delta}(\mathbb{E}_{\boldsymbol{\theta}}[\|\boldsymbol{\theta}^{\delta} - \boldsymbol{\theta}\|_2^2]) = 2N\sigma^4/p^2$. In the underparameterized regime, when $\alpha = 0$, we have that $\mathcal{E}_{\text{noise}} = p\sigma^2/N$. The corresponding variance reduces to $\text{Var}_{\delta}(\mathbb{E}_{\boldsymbol{\theta}}[\|\boldsymbol{\theta}^{\delta} - \boldsymbol{\theta}\|_2^2]) = 2p\sigma^4/N^2$.

In the limit when $p \to N$, that is, in the formally determined regime, the mean generalization error due to random noise (resp. the variance of the generalization error with respect to noise) converges to the same value $\mathcal{E}_{\text{noise}}(P,p,N) = \sigma^2$ (resp. $\text{Var}_{\delta}(\mathbb{E}_{\boldsymbol{\theta}}[\|\boldsymbol{\theta}^{\delta} - \boldsymbol{\theta}\|_2^2]) = 2\sigma^4/N$) from both the overparameterized and the underparameterized regimes. This is the classical result in statistical learning theory (Kaipio & Somersalo, 2005).

The explicit error characterization above is based on the assumption that we solve the optimization exactly by minimizing the mismatch to 0 in the learning process. In practical applications, it is often the case that we stop the minimization process when the value of the loss function reaches a level that is comparable to the size of the noise in the data (normalized by the size of the target function, for instance). We now present the generalization error bounds for such a case.

**Theorem 4.2** (Training with noisy data: error bounds). *In the same setup as Lemma 4.1, for fixed $N$, we have the following general bound on the generalization error when $p$ is sufficiently large:*

$$\mathcal{E}^{\delta}_{\alpha,\beta}(P,p,N) \lesssim p^{-2\widehat{\alpha}}\mathbb{E}_{\boldsymbol{\delta}}[\|\boldsymbol{\delta}\|^2_{2,\boldsymbol{\Lambda}_{[N]}^{-\alpha}}] + p^{-2\beta}\mathbb{E}_{\boldsymbol{\theta}}[\|\boldsymbol{\theta}\|^2_{2,\boldsymbol{\Lambda}_{[p]}^{-\beta}}]\,,$$

*where* $\widehat{\alpha} := \alpha + 1/2$ *(resp* $\widehat{\alpha} := \alpha$*) in the overparameterized (resp. underparameterized) regime. When* $\widehat{\alpha} \geq 0$*, the error decreases monotonically with respect to $p$. When* $\widehat{\alpha} < 0$*, the bound is minimized by selecting*

$$p \sim \Big(\mathbb{E}_{\boldsymbol{\delta}}[\|\boldsymbol{\delta}\|^2_{2,\boldsymbol{\Lambda}_{[N]}^{-\alpha}}]^{-1}\mathbb{E}_{\boldsymbol{\theta}}[\|\boldsymbol{\theta}\|^2_{2,\boldsymbol{\Lambda}_{[p]}^{-\beta}}]\Big)^{\frac{1}{2(\beta-\widehat{\alpha})}},$$

(20)

*in which case we have*

$$\mathcal{E}^{\delta}_{\alpha,\beta}(P,p,N) \lesssim \mathbb{E}_{\boldsymbol{\theta}}[\|\boldsymbol{\theta}\|^2_{2,\boldsymbol{\Lambda}_{[p]}^{-\beta}}]^{\frac{-2\widehat{\alpha}}{2(\beta-\widehat{\alpha})}} \mathbb{E}_{\boldsymbol{\delta}}[\|\boldsymbol{\delta}\|^2_{2,\boldsymbol{\Lambda}_{[N]}^{-\alpha}}]^{\frac{2\beta}{2(\beta-\widehat{\alpha})}}.$$

(21)

Note that the calculations in (20) and (21) are only to the leading order. We neglected the contribution from the term involving $\gamma$. Also, the quantities in expectations can be simplified. We avoided doing so to make these quantities easily recognizable as they are the main quantities of interests.

In the underparameterized regime, that is, the regime of reconstruction, classical statistical inversion theory shows that it is statistically beneficial to introduce the weight matrix $\mathbf{\Lambda}_{[N]}^{-\alpha}$ that is related to the covariance matrix of the noise in the data (Kaipio & Somersalo, 2005) (see also Bal & Ren (2009) for an adaptive way of adjusting the weight matrix for some specific applications). Our result here is consistent with the classical result as we could see that if we take $\mathbf{\Lambda}_{[N]}^{-\alpha}$ to be the inverse of the covariance matrix for the noise distribution, the size of the noise contribution in the generalization error is minimized.

Theorem 4.2 shows that, in the case of training with noisy data, the $\alpha$ parameter can be tuned to reduce the generalization error of the learning process just as the $\beta$ parameter in the weight matrix $\mathbf{\Lambda}_{[P]}^{-\beta}$. Moreover, this result can serve as a guidance on the selection of the number of features to be pursued in the training process to minimize the generalization error, depending on the level of random noise in the training data as well as the regime of the problem.

## 5 EXTENSION TO GENERAL FEATURE REGRESSION

The results in the previous sections, even though are obtained for the specific form of Fourier feature regression, also hold in more general settings. Let $f_{\boldsymbol{\theta}}$ be a general random feature model of the form

$$f_{\boldsymbol{\theta}}(x) = \sum_{k=0}^{P-1} \theta_k \varphi_k(x), \quad x \in X, \tag{22}$$

constructed from a family of orthonormal features $\{\varphi_k\}_{k=0}^{P}$ in $L^2(X)$. Let $\Psi$ be the feature matrix constructed from the dataset $\{x_j, y_j\}_{j=0}^{N-1}$:

$$(\Psi)_{jk} = \varphi_k(x_j), \quad 0 \le j, k \le P - 1.$$

We can then apply the same weighted optimization framework (7) to this general model. As in the case of RFF model, the data in the learning process allows decomposition

$$\mathbf{y}^{\boldsymbol{\delta}} = \Psi_{[N] \times [p]} \boldsymbol{\theta}_{[p]} + \Psi_{[N] \times ([P] \backslash [p])} \boldsymbol{\theta}_{[P] \backslash [p]} + \boldsymbol{\delta} \,.$$

The learning process only aims to learn the first $p$ modes, i.e., $\boldsymbol{\theta}_{[p]}$, indicating that the effective noise that we backpropagate into $\boldsymbol{\theta}_{[p]}$ in the learning process is $\Psi_{[N] \times ([P] \backslash [p])} \boldsymbol{\theta}_{[P] \backslash [p]} + \boldsymbol{\delta}$. The frequency contents of this effective noise can come from the true noise $\boldsymbol{\delta}$, the part of Fourier modes that we are not learning (i.e., $\boldsymbol{\theta}_{[P] \backslash [p]}$), or even the feature matrix $\Psi_{[N] \times ([P] \backslash [p])}$. For the RFF model, the feature matrix $\Psi_{[N] \times [N]}$ is unitary after being normalized by $1/\sqrt{N}$. Therefore, all its singular values are homogeneously $\sqrt{N}$. For learning with many other features in practical applications, we could have feature matrices $\Psi_{[N] \times [N]}$ with fast-decaying singular values. This, on one hand, means that $\Psi_{[N] \times ([P] \backslash [p])} \boldsymbol{\theta}_{[P] \backslash [p]}$ will decay even faster than $\boldsymbol{\theta}_{[P] \backslash [p]}$, making its impact on the learning process smaller. On the other hand, $\Psi_{[N] \times [N]}$ having fast-decaying singular values will make learning $\boldsymbol{\theta}_{[p]}$ harder (since it is less stable).

The impact of the weighting scheme on the generalization error as an expectation over $\boldsymbol{\theta}$ and $\boldsymbol{\delta}$ (defined in Lemma 4.1) is summarized in Theorem 5.1. Its proof is in Appendix A.4. Note that here we do not assume any specific structure on the distributions of $\boldsymbol{\theta}$ and $\boldsymbol{\delta}$ except there independence.

**Theorem 5.1.** *Let $\Psi = \mathbf{U}\boldsymbol{\Sigma}\mathbf{V}^*$ be the singular value decomposition of $\Psi$. Under the assumptions (A-I)-(A-III), assume further that $\Sigma_{kk} \sim t_k^{-\zeta}$ for some $\zeta > 0$. Then the generalization error of training, using the weighted optimization scheme (7) with $\mathbf{\Lambda}_{[N]}^{-\alpha}$ replaced with $\mathbf{\Lambda}_{[N]}^{-\alpha}\mathbf{U}^*$ and $\mathbf{\Lambda}_{[P]}^{-\beta}$ replaced with $\mathbf{V}\mathbf{\Lambda}_{[P]}^{-\beta}$, satisfies*

$$\mathcal{E}_{\alpha, \beta}^{\boldsymbol{\delta}}(P, p, N) \lesssim p^{2(\zeta - \alpha)} \mathbb{E}_{\boldsymbol{\delta}}[\|\boldsymbol{\delta}\|_{2, \mathbf{\Lambda}_{[N]}^{-\alpha}}^2] + p^{-2\beta} \mathbb{E}_{\boldsymbol{\theta}}[\|\boldsymbol{\theta}\|_{2, \mathbf{\Lambda}_{[p]}^{-\beta}}^2] \,,$$

*in the asymptotic limit $p \sim N \sim P \to \infty$. The bound is minimized, as a function of $p$, when*

$$p \sim (\mathbb{E}_{\boldsymbol{\delta}}[\|\boldsymbol{\delta}\|_{2, \mathbf{\Lambda}_{[N]}^{-\alpha}}^2]^{-1} \mathbb{E}_{\boldsymbol{\theta}}[\|\boldsymbol{\theta}\|_{2, \mathbf{\Lambda}_{[p]}^{-\beta}}^2])^{\frac{1}{2(\zeta + \beta - \alpha)}} \,, \tag{23}$$

*in which case we have that*

$$\mathcal{E}_{\alpha,\beta}^{\delta}(P,p,N) \lesssim \mathbb{E}_{\boldsymbol{\theta}}[\|\boldsymbol{\theta}\|_{2,\mathbf{\Lambda}_{[p]}^{-\beta}}^{2}]^{\frac{\zeta-\alpha}{(\zeta+\beta-\alpha)}} \mathbb{E}_{\boldsymbol{\delta}}[\|\boldsymbol{\delta}\|_{2,\mathbf{\Lambda}_{[N]}^{-\alpha}}^{2}]^{\frac{\beta}{(\zeta+\beta-\alpha)}}. \tag{24}$$

On the philosophical level, the result says that when the learning model is smoothing, that is, when the singular values of $\Psi$ decays fast, we can select the appropriate weight matrix to compensate the smoothing effect so that the generalization error is optimized. The assumption that we have access to the exact form of the singular value decomposition of the feature matrix is only made to trivialize the calculations, and is by no means essential. When the optimization algorithm is stopped before perfect matching can be achieve, the fitting from the weighted optimization scheme generated a smoother approximation (with a smaller $p$, according to (23), than what we would obtain with a regular least-squares minimization).

Feature matrices with fast-decaying singular values are ubiquitous in applications. In Appendix B, we provide some discussions on the applicability of this result in understanding general kernel learning (Amari & Wu, 1999; Kamnitsas et al., 2018; Jean et al., 2018; Owhadi & Yoo, 2019; Bordelon et al., 2020; Canatar et al., 2021) and learning with simplified neural networks.

## 6 Concluding remarks

In this work, we analyzed the impact of weighted optimization on the generalization capability of feature regression with noisy data for the RFF model and generalized the result to the case of feature regression and kernel regression. For the RFF model, we show that the proposed weighting scheme (7) allows us to minimize the impact of noise in the training data while emphasizing the corresponding features according to the *a priori* knowledge we have on the distribution of the features. In general, emphasizing low-frequency features (i.e., searching for smoother functions) provide better generalization ability.

While what we analyze in this paper is mainly motivated by the machine learning literature, the problem of fitting models such as the RFF model (1) to the observed data is a standard inverse problem that has been extensively studied (Engl et al., 1996; Tarantola, 2005). The main focus of classical inversion theory is on the case when $\Psi_{[N \times p]}$ is at least rank $p$ so that there is a unique least-squares solution to the equation $\Psi\boldsymbol{\theta} = \mathbf{y}$ for any given dataset $\mathbf{y}$. This corresponds to the learning problem we described above in the underparameterized regime.

In general, weighting the least-squares allows us to have an optimization algorithm that prioritize the modes that we are interested in during the iteration process. This can be seen as a preconditioning strategy from the computational optimization perspective.

It would be of great interest to derive a rigorous theory for the weighted optimization (7) for general non-convex problems (note that while the RFF model itself is nonlinear from input $x$ to output $f_{\boldsymbol{\theta}}(x)$, the regression problem is linear). Take a general nonlinear model of the form

$$F(\mathbf{x}; \boldsymbol{\theta}) = \mathbf{y},$$

where $F$ is nonlinear with respect to $\boldsymbol{\theta}$. For instance, $F$ could be a deep neural network with $\boldsymbol{\theta}$ representing the parameters of the network (for instance, the weight matrices and bias vectors at different layers). We formulate the learning problem with a weighted least-squares as

$$\widehat{\boldsymbol{\theta}} = \mathbf{\Lambda}_{[p]}^{-\beta}\widehat{\mathbf{w}}, \quad \text{with} \quad \widehat{\mathbf{w}} = \underset{\mathbf{w}}{\arg\min} \|\mathbf{\Lambda}_{[N]}^{-\alpha}(F(\mathbf{x}; \mathbf{\Lambda}_{[p]}^{-\beta}\mathbf{w}) - \mathbf{y})\|_2^2,$$

where the weight matrices are used to control the smoothness of the gradient of the optimization procedure. The linearized problem for the learning of $\mathbf{w}$ gives a problem of the form $\Psi\widetilde{\boldsymbol{\theta}} = \widetilde{\mathbf{y}}$ with

$$\Psi = (\mathbf{\Lambda}_{[N]}^{s}F(\mathbf{x}; \mathbf{\Lambda}_{[p]}^{\beta}\mathbf{w}))^*(\mathbf{\Lambda}_{[p]}^{\beta})^*(F')^*(\mathbf{x}; \mathbf{\Lambda}_{[p]}^{\beta}\mathbf{w})(\mathbf{\Lambda}_{[N]}^{\beta})^*, \quad \widetilde{\mathbf{y}} = (\mathbf{\Lambda}_{[p]}^{\beta})^*(F')^*(\mathbf{x}; \mathbf{\Lambda}_{[p]}^{\beta}\mathbf{w})(\mathbf{\Lambda}_{[N]}^{s})^*\mathbf{y}.$$

For many practical applications in computational learning and inversion, $F'$ (with respect to $\boldsymbol{\theta}$) has the properties we need for Theorem 5.1 to hold. Therefore, a local theory could be obtained. The question is, can we show that the accumulation of the results through an iterative optimization procedure (e.g., a stochastic gradient descent algorithm) does not destroy the local theory so that the conclusions we have in this work would hold globally? We believe a thorough analysis along the lines of the recent work of Ma & Ying (2021) is possible with reasonable assumptions on the convergence properties of the iterative process.

ACKNOWLEDGMENTS

This work is partially supported by the National Science Foundation through grants DMS-1620396, DMS-1620473, DMS-1913129, and DMS-1913309. Y. Yang acknowledges supports from Dr. Max Rössler, the Walter Haefner Foundation and the ETH Zürich Foundation. This work was done in part while Y. Yang was visiting the Simons Institute for the Theory of Computing in Fall 2021. The authors would like to thank Yuege Xie for comments that helped us correct a mistake in an earlier version of the paper.

## SUPPLEMENTARY MATERIAL

Supplementary material for the paper "A Generalized Weighted Optimization Method for Computational Learning and Inversion" is organized as follows.

## A    PROOF OF MAIN RESULTS

We provide here the proofs for all the results that we summarized in the main part of the paper. To simplify the presentation, we introduce the following notations. For the feature matrix $\Psi \in \mathbb{C}^{N \times P}$, we denote by

$$\Psi_{\mathbb{T}} \in \mathbb{C}^{N \times p} \quad \text{and} \quad \Psi_{\mathbb{T}^c} \in \mathbb{C}^{N \times (P-p)}$$

the submatrices of $\Psi$ corresponding to the first $p$ columns and the last $P - p$ columns respectively. In the underparameterized regime (13), we will also use the following submatrices

$$\Psi_{[N]} \in \mathbb{C}^{N \times N} \quad \text{and} \quad \Psi_{[N] \setminus \mathbb{T}} \in \mathbb{C}^{N \times (N-p)},$$

corresponding to the first $N$ columns of $\Psi$ and the last $N - p$ columns of $\Psi_{[N]}$, respectively. We will use $\Psi^*$ to denote the Hermitian transpose of $\Psi$. For a $P \times P$ diagonal matrix $\boldsymbol{\Lambda}$, $\boldsymbol{\Lambda}_{\mathbb{T}}$ ($\equiv \boldsymbol{\Lambda}_{[p]}$) and $\boldsymbol{\Lambda}_{\mathbb{T}^c}$ denote respectively the diagonal matrices of sizes $p \times p$ and $(P-p) \times (P-p)$ that contain the first $p$ and the last $P - p$ diagonal elements of $\boldsymbol{\Lambda}$. Matrix $\boldsymbol{\Lambda}_{[N]}$ is of size $N \times N$ and used to denote the first $N$ diagonal elements of $\boldsymbol{\Lambda}$. For any column vector $\boldsymbol{\theta} \in \mathbb{C}^{P \times 1}$, $\boldsymbol{\theta}_{\mathbb{T}}$ and $\boldsymbol{\theta}_{\mathbb{T}^c}$ denote respectively the column vectors that contain the first $p$ elements and the last $P - p$ elements of $\boldsymbol{\theta}$.

To study the impact of the weight matrices $\boldsymbol{\Lambda}_{[N]}^{-\alpha}$ and $\boldsymbol{\Lambda}_{[p]}^{-\beta}$, we introduce the re-scaled version of the feature matrix $\Psi$, denoted by $\Phi$, as

$$\Phi := \boldsymbol{\Lambda}_{[N]}^{-\alpha} \Psi \boldsymbol{\Lambda}_{[P]}^{-\beta} . \tag{25}$$

In terms of the notations above, we have that

$$\Phi_{\mathbb{T}} := \boldsymbol{\Lambda}_{[N]}^{-\alpha} \Psi_{\mathbb{T}} \boldsymbol{\Lambda}_{\mathbb{T}}^{-\beta} , \quad \text{and} \quad \Phi_{\mathbb{T}^c} := \boldsymbol{\Lambda}_{[N]}^{-\alpha} \Psi_{\mathbb{T}^c} \boldsymbol{\Lambda}_{\mathbb{T}^c}^{-\beta} . \tag{26}$$

For a given column vector $\boldsymbol{\theta} \in \mathbb{C}^d$ and a real-valued square-matrix $\mathbf{X} \in \mathbb{R}^{d \times d}$, we denote by

$$\|\boldsymbol{\theta}\|_{2,\mathbf{X}} := \|\mathbf{X}\boldsymbol{\theta}\|_2 = \sqrt{(\mathbf{X}\boldsymbol{\theta})^* \mathbf{X}\boldsymbol{\theta}}$$

the $\mathbf{X}$-weighted 2-norm of $\boldsymbol{\theta}$. We will not differentiate between finite- and infinite-dimensional vectors. In the infinite-dimensional case, we simply understand the 2-norm as the usual $\ell^2$-norm.

We first derive a general form for the generalization error for training with $\boldsymbol{\theta}$ sampled from a distribution with given diagonal covariance matrix. It is the starting point for most of the calculations in this paper. The calculation procedure is similar to that of Lemma 1 of Xie et al. (2020). However, our result is for the case with the additional weight matrix $\boldsymbol{\Lambda}_{[N]}^{-\alpha}$.

### A.1    PROOFS OF LEMMA A.1-LEMMA A.3

**Lemma A.1** (General form of $\mathcal{E}_{\alpha,\beta}^0(P, p, N)$)**.** *Let* $\mathbf{K} \in \mathbb{C}^{P \times P}$ *be a diagonal matrix, and* $\boldsymbol{\theta}$ *be drawn from a distribution such that*

$$\mathbb{E}_{\boldsymbol{\theta}}[\boldsymbol{\theta}] = \mathbf{0}, \qquad \mathbb{E}_{\boldsymbol{\theta}}[\boldsymbol{\theta}\boldsymbol{\theta}^*] = \mathbf{K} . \tag{27}$$

*Then the generalization error for training with weighted optimization* (7) *can be written as*

$$\mathcal{E}_{\alpha,\beta}^0(P, p, N) = \mathrm{tr}(\mathbf{K}) + \mathcal{P}_{\alpha,\beta} + \mathcal{Q}_{\alpha,\beta}, \tag{28}$$

*where*

$$\mathcal{P}_{\alpha,\beta} = \mathrm{tr}\big(\Phi_{\mathbb{T}}^+\Phi_{\mathbb{T}}\boldsymbol{\Lambda}_{\mathbb{T}}^{-2\beta}\Phi_{\mathbb{T}}^+\Phi_{\mathbb{T}}\boldsymbol{\Lambda}_{\mathbb{T}}^\beta\mathbf{K}_{\mathbb{T}}\boldsymbol{\Lambda}_{\mathbb{T}}^\beta\big) - 2\mathrm{tr}\big(\mathbf{K}_{\mathbb{T}}\Phi_{\mathbb{T}}^+\Phi_{\mathbb{T}}\big),$$

*and*

$$\mathcal{Q}_{\alpha,\beta} = \mathrm{tr}\big((\Phi_{\mathbb{T}}^+)^*\boldsymbol{\Lambda}_{\mathbb{T}}^{-2\beta}\Phi_{\mathbb{T}}^+\Phi_{\mathbb{T}^c}\boldsymbol{\Lambda}_{\mathbb{T}^c}^\beta\mathbf{K}_{\mathbb{T}^c}\boldsymbol{\Lambda}_{\mathbb{T}^c}^\beta\Phi_{\mathbb{T}^c}^*\big),$$

*with $\Phi_{\mathbb{T}}$ and $\Phi_{\mathbb{T}^c}$ given in* (26).

*Proof.* We introduce the new variables

$$\mathbf{w} = \boldsymbol{\Lambda}_{[P]}^\beta\boldsymbol{\theta}, \text{ and } \mathbf{z} = \boldsymbol{\Lambda}_{[N]}^{-\alpha}\mathbf{y}.$$

We can then write the solution to the weighted least-square problem (7) as

$$\widehat{\mathbf{w}}_{\mathbb{T}} = \Phi_{\mathbb{T}}^+\mathbf{z}, \text{ and } \widehat{\mathbf{w}}_{\mathbb{T}^c} = \mathbf{0},$$

where $\Phi_{\mathbb{T}}^+$ is the Moore–Penrose inverse of $\Phi_{\mathbb{T}}$. Using the fact that the data $\mathbf{y}$ contain no random noise, we write

$$\mathbf{y} = \Psi_{\mathbb{T}}\boldsymbol{\Lambda}_{\mathbb{T}}^{-\beta}\mathbf{w}_{\mathbb{T}} + \Psi_{\mathbb{T}^c}\boldsymbol{\Lambda}_{\mathbb{T}^c}^{-\beta}\mathbf{w}_{\mathbb{T}^c}, \text{ and } \mathbf{z} = \boldsymbol{\Lambda}_{[N]}^{-\alpha}\mathbf{y} = \Phi_{\mathbb{T}}\mathbf{w}_{\mathbb{T}} + \Phi_{\mathbb{T}^c}\mathbf{w}_{\mathbb{T}^c}.$$

Therefore, we have, following simple linear algebra, that

$$
\begin{aligned}
\|\widehat{\boldsymbol{\theta}} - \boldsymbol{\theta}\|_2^2 &= \|\boldsymbol{\Lambda}_{\mathbb{T}}^{-\beta}(\widehat{\mathbf{w}}_{\mathbb{T}} - \mathbf{w}_{\mathbb{T}})\|_2^2 + \|\boldsymbol{\Lambda}_{\mathbb{T}^c}^{-\beta}(\widehat{\mathbf{w}}_{\mathbb{T}^c} - \mathbf{w}_{\mathbb{T}^c})\|_2^2 \\
&= \|\boldsymbol{\Lambda}_{\mathbb{T}}^{-\beta}\Phi_{\mathbb{T}}^+(\Phi_{\mathbb{T}}\mathbf{w}_{\mathbb{T}} + \Phi_{\mathbb{T}^c}\mathbf{w}_{\mathbb{T}^c}) - \boldsymbol{\Lambda}_{\mathbb{T}}^{-\beta}\mathbf{w}_{\mathbb{T}}\|_2^2 + \|\boldsymbol{\Lambda}_{\mathbb{T}^c}^{-\beta}\mathbf{w}_{\mathbb{T}^c}\|_2^2 \\
&= \|\boldsymbol{\Lambda}_{\mathbb{T}}^{-\beta}\Phi_{\mathbb{T}}^+\Phi_{\mathbb{T}^c}\mathbf{w}_{\mathbb{T}^c} - \boldsymbol{\Lambda}_{\mathbb{T}}^{-\beta}(\mathbf{I}_{\mathbb{T}} - \Phi_{\mathbb{T}}^+\Phi_{\mathbb{T}})\mathbf{w}_{\mathbb{T}}\|_2^2 + \|\boldsymbol{\Lambda}_{\mathbb{T}^c}^{-\beta}\mathbf{w}_{\mathbb{T}^c}\|_2^2. \quad (29)
\end{aligned}
$$

Next, we make the following expansions:

$$\|\boldsymbol{\Lambda}_{\mathbb{T}}^{-\beta}\Phi_{\mathbb{T}}^+\Phi_{\mathbb{T}^c}\mathbf{w}_{\mathbb{T}^c} - \boldsymbol{\Lambda}_{\mathbb{T}}^{-\beta}(\mathbf{I} - \Phi_{\mathbb{T}}^+\Phi_{\mathbb{T}})\mathbf{w}_{\mathbb{T}}\|_2^2 = \|\boldsymbol{\Lambda}_{\mathbb{T}}^{-\beta}\Phi_{\mathbb{T}}^+\Phi_{\mathbb{T}^c}\mathbf{w}_{\mathbb{T}^c}\|_2^2 + \|\boldsymbol{\Lambda}_{\mathbb{T}}^{-\beta}(\mathbf{I} - \Phi_{\mathbb{T}}^+\Phi_{\mathbb{T}})\mathbf{w}_{\mathbb{T}}\|_2^2 - \mathcal{T}_1,$$

$$\|\boldsymbol{\Lambda}_{\mathbb{T}}^{-\beta}(\mathbf{I} - \Phi_{\mathbb{T}}^+\Phi_{\mathbb{T}})\mathbf{w}_{\mathbb{T}}\|_2^2 = \|\boldsymbol{\Lambda}_{\mathbb{T}}^{-\beta}\mathbf{w}_{\mathbb{T}}\|_2^2 + \|\boldsymbol{\Lambda}_{\mathbb{T}}^{-\beta}\Phi_{\mathbb{T}}^+\Phi_{\mathbb{T}}\mathbf{w}_{\mathbb{T}}\|_2^2 - \mathcal{T}_2$$

with $\mathcal{T}_1$ and $\mathcal{T}_2$ given respectively as

$$\mathcal{T}_1 := 2\Re\Big(\big(\boldsymbol{\Lambda}_{\mathbb{T}}^{-\beta}\Phi_{\mathbb{T}}^+\Phi_{\mathbb{T}^c}\mathbf{w}_{\mathbb{T}^c}\big)^*\boldsymbol{\Lambda}_{\mathbb{T}}^{-\beta}(\mathbf{I} - \Phi_{\mathbb{T}}^+\Phi_{\mathbb{T}})\mathbf{w}_{\mathbb{T}}\Big),$$

and

$$\mathcal{T}_2 := 2\Re\Big(\mathbf{w}_{\mathbb{T}}^*\boldsymbol{\Lambda}_{\mathbb{T}}^{-\beta}\boldsymbol{\Lambda}_{\mathbb{T}}^{-\beta}\Phi_{\mathbb{T}}^+\Phi_{\mathbb{T}}\mathbf{w}_{\mathbb{T}}\Big).$$

We therefore conclude from these expansions and (29), using the linearity property of the expectation over $\boldsymbol{\theta}$, that

$$
\begin{aligned}
\mathbb{E}_{\boldsymbol{\theta}}[\|\widehat{\boldsymbol{\theta}} - \boldsymbol{\theta}\|_2^2] = {}&\mathbb{E}_{\boldsymbol{\theta}}[\|\boldsymbol{\Lambda}_{\mathbb{T}}^{-\beta}\mathbf{w}_{\mathbb{T}}\|_2^2] + \mathbb{E}_{\boldsymbol{\theta}}[\|\boldsymbol{\Lambda}_{\mathbb{T}^c}^{-\beta}\mathbf{w}_{\mathbb{T}^c}\|_2^2] \\
&+ \mathbb{E}_{\boldsymbol{\theta}}[\|\boldsymbol{\Lambda}_{\mathbb{T}}^{-\beta}\Phi_{\mathbb{T}}^+\Phi_{\mathbb{T}}\mathbf{w}_{\mathbb{T}}\|_2^2] + \mathbb{E}_{\boldsymbol{\theta}}[\|\boldsymbol{\Lambda}_{\mathbb{T}}^{-\beta}\Phi_{\mathbb{T}}^+\Phi_{\mathbb{T}^c}\mathbf{w}_{\mathbb{T}^c}\|_2^2] - \mathbb{E}_{\boldsymbol{\theta}}[\mathcal{T}_1] - \mathbb{E}_{\boldsymbol{\theta}}[\mathcal{T}_2]. \quad (30)
\end{aligned}
$$

We first observe that the first two terms in the error are simply $\mathrm{tr}(\mathbf{K})$; that is,

$$\mathbb{E}_{\boldsymbol{\theta}}[\|\boldsymbol{\Lambda}_{\mathbb{T}}^{-\beta}\mathbf{w}_{\mathbb{T}}\|_2^2] + \mathbb{E}_{\boldsymbol{\theta}}[\|\boldsymbol{\Lambda}_{\mathbb{T}^c}^{-\beta}\mathbf{w}_{\mathbb{T}^c}\|_2^2] = \mathrm{tr}(\mathbf{K}). \tag{31}$$

By the assumption (27), we also have that $\mathbb{E}_{\boldsymbol{\theta}}[\boldsymbol{\theta}_{\mathbb{T}^c}\mathbf{X}\boldsymbol{\theta}_{\mathbb{T}}^*] = \mathbf{0}$ for any matrix $\mathbf{X}$ such that the product is well-defined. Using the relation between $\boldsymbol{\theta}$ and $\mathbf{w}$, we conclude that

$$\mathbb{E}_{\boldsymbol{\theta}}[\mathcal{T}_1] = \mathbf{0}. \tag{32}$$

We then verify, using simple trace tricks, that

$$\mathbb{E}_{\boldsymbol{\theta}}[\mathcal{T}_2] = 2\mathbb{E}_{\boldsymbol{\theta}}\Big[\Re\Big(\mathrm{tr}\big(\mathbf{w}_{\mathbb{T}}^*\boldsymbol{\Lambda}_{\mathbb{T}}^{-\beta}\boldsymbol{\Lambda}_{\mathbb{T}}^{-\beta}\Phi_{\mathbb{T}}^+\Phi_{\mathbb{T}}\mathbf{w}_{\mathbb{T}}\big)\Big)\Big] = 2\Re\Big(\mathrm{tr}\big(\mathbf{K}_{\mathbb{T}}\Phi_{\mathbb{T}}^+\Phi_{\mathbb{T}}\big)\Big) = 2\mathrm{tr}\big(\mathbf{K}_{\mathbb{T}}\Phi_{\mathbb{T}}^+\Phi_{\mathbb{T}}\big), \tag{33}$$

where we have also used the fact that $\mathbf{K}_\mathbb{T}\boldsymbol{\Lambda}_\mathbb{T}^{-\beta} = \boldsymbol{\Lambda}_\mathbb{T}^{-\beta}\mathbf{K}_\mathbb{T}$ since both matrices are diagonal, and the last step comes from the fact that $\Phi_\mathbb{T}^+\Phi_\mathbb{T}$ is Hermitian.

Next, we observe, using standard trace tricks and the fact that $\Phi_\mathbb{T}^+\Phi_\mathbb{T}$ is Hermitian, that

$$
\begin{aligned}
\mathbb{E}_{\boldsymbol{\theta}}[\|\boldsymbol{\Lambda}_\mathbb{T}^{-\beta}\Phi_\mathbb{T}^+\Phi_\mathbb{T}\mathbf{w}_\mathbb{T}\|_2^2] &= \mathbb{E}_{\boldsymbol{\theta}}[\operatorname{tr}(\mathbf{w}_\mathbb{T}^*\Phi_\mathbb{T}^+\Phi_\mathbb{T}\boldsymbol{\Lambda}_\mathbb{T}^{-2\beta}\Phi_\mathbb{T}^+\Phi_\mathbb{T}\mathbf{w}_\mathbb{T})] \\
&= \operatorname{tr}(\Phi_\mathbb{T}^+\Phi_\mathbb{T}\boldsymbol{\Lambda}_\mathbb{T}^{-2\beta}\Phi_\mathbb{T}^+\Phi_\mathbb{T}\mathbb{E}_{\boldsymbol{\theta}}[\mathbf{w}_\mathbb{T}\mathbf{w}_\mathbb{T}^*]) \\
&= \operatorname{tr}(\Phi_\mathbb{T}^+\Phi_\mathbb{T}\boldsymbol{\Lambda}_\mathbb{T}^{-2\beta}\Phi_\mathbb{T}^+\Phi_\mathbb{T}\boldsymbol{\Lambda}_\mathbb{T}^{\beta}\mathbf{K}_\mathbb{T}\boldsymbol{\Lambda}_\mathbb{T}^{\beta}),
\end{aligned} \tag{34}
$$

and

$$
\begin{aligned}
\mathbb{E}_{\boldsymbol{\theta}}[\|\boldsymbol{\Lambda}_\mathbb{T}^{-\beta}\Phi_\mathbb{T}^+\Phi_{\mathbb{T}^c}\mathbf{w}_{\mathbb{T}^c}\|_2^2] &= \mathbb{E}_{\boldsymbol{\theta}}[\operatorname{tr}(\mathbf{w}_{\mathbb{T}^c}^*\Phi_{\mathbb{T}^c}^*(\Phi_\mathbb{T}^+)^*\boldsymbol{\Lambda}_\mathbb{T}^{-2\beta}\Phi_\mathbb{T}^+\Phi_{\mathbb{T}^c}\mathbf{w}_{\mathbb{T}^c})] \\
&= \operatorname{tr}(\Phi_{\mathbb{T}^c}^*(\Phi_\mathbb{T}^+)^*\boldsymbol{\Lambda}_\mathbb{T}^{-2\beta}\Phi_\mathbb{T}^+\Phi_{\mathbb{T}^c}\mathbb{E}_{\boldsymbol{\theta}}[\mathbf{w}_{\mathbb{T}^c}\mathbf{w}_{\mathbb{T}^c}^*]) \\
&= \operatorname{tr}(\Phi_{\mathbb{T}^c}^*(\Phi_\mathbb{T}^+)^*\boldsymbol{\Lambda}_\mathbb{T}^{-2\beta}\Phi_\mathbb{T}^+\Phi_{\mathbb{T}^c}\boldsymbol{\Lambda}_{\mathbb{T}^c}^{\beta}\mathbf{K}_{\mathbb{T}^c}\boldsymbol{\Lambda}_{\mathbb{T}^c}^{\beta}) \\
&= \operatorname{tr}((\Phi_\mathbb{T}^+)^*\boldsymbol{\Lambda}_\mathbb{T}^{-2\beta}\Phi_\mathbb{T}^+\Phi_{\mathbb{T}^c}\boldsymbol{\Lambda}_{\mathbb{T}^c}^{\beta}\mathbf{K}_{\mathbb{T}^c}\boldsymbol{\Lambda}_{\mathbb{T}^c}^{\beta}\Phi_{\mathbb{T}^c}^*).
\end{aligned} \tag{35}
$$

The proof is complete when we put (31), (32), (33), (34) and (35) into (30). □

Next we derive the general formula for the generalization error for training with noisy data.

**Lemma A.2** (General form of $\mathcal{E}_{\alpha,\beta}^\delta(P,p,N)$). *Let $\boldsymbol{\theta}$ be given as in Lemma A.1. Then, under the assumptions (**A-I**)-(**A-II**), the generalization error for training with weighted optimization (7) from noisy data $\mathbf{y}^\delta$ can be written as*

$$
\mathcal{E}_{\alpha,\beta}^\delta(P,p,N) = \mathcal{E}_{\alpha,\beta}^0(P,p,N) + \mathcal{E}_{\text{noise}}(P,p,N) \tag{36}
$$

*where $\mathcal{E}_{\alpha,\beta}^0(P,p,N)$ is given as in (28) and $\mathcal{E}_{\text{noise}}(P,p,N)$ is given as*

$$
\mathcal{E}_{\text{noise}}(P,p,N) = \operatorname{tr}(\boldsymbol{\Lambda}_{[N]}^{-\alpha}(\Phi_\mathbb{T}^+)^*\boldsymbol{\Lambda}_\mathbb{T}^{-2\beta}\Phi_\mathbb{T}^+\boldsymbol{\Lambda}_{[N]}^{-\alpha}\mathbb{E}_{\boldsymbol{\delta}}[\boldsymbol{\delta}\boldsymbol{\delta}^*]).
$$

*The variance of the generalization error with respect to the random noise is*

$$
\operatorname{Var}_\delta\left(\mathbb{E}_{\boldsymbol{\theta}}[\|\widehat{\boldsymbol{\theta}}^\delta - \boldsymbol{\theta}\|_2^2]\right) = \mathbb{E}_{\boldsymbol{\delta}}\left[\operatorname{tr}(\boldsymbol{\delta}^*\boldsymbol{\Lambda}_{[N]}^{-\alpha}(\Phi_\mathbb{T}^+)^*\boldsymbol{\Lambda}_\mathbb{T}^{-2\beta}\Phi_\mathbb{T}^+\boldsymbol{\Lambda}_{[N]}^{-\alpha}\boldsymbol{\delta}\boldsymbol{\delta}^*\boldsymbol{\Lambda}_{[N]}^{-\alpha}(\Phi_\mathbb{T}^+)^*\boldsymbol{\Lambda}_\mathbb{T}^{-2\beta}\Phi_\mathbb{T}^+\boldsymbol{\Lambda}_{[N]}^{-\alpha}\boldsymbol{\delta})\right]
$$
$$
- \left(\mathcal{E}_{\text{noise}}(P,p,N)\right)^2. \tag{37}
$$

*Proof.* We start with the following standard error decomposition

$$
\|\widehat{\boldsymbol{\theta}}^\delta - \boldsymbol{\theta}\|_2^2 = \|\widehat{\boldsymbol{\theta}}^\delta - \widehat{\boldsymbol{\theta}} + \widehat{\boldsymbol{\theta}} - \boldsymbol{\theta}\|_2^2 = \|\widehat{\boldsymbol{\theta}} - \boldsymbol{\theta}\|_2^2 + \|\widehat{\boldsymbol{\theta}}^\delta - \widehat{\boldsymbol{\theta}}\|_2^2 + 2\Re\left((\widehat{\boldsymbol{\theta}}^\delta - \widehat{\boldsymbol{\theta}})^*(\widehat{\boldsymbol{\theta}} - \boldsymbol{\theta})\right).
$$

Using the fact that

$$
\widehat{\boldsymbol{\theta}}_{\mathbb{T}^c}^\delta = \widehat{\boldsymbol{\theta}}_{\mathbb{T}^c} = \mathbf{0},
$$

the error can be simplified to

$$
\|\widehat{\boldsymbol{\theta}}^\delta - \boldsymbol{\theta}\|_2^2 = \|\widehat{\boldsymbol{\theta}} - \boldsymbol{\theta}\|_2^2 + \|\widehat{\boldsymbol{\theta}}_\mathbb{T}^\delta - \widehat{\boldsymbol{\theta}}_\mathbb{T}\|_2^2 + \mathcal{T}_3, \tag{38}
$$

where

$$
\mathcal{T}_3 = 2\Re\left((\widehat{\boldsymbol{\theta}}_\mathbb{T}^\delta - \widehat{\boldsymbol{\theta}}_\mathbb{T})^*(\widehat{\boldsymbol{\theta}}_\mathbb{T} - \boldsymbol{\theta}_\mathbb{T})\right).
$$

Taking expectation with respect to $\boldsymbol{\theta}$ and then the noise $\boldsymbol{\delta}$, we have that

$$
\mathbb{E}_{\boldsymbol{\delta},\boldsymbol{\theta}}[\|\widehat{\boldsymbol{\theta}}^\delta - \boldsymbol{\theta}\|_2^2] = \mathbb{E}_{\boldsymbol{\theta}}[\|\widehat{\boldsymbol{\theta}} - \boldsymbol{\theta}\|_2^2] + \mathbb{E}_{\boldsymbol{\delta}}[\|\widehat{\boldsymbol{\theta}}_\mathbb{T}^\delta - \widehat{\boldsymbol{\theta}}_\mathbb{T}\|_2^2] + \mathbb{E}_{\boldsymbol{\delta},\boldsymbol{\theta}}[\mathcal{T}_3]. \tag{39}
$$

The first term on the right-hand side is simply $\mathcal{E}_{\alpha,\beta}^0(P,p,N)$ and does not depend on $\delta$. To evaluate the second term which does not depend on $\theta$, we use the fact that

$$
\widehat{\boldsymbol{\theta}}_\mathbb{T}^\delta - \widehat{\boldsymbol{\theta}}_\mathbb{T} = \boldsymbol{\Lambda}_\mathbb{T}^{-\beta}\Phi_\mathbb{T}^+(\mathbf{z}^\delta - \mathbf{z}) = \boldsymbol{\Lambda}_\mathbb{T}^{-\beta}\Phi_\mathbb{T}^+\boldsymbol{\Lambda}_{[N]}^{-\alpha}(\mathbf{y}^\delta - \mathbf{y}) = \boldsymbol{\Lambda}_\mathbb{T}^{-\beta}\Phi_\mathbb{T}^+\boldsymbol{\Lambda}_{[N]}^{-\alpha}\boldsymbol{\delta}.
$$

This leads to

$$
\begin{aligned}
\mathbb{E}_{\boldsymbol{\delta}}[\|\widehat{\boldsymbol{\theta}}_{\mathbb{T}}^{\delta} - \widehat{\boldsymbol{\theta}}_{\mathbb{T}}\|_2^2] &= \mathbb{E}_{\boldsymbol{\delta}}[\|\boldsymbol{\Lambda}_{\mathbb{T}}^{-\beta}\Phi_{\mathbb{T}}^{+}\boldsymbol{\Lambda}_{[N]}^{-\alpha}\boldsymbol{\delta}\|_2^2] \\
&= \mathbb{E}_{\boldsymbol{\delta}}[\mathrm{tr}\big(\boldsymbol{\delta}^{*}\boldsymbol{\Lambda}_{[N]}^{-\alpha}(\Phi_{\mathbb{T}}^{+})^{*}\boldsymbol{\Lambda}_{\mathbb{T}}^{-2\beta}\Phi_{\mathbb{T}}^{+}\boldsymbol{\Lambda}_{[N]}^{-\alpha}\boldsymbol{\delta}\big)] \\
&= \mathrm{tr}\big(\boldsymbol{\Lambda}_{[N]}^{-\alpha}(\Phi_{\mathbb{T}}^{+})^{*}\boldsymbol{\Lambda}_{\mathbb{T}}^{-2\beta}\Phi_{\mathbb{T}}^{+}\boldsymbol{\Lambda}_{[N]}^{-\alpha}\mathbb{E}_{\boldsymbol{\delta}}[\boldsymbol{\delta}\boldsymbol{\delta}^{*}]\big).
\end{aligned}
$$

The last step is to realize that

$$
\mathbb{E}_{\boldsymbol{\delta},\boldsymbol{\theta}}[\mathcal{T}_3] = 2\Re\Big(\mathbb{E}_{\boldsymbol{\delta},\boldsymbol{\theta}}\Big[\big(\boldsymbol{\Lambda}_{\mathbb{T}}^{-\beta}\Phi_{\mathbb{T}}^{+}\boldsymbol{\Lambda}_{[N]}^{-\alpha}\boldsymbol{\delta}\big)^{*}(\widehat{\boldsymbol{\theta}}_{\mathbb{T}} - \boldsymbol{\theta}_{\mathbb{T}})\Big]\Big) = 0
$$

due to the assumption that $\boldsymbol{\delta}$ and $\boldsymbol{\theta}$ are independent.

Utilizing the formulas in (38) and (39), and the fact that $\mathbb{E}_{\boldsymbol{\theta}}[\mathcal{T}_3] = 0$, we can write down the variance of the generalization error with respect to the random noise as

$$
\mathrm{Var}_{\delta}\Big(\mathbb{E}_{\boldsymbol{\theta}}[\|\widehat{\boldsymbol{\theta}}^{\delta} - \boldsymbol{\theta}\|_2^2]\Big) = \mathbb{E}_{\boldsymbol{\delta}}[\|\widehat{\boldsymbol{\theta}}_{\mathbb{T}}^{\delta} - \widehat{\boldsymbol{\theta}}_{\mathbb{T}}\|_2^4] - \Big(\mathbb{E}_{\boldsymbol{\delta}}[\|\widehat{\boldsymbol{\theta}}_{\mathbb{T}}^{\delta} - \widehat{\boldsymbol{\theta}}_{\mathbb{T}}\|_2^2]\Big)^{2}
$$

$$
= \mathbb{E}_{\boldsymbol{\delta}}[\mathrm{tr}\big(\boldsymbol{\delta}^{*}\boldsymbol{\Lambda}_{[N]}^{-\alpha}(\Phi_{\mathbb{T}}^{+})^{*}\boldsymbol{\Lambda}_{\mathbb{T}}^{-2\beta}\Phi_{\mathbb{T}}^{+}\boldsymbol{\Lambda}_{[N]}^{-\alpha}\boldsymbol{\delta}\boldsymbol{\delta}^{*}\boldsymbol{\Lambda}_{[N]}^{-\alpha}(\Phi_{\mathbb{T}}^{+})^{*}\boldsymbol{\Lambda}_{\mathbb{T}}^{-2\beta}\Phi_{\mathbb{T}}^{+}\boldsymbol{\Lambda}_{[N]}^{-\alpha}\boldsymbol{\delta}\big)] - \big(\mathcal{E}_{\mathrm{noise}}(P, p, N)\big)^{2}.
$$

The proof is complete. $\qquad\square$

We also need the following properties on the the feature matrix $\Psi$ of the random Fourier model.

**Lemma A.3.** *(i) For any $\zeta \geq 0$, we define, in the overparameterized regime (12), the matrices $\boldsymbol{\Pi}_1 \in \mathbb{C}^{N \times N}$ and $\boldsymbol{\Pi}_2 \in \mathbb{C}^{N \times N}$ respectively as*

$$
\boldsymbol{\Pi}_1 := \Psi_{\mathbb{T}}\boldsymbol{\Lambda}_{\mathbb{T}}^{-\zeta}\Psi_{\mathbb{T}}^{*} \quad and \quad \boldsymbol{\Pi}_2 = \Psi_{\mathbb{T}^c}\boldsymbol{\Lambda}_{\mathbb{T}^c}^{-\zeta}\Psi_{\mathbb{T}^c}^{*}.
$$

*Then $\boldsymbol{\Pi}_1$ and $\boldsymbol{\Pi}_2$ admit the decomposition $\boldsymbol{\Pi}_1 = \Psi_{[N]}\boldsymbol{\Lambda}_{\Pi_1,\zeta}\Psi_{[N]}^{*}$ and $\boldsymbol{\Pi}_2 = \Psi_{[N]}\boldsymbol{\Lambda}_{\Pi_2,\zeta}\Psi_{[N]}^{*}$ respectively with $\boldsymbol{\Lambda}_{\Pi_1,\zeta}$ and $\boldsymbol{\Lambda}_{\Pi_2,\zeta}$ diagonal matrices whose $m$-th diagonal elements are respectively*

$$
\lambda_{\Pi_1,\zeta}^{(m)} = \sum_{k=0}^{N-1}\Big(\sum_{\eta=0}^{\nu-1}t_{k+N\eta}^{-\zeta}\Big)e_{m,k}^{(N)}, \quad 0 \leq m \leq N-1,
$$

*and*

$$
\lambda_{\Pi_2,\zeta}^{(m)} = \sum_{k=0}^{N-1}\Big(\sum_{\eta=\nu}^{\mu-1}t_{k+N\eta}^{-\zeta}\Big)e_{m,k}^{(N)}, \quad 0 \leq m \leq N-1,
$$

*where $e_{m,k}^{(N)}$ is defined as, denoting $\omega_N := e^{\frac{2\pi i}{N}}$,*

$$
e_{m,k}^{(N)} := \frac{1}{N}\sum_{j=0}^{N-1}\omega_N^{(m-k)j} = \begin{cases} 1, & k = m \\ 0, & k \neq m \end{cases}, \qquad 0 \leq m, k \leq N-1.
$$

*(ii) For any $\alpha \geq 0$, we define, in the underparameterized regime (13) with $p < N$, the matrix $\boldsymbol{\Xi} \in \mathbb{C}^{p \times (N-p)}$ as*

$$
\boldsymbol{\Xi} := \big(\Psi_{\mathbb{T}}^{*}\boldsymbol{\Lambda}_{[N]}^{-2\alpha}\Psi_{\mathbb{T}}\big)^{-1}\Psi_{\mathbb{T}}^{*}\boldsymbol{\Lambda}_{[N]}^{-2\alpha}\Psi_{[N]\setminus\mathbb{T}}.
$$

*Then $\boldsymbol{\Xi}^{*}\boldsymbol{\Xi}$ has the following representation:*

$$
\boldsymbol{\Xi}^{*}\boldsymbol{\Xi} = N(\Psi_{[N]\setminus\mathbb{T}}^{*}\boldsymbol{\Lambda}_{[N]}^{2\alpha}\Psi_{[N]\setminus\mathbb{T}})^{-*}\Psi_{[N]\setminus\mathbb{T}}^{*}\boldsymbol{\Lambda}^{4\alpha}\Psi_{[N]\setminus\mathbb{T}}(\Psi_{[N]\setminus\mathbb{T}}^{*}\boldsymbol{\Lambda}_{[N]}^{2\alpha}\Psi_{[N]\setminus\mathbb{T}})^{-1} - \mathbf{I}_{[N-p]}.
$$

*Proof.* Part (i) is simply Lemma 2 of Xie et al. (2020). It was proved by verifying that $\boldsymbol{\Pi}_1$ and $\boldsymbol{\Pi}_2$ are both circulant matrices whose eigendecompositions are standard results in linear algebra.

To prove part (ii), let us introduce the matrices $\mathbf{P}$, $\mathbf{Q}$, $\mathbf{R}$ through the relation

$$
\Psi_{[N]}^{*}\boldsymbol{\Lambda}_{[N]}^{-2\alpha}\Psi_{[N]} = \begin{pmatrix} \Psi_{\mathbb{T}}^{*}\boldsymbol{\Lambda}_{[N]}^{-2\alpha}\Psi_{\mathbb{T}} & \Psi_{\mathbb{T}}^{*}\boldsymbol{\Lambda}_{[N]}^{-2\alpha}\Psi_{[N]\setminus\mathbb{T}} \\ \Psi_{[N]\setminus\mathbb{T}}^{*}\boldsymbol{\Lambda}_{[N]}^{-2\alpha}\Psi_{\mathbb{T}} & \Psi_{[N]\setminus\mathbb{T}}^{*}\boldsymbol{\Lambda}_{[N]}^{-2\alpha}\Psi_{[N]\setminus\mathbb{T}} \end{pmatrix} \equiv \begin{pmatrix} \mathbf{P} & \mathbf{R} \\ \mathbf{R}^{*} & \mathbf{Q} \end{pmatrix}.
$$

Then by standard linear algebra, we have that

$$\left(\Psi_{[N]}^{*}\mathbf{\Lambda}_{[N]}^{-2\alpha}\Psi_{[N]}\right)^{-1} = \begin{pmatrix} \mathbf{P}^{-1} + \mathbf{P}^{-1}\mathbf{R}(\mathbf{Q} - \mathbf{R}^{*}\mathbf{P}^{-1}\mathbf{R})^{-1}\mathbf{R}^{*}\mathbf{P}^{-1} & -\mathbf{P}^{-1}\mathbf{R}(\mathbf{Q} - \mathbf{R}^{*}\mathbf{P}^{-1}\mathbf{R})^{-1} \\ -(\mathbf{Q} - \mathbf{R}^{*}\mathbf{P}^{-1}\mathbf{R})^{-1}\mathbf{R}^{*}\mathbf{P}^{-1} & (\mathbf{Q} - \mathbf{R}^{*}\mathbf{P}^{-1}\mathbf{R})^{-1} \end{pmatrix}$$

since $\mathbf{P} = \Psi_{\mathbb{T}}^{*}\mathbf{\Lambda}_{[N]}^{-2\alpha}\Psi_{\mathbb{T}}$ is invertible.

Meanwhile, since $\frac{1}{\sqrt{N}}\Psi_{[N]}$ is unitary, we have that

$$
\begin{aligned}
\left(\Psi_{[N]}^{*}\mathbf{\Lambda}_{[N]}^{-2\alpha}\Psi_{[N]}\right)^{-1} &= \frac{1}{N^2}\Psi_{[N]}^{*}\mathbf{\Lambda}_{[N]}^{2\alpha}\Psi_{[N]} \\
&= \frac{1}{N^2}\begin{pmatrix} \Psi_{\mathbb{T}}^{*}\mathbf{\Lambda}_{[N]}^{2\alpha}\Psi_{\mathbb{T}} & \Psi_{\mathbb{T}}^{*}\mathbf{\Lambda}_{[N]}^{2\alpha}\Psi_{[N]\backslash\mathbb{T}} \\ \Psi_{[N]\backslash\mathbb{T}}^{*}\mathbf{\Lambda}_{[N]}^{2\alpha}\Psi_{\mathbb{T}} & \Psi_{[N]\backslash\mathbb{T}}^{*}\mathbf{\Lambda}_{[N]}^{2\alpha}\Psi_{[N]\backslash\mathbb{T}} \end{pmatrix} \equiv \begin{pmatrix} \widetilde{\mathbf{P}} & \widetilde{\mathbf{R}} \\ \widetilde{\mathbf{R}}^{*} & \widetilde{\mathbf{Q}} \end{pmatrix}.
\end{aligned}
$$

Comparing the two inverse matrices lead us to the following identity

$$\mathbf{P}^{-1} - \mathbf{P}^{-1}\mathbf{R}\widetilde{\mathbf{R}}^{*} = \widetilde{\mathbf{P}}.$$

This gives us that

$$\mathbf{P}^{-1}\mathbf{R} = \widetilde{\mathbf{P}}\mathbf{R}(\mathbf{I}_{[N-p]} - \widetilde{\mathbf{R}}^{*}\mathbf{R})^{-1}.$$

Therefore, we have

$$\mathbf{R}^{*}\mathbf{P}^{-*}\mathbf{P}^{-1}\mathbf{R} = (\mathbf{I}_{[N-p]} - \widetilde{\mathbf{R}}^{*}\mathbf{R})^{-*}\mathbf{R}^{*}\widetilde{\mathbf{P}}^{*}\widetilde{\mathbf{P}}\mathbf{R}(\mathbf{I}_{[N-p]} - \widetilde{\mathbf{R}}^{*}\mathbf{R})^{-1}.$$

Utilizing the fact that

$$
\begin{aligned}
\widetilde{\mathbf{R}}^{*}\mathbf{R} &= \frac{1}{N^2}\Psi_{[N]\backslash\mathbb{T}}^{*}\mathbf{\Lambda}_{[N]}^{2\alpha}\Psi_{\mathbb{T}}\Psi_{\mathbb{T}}^{*}\mathbf{\Lambda}_{[N]}^{-2\alpha}\Psi_{[N]\backslash\mathbb{T}} \\
&= \frac{1}{N^2}\Psi_{[N]\backslash\mathbb{T}}^{*}\mathbf{\Lambda}_{[N]}^{2\alpha}\left(N\mathbf{I}_{[N]} - \Psi_{[N]\backslash\mathbb{T}}\Psi_{[N]\backslash\mathbb{T}}^{*}\right)\mathbf{\Lambda}_{[N]}^{-2\alpha}\Psi_{[N]\backslash\mathbb{T}} = \mathbf{I}_{[N-p]} - \widetilde{\mathbf{Q}}\mathbf{Q},
\end{aligned}
$$

we can now have the following identity

$$\mathbf{R}^{*}\mathbf{P}^{-*}\mathbf{P}^{-1}\mathbf{R} = (\widetilde{\mathbf{Q}}\mathbf{Q})^{-*}\mathbf{R}^{*}\widetilde{\mathbf{P}}^{*}\widetilde{\mathbf{P}}\mathbf{R}(\widetilde{\mathbf{Q}}\mathbf{Q})^{-1}.$$

Using the formulas for $\widetilde{\mathbf{P}}$ and $\mathbf{R}$, as well as $\Psi_{\mathbb{T}}\Psi_{\mathbb{T}}^{*} = N\mathbf{I}_{[N]} - \Psi_{[N]\backslash\mathbb{T}}\Psi_{[N]\backslash\mathbb{T}}^{*}$, we have

$$
\begin{aligned}
\mathbf{R}^{*}\widetilde{\mathbf{P}}^{*}\widetilde{\mathbf{P}}\mathbf{R} &= \frac{1}{N^4}\Psi_{[N]\backslash\mathbb{T}}^{*}\mathbf{\Lambda}_{[N]}^{-2\alpha}\Psi_{\mathbb{T}}\Psi_{\mathbb{T}}^{*}\mathbf{\Lambda}_{[N]}^{2\alpha}\Psi_{\mathbb{T}}\Psi_{\mathbb{T}}^{*}\mathbf{\Lambda}_{[N]}^{2\alpha}\Psi_{\mathbb{T}}\Psi_{\mathbb{T}}^{*}\mathbf{\Lambda}_{[N]}^{-2\alpha}\Psi_{[N]\backslash\mathbb{T}} \\
&= \frac{1}{N^3}\mathbf{Q}\Psi_{[N]\backslash\mathbb{T}}^{*}\mathbf{\Lambda}_{[N]}^{4\alpha}\Psi_{[N]\backslash\mathbb{T}}\mathbf{Q} - \mathbf{Q}\widetilde{\mathbf{Q}}\widetilde{\mathbf{Q}}\mathbf{Q}.
\end{aligned}
$$

This finally gives us

$$\mathbf{R}^{*}\mathbf{P}^{-*}\mathbf{P}^{-1}\mathbf{R} = \frac{1}{N^3}\widetilde{\mathbf{Q}}^{-*}\Psi_{[N]\backslash\mathbb{T}}^{*}\mathbf{\Lambda}_{[N]}^{4\alpha}\Psi_{[N]\backslash\mathbb{T}}\widetilde{\mathbf{Q}}^{-1} - \mathbf{I}_{[N-p]}.$$

The proof is complete when we insert the definion of $\widetilde{\mathbf{Q}}$ into the result. $\qquad\square$

## A.2 PROOF OF THEOREM 3.1

We provide the proof of Theorem 3.1 here. We split the proof into the overparameterized and the underparameterized regimes.

*Proof of Theorem 3.1 (Overparameterized Regime).* In the overparameterized regime (12), we have that the Moore–Penrose inverse $\Phi_{\mathbb{T}}^{+} = \Phi_{\mathbb{T}}^{*}(\Phi_{\mathbb{T}}\Phi_{\mathbb{T}}^{*})^{-1}$. Using the definitions of $\Phi_{\mathbb{T}}$ in (26), we can verify that

$$
\begin{aligned}
\Phi_{\mathbb{T}}^{+}\Phi_{\mathbb{T}}\mathbf{\Lambda}_{\mathbb{T}}^{-\beta} &= \mathbf{\Lambda}_{\mathbb{T}}^{-\beta}\Psi_{\mathbb{T}}^{*}(\Psi_{\mathbb{T}}\mathbf{\Lambda}_{\mathbb{T}}^{-2\beta}\Psi_{\mathbb{T}}^{*})^{-1}\Psi_{\mathbb{T}}\mathbf{\Lambda}_{\mathbb{T}}^{-2\beta}, \\
\mathbf{\Lambda}_{\mathbb{T}}^{-\beta}\Phi_{\mathbb{T}}^{+}\Phi_{\mathbb{T}} &= \mathbf{\Lambda}_{\mathbb{T}}^{-2\beta}\Psi_{\mathbb{T}}^{*}(\Psi_{\mathbb{T}}\mathbf{\Lambda}_{\mathbb{T}}^{-2\beta}\Psi_{\mathbb{T}}^{*})^{-1}\Psi_{\mathbb{T}}\mathbf{\Lambda}_{\mathbb{T}}^{-\beta},
\end{aligned}
$$

and

$$\Phi_{\mathbb{T}}^{+}\Phi_{\mathbb{T}}\mathbf{\Lambda}_{\mathbb{T}}^{-2\beta}\Phi_{\mathbb{T}}^{+}\Phi_{\mathbb{T}} = \mathbf{\Lambda}_{\mathbb{T}}^{-\beta}\Psi_{\mathbb{T}}^{*}(\Psi_{\mathbb{T}}\mathbf{\Lambda}_{\mathbb{T}}^{-2\beta}\Psi_{\mathbb{T}}^{*})^{-1}\Psi_{\mathbb{T}}\mathbf{\Lambda}_{\mathbb{T}}^{-4\beta}\Psi_{\mathbb{T}}^{*}(\Psi_{\mathbb{T}}\mathbf{\Lambda}_{\mathbb{T}}^{-2\beta}\Psi_{\mathbb{T}}^{*})^{-1}\Psi_{\mathbb{T}}\mathbf{\Lambda}_{\mathbb{T}}^{-\beta}.$$

We therefore have

$$\text{tr}\big(\boldsymbol{\Phi}_{\mathbb{T}}^{+}\boldsymbol{\Phi}_{\mathbb{T}}\boldsymbol{\Lambda}_{\mathbb{T}}^{-2\beta}\boldsymbol{\Phi}_{\mathbb{T}}^{+}\boldsymbol{\Phi}_{\mathbb{T}}\boldsymbol{\Lambda}_{\mathbb{T}}^{\beta}\mathbf{K}_{\mathbb{T}}\boldsymbol{\Lambda}_{\mathbb{T}}^{\beta}\big)$$
$$= \text{tr}\big((\boldsymbol{\Psi}_{\mathbb{T}}\boldsymbol{\Lambda}_{\mathbb{T}}^{-2\beta}\boldsymbol{\Psi}_{\mathbb{T}}^{*})^{-1}\boldsymbol{\Psi}_{\mathbb{T}}\boldsymbol{\Lambda}_{\mathbb{T}}^{-4\beta}\boldsymbol{\Psi}_{\mathbb{T}}^{*}(\boldsymbol{\Psi}_{\mathbb{T}}\boldsymbol{\Lambda}_{\mathbb{T}}^{-2\beta}\boldsymbol{\Psi}_{\mathbb{T}}^{*})^{-1}\boldsymbol{\Psi}_{\mathbb{T}}\mathbf{K}_{\mathbb{T}}\boldsymbol{\Psi}_{\mathbb{T}}^{*}\big)\,.$$

Meanwhile, a trace trick leads to

$$\text{tr}\big(\mathbf{K}_{\mathbb{T}}\boldsymbol{\Phi}_{\mathbb{T}}^{+}\boldsymbol{\Phi}_{\mathbb{T}}\big) = \text{tr}\big(\mathbf{K}_{\mathbb{T}}\boldsymbol{\Phi}_{\mathbb{T}}^{*}(\boldsymbol{\Phi}_{\mathbb{T}}\boldsymbol{\Phi}_{\mathbb{T}}^{*})^{-1}\boldsymbol{\Phi}_{\mathbb{T}}\big) = \text{tr}\big((\boldsymbol{\Phi}_{\mathbb{T}}\boldsymbol{\Phi}_{\mathbb{T}}^{*})^{-1}\boldsymbol{\Phi}_{\mathbb{T}}\mathbf{K}_{\mathbb{T}}\boldsymbol{\Phi}_{\mathbb{T}}^{*}\big)$$
$$= \text{tr}\big((\boldsymbol{\Lambda}_{[N]}^{-\alpha}\boldsymbol{\Psi}_{\mathbb{T}}\boldsymbol{\Lambda}_{\mathbb{T}}^{-2\beta}\boldsymbol{\Psi}_{\mathbb{T}}^{*}\boldsymbol{\Lambda}_{[N]}^{-\alpha})^{-1}\boldsymbol{\Lambda}_{[N]}^{-\alpha}\boldsymbol{\Psi}_{\mathbb{T}}\boldsymbol{\Lambda}_{\mathbb{T}}^{-\beta}\mathbf{K}_{\mathbb{T}}\boldsymbol{\Lambda}_{\mathbb{T}}^{-\beta}\boldsymbol{\Psi}_{\mathbb{T}}^{*}\boldsymbol{\Lambda}_{[N]}^{-\alpha}\big)$$
$$= \text{tr}\big((\boldsymbol{\Psi}_{\mathbb{T}}\boldsymbol{\Lambda}_{\mathbb{T}}^{-2\beta}\boldsymbol{\Psi}_{\mathbb{T}}^{*})^{-1}\boldsymbol{\Psi}_{\mathbb{T}}\boldsymbol{\Lambda}_{\mathbb{T}}^{-\beta}\mathbf{K}_{\mathbb{T}}\boldsymbol{\Lambda}_{\mathbb{T}}^{-\beta}\boldsymbol{\Psi}_{\mathbb{T}}^{*}\big)\,. \tag{40}$$

Therefore, based on Lemma A.1, we have finally that

$$\mathcal{P}_{\alpha,\beta} = \text{tr}\big((\boldsymbol{\Psi}_{\mathbb{T}}\boldsymbol{\Lambda}_{\mathbb{T}}^{-2\beta}\boldsymbol{\Psi}_{\mathbb{T}}^{*})^{-1}\boldsymbol{\Psi}_{\mathbb{T}}\boldsymbol{\Lambda}_{\mathbb{T}}^{-4\beta}\boldsymbol{\Psi}_{\mathbb{T}}^{*}(\boldsymbol{\Psi}_{\mathbb{T}}\boldsymbol{\Lambda}_{\mathbb{T}}^{-2\beta}\boldsymbol{\Psi}_{\mathbb{T}}^{*})^{-1}\boldsymbol{\Psi}_{\mathbb{T}}\mathbf{K}_{\mathbb{T}}\boldsymbol{\Psi}_{\mathbb{T}}^{*}\big)$$
$$- 2\text{tr}\big((\boldsymbol{\Psi}_{\mathbb{T}}\boldsymbol{\Lambda}_{\mathbb{T}}^{-2\beta}\boldsymbol{\Psi}_{\mathbb{T}}^{*})^{-1}\boldsymbol{\Psi}_{\mathbb{T}}\boldsymbol{\Lambda}_{\mathbb{T}}^{-\beta}\mathbf{K}_{\mathbb{T}}\boldsymbol{\Lambda}_{\mathbb{T}}^{-\beta}\boldsymbol{\Psi}_{\mathbb{T}}^{*}\big)\,. \tag{41}$$

In a similar manner, we can check that

$$\mathcal{Q}_{\alpha,\beta} = \text{tr}\big((\boldsymbol{\Phi}_{\mathbb{T}}^{+})^{*}\boldsymbol{\Lambda}_{\mathbb{T}}^{-2\beta}\boldsymbol{\Phi}_{\mathbb{T}}^{+}\boldsymbol{\Phi}_{\mathbb{T}^c}\boldsymbol{\Lambda}_{\mathbb{T}^c}^{\beta}\mathbf{K}_{\mathbb{T}^c}\boldsymbol{\Lambda}_{\mathbb{T}^c}^{\beta}\boldsymbol{\Phi}_{\mathbb{T}^c}^{*}\big)$$
$$= \text{tr}\big((\boldsymbol{\Phi}_{\mathbb{T}}\boldsymbol{\Phi}_{\mathbb{T}}^{*})^{-*}\boldsymbol{\Phi}_{\mathbb{T}}\boldsymbol{\Lambda}_{\mathbb{T}}^{-2\beta}\boldsymbol{\Phi}_{\mathbb{T}}^{*}(\boldsymbol{\Phi}_{\mathbb{T}}\boldsymbol{\Phi}_{\mathbb{T}}^{*})^{-1}\boldsymbol{\Phi}_{\mathbb{T}^c}\boldsymbol{\Lambda}_{\mathbb{T}^c}^{\beta}\mathbf{K}_{\mathbb{T}^c}\boldsymbol{\Lambda}_{\mathbb{T}^c}^{\beta}\boldsymbol{\Phi}_{\mathbb{T}^c}^{*}\big)$$
$$= \text{tr}\big((\boldsymbol{\Psi}_{\mathbb{T}}\boldsymbol{\Lambda}_{\mathbb{T}}^{-2\beta}\boldsymbol{\Psi}_{\mathbb{T}}^{*})^{-*}\boldsymbol{\Psi}_{\mathbb{T}}\boldsymbol{\Lambda}_{\mathbb{T}}^{-4\beta}\boldsymbol{\Psi}_{\mathbb{T}}^{*}(\boldsymbol{\Psi}_{\mathbb{T}}\boldsymbol{\Lambda}_{\mathbb{T}}^{-2\beta}\boldsymbol{\Psi}_{\mathbb{T}}^{*})^{-1}\boldsymbol{\Psi}_{\mathbb{T}^c}\mathbf{K}_{\mathbb{T}^c}\boldsymbol{\Psi}_{\mathbb{T}^c}^{*}\big)\,. \tag{42}$$

The above calculations give us

$$\mathcal{E}_{\alpha,\beta}^{0}(P,p,N) = \text{tr}\big(\mathbf{K}\big) - 2\text{tr}\big((\boldsymbol{\Psi}_{\mathbb{T}}\boldsymbol{\Lambda}_{\mathbb{T}}^{-2\beta}\boldsymbol{\Psi}_{\mathbb{T}}^{*})^{-1}\boldsymbol{\Psi}_{\mathbb{T}}\boldsymbol{\Lambda}_{\mathbb{T}}^{-\beta}\mathbf{K}_{\mathbb{T}}\boldsymbol{\Lambda}_{\mathbb{T}}^{-\beta}\boldsymbol{\Psi}_{\mathbb{T}}^{*}\big)$$
$$+ \text{tr}\big((\boldsymbol{\Psi}_{\mathbb{T}}\boldsymbol{\Lambda}_{\mathbb{T}}^{-2\beta}\boldsymbol{\Psi}_{\mathbb{T}}^{*})^{-*}\boldsymbol{\Psi}_{\mathbb{T}}\boldsymbol{\Lambda}_{\mathbb{T}}^{-4\beta}\boldsymbol{\Psi}_{\mathbb{T}}^{*}(\boldsymbol{\Psi}_{\mathbb{T}}\boldsymbol{\Lambda}_{\mathbb{T}}^{-2\beta}\boldsymbol{\Psi}_{\mathbb{T}}^{*})^{-1}\boldsymbol{\Psi}\mathbf{K}\boldsymbol{\Psi}^{*}\big)\,. \tag{43}$$

We are now ready to evaluate the terms in the generalization error. First, we have that since $\mathbf{K} = c_{\gamma}\boldsymbol{\Lambda}_{[P]}^{-2\gamma}$, we have, using the definition of $c_{\gamma}$, that

$$\text{tr}\big(\mathbf{K}\big) = c_{\gamma}\sum_{j=0}^{P-1}t_{j}^{-2\gamma} = 1\,. \tag{44}$$

Second, using the results in part (i) of Lemma A.3 and the fact that $\frac{1}{\sqrt{N}}\boldsymbol{\Psi}_{[N]}$ is unitary, we have

$$\text{tr}\big((\boldsymbol{\Psi}_{\mathbb{T}}\boldsymbol{\Lambda}_{\mathbb{T}}^{-2\beta}\boldsymbol{\Psi}_{\mathbb{T}}^{*})^{-1}\boldsymbol{\Psi}_{\mathbb{T}}\boldsymbol{\Lambda}_{\mathbb{T}}^{-\beta}\mathbf{K}_{\mathbb{T}}\boldsymbol{\Lambda}_{\mathbb{T}}^{-\beta}\boldsymbol{\Psi}_{\mathbb{T}}^{*}\big)$$
$$= c_{\gamma}\text{tr}\big(\frac{1}{N^2}\boldsymbol{\Psi}_{[N]}\boldsymbol{\Lambda}_{\Pi_1,2\beta}^{-1}\boldsymbol{\Psi}_{[N]}^{*}\boldsymbol{\Psi}_{[N]}\boldsymbol{\Lambda}_{\Pi_1,2\beta+2\gamma}\boldsymbol{\Psi}_{[N]}^{*}\big)$$
$$= c_{\gamma}\text{tr}\big(\frac{1}{N}\boldsymbol{\Psi}_{[N]}\Lambda_{\Pi_1,2\beta}^{-1}\Lambda_{\Pi_1,2\beta+2\gamma}\boldsymbol{\Psi}_{[N]}^{*}\big) = c_{\gamma}\text{tr}\big(\Lambda_{\Pi_1,2\beta}^{-1}\Lambda_{\Pi_1,2\beta+2\gamma}\big)$$
$$= c_{\gamma}\sum_{m=0}^{N-1}\frac{\sum_{k=0}^{N-1}\left(\sum_{\eta=0}^{\nu-1}t_{k+N\eta}^{-2\beta-2\gamma}\right)e_{m,k}^{(N)}}{\sum_{k=0}^{N-1}\left(\sum_{\eta=0}^{\nu-1}t_{k+N\eta}^{-2\beta}\right)e_{m,k}^{(N)}} = c_{\gamma}\sum_{k=0}^{N-1}\frac{\sum_{\eta=0}^{\nu-1}t_{k+N\eta}^{-2\beta-2\gamma}}{\sum_{\eta=0}^{\nu-1}t_{k+N\eta}^{-2\beta}}\,. \tag{45}$$

The results in Lemma A.3 also give that

$$\text{tr}\big((\boldsymbol{\Psi}_{\mathbb{T}}\boldsymbol{\Lambda}_{\mathbb{T}}^{-2\beta}\boldsymbol{\Psi}_{\mathbb{T}}^{*})^{-*}\boldsymbol{\Psi}_{\mathbb{T}}\boldsymbol{\Lambda}_{\mathbb{T}}^{-4\beta}\boldsymbol{\Psi}_{\mathbb{T}}^{*}(\boldsymbol{\Psi}_{\mathbb{T}}\boldsymbol{\Lambda}_{\mathbb{T}}^{-2\beta}\boldsymbol{\Psi}_{\mathbb{T}}^{*})^{-1}\boldsymbol{\Psi}\mathbf{K}\boldsymbol{\Psi}^{*}\big)$$
$$= \text{tr}\big(\frac{1}{N^2}\boldsymbol{\Psi}_{[N]}\boldsymbol{\Lambda}_{\Pi_1,2\beta}^{-1}\boldsymbol{\Psi}_{[N]}^{*}\boldsymbol{\Psi}_{[N]}\boldsymbol{\Lambda}_{\Pi_1,4\beta}\boldsymbol{\Psi}_{[N]}^{*}\frac{1}{N^2}\boldsymbol{\Psi}_{[N]}\boldsymbol{\Lambda}_{\Pi_1,2\beta}^{-1}\boldsymbol{\Psi}_{[N]}^{*}\boldsymbol{\Psi}_{[N]}\big(\boldsymbol{\Lambda}_{\Pi_1,2\gamma}+\boldsymbol{\Lambda}_{\Pi_2,2\gamma}\big)\boldsymbol{\Psi}_{[N]}^{*}\big)$$
$$= \text{tr}\big(\boldsymbol{\Lambda}_{\Pi_1,2\beta}^{-1}\boldsymbol{\Lambda}_{\Pi_1,4\beta}\boldsymbol{\Lambda}_{\Pi_1,2\beta}^{-1}\big(\boldsymbol{\Lambda}_{\Pi_1,2\gamma}+\boldsymbol{\Lambda}_{\Pi_2,2\gamma}\big)\big)\,.$$

We can plug in the formula of A.3 to get

$$\text{tr}\big((\boldsymbol{\Psi}_{\mathbb{T}}\boldsymbol{\Lambda}_{\mathbb{T}}^{-2\beta}\boldsymbol{\Psi}_{\mathbb{T}}^{*})^{-*}\boldsymbol{\Psi}_{\mathbb{T}}\boldsymbol{\Lambda}_{\mathbb{T}}^{-4\beta}\boldsymbol{\Psi}_{\mathbb{T}}^{*}(\boldsymbol{\Psi}_{\mathbb{T}}\boldsymbol{\Lambda}_{\mathbb{T}}^{-2\beta}\boldsymbol{\Psi}_{\mathbb{T}}^{*})^{-1}\boldsymbol{\Psi}\mathbf{K}\boldsymbol{\Psi}^{*}\big)$$
$$= \sum_{m=0}^{N-1}\frac{\left[\sum_{k=0}^{N-1}\left(\sum_{\eta=0}^{\nu-1}t_{k+N\eta}^{-4\beta}\right)e_{m,k}^{(N)}\right]\left[\sum_{k=0}^{N-1}\left(\sum_{\eta=0}^{\mu-1}t_{k+N\eta}^{-2\gamma}\right)e_{m,k}^{(N)}\right]}{\left[\sum_{k=0}^{N-1}\left(\sum_{\eta=0}^{\nu-1}t_{k+N\eta}^{-2\beta}\right)e_{m,k}^{(N)}\right]^2}$$
$$= \sum_{k=0}^{N-1}\frac{\left(\sum_{\eta=0}^{\nu-1}t_{k+N\eta}^{-4\beta}\right)\left(\sum_{\eta=0}^{\mu-1}t_{k+N\eta}^{-2\gamma}\right)}{\left[\sum_{\eta=0}^{\nu-1}t_{k+N\eta}^{-2\beta}\right]^2}\,. \tag{46}$$

We can now put (44), (45) and (46) together into the general error formula (43) to get the result (14) in Theorem 3.1. The proof is complete. $\qquad\square$

*Proof of Theorem 3.1 (Underparameterized Regime).* In the underparameterized regime (13), we have that the Moore–Penrose inverse $\Phi_{\mathbb{T}}^{+} = (\Phi_{\mathbb{T}}^{*}\Phi_{\mathbb{T}})^{-1}\Phi_{\mathbb{T}}^{*}$. This leads to

$$\Phi_{\mathbb{T}}^{+}\Phi_{\mathbb{T}} = \mathbf{I}_{\mathbb{T}} \quad \text{and} \quad (\Phi_{\mathbb{T}}^{+})^{*}\mathbf{\Lambda}_{\mathbb{T}}^{-2\beta}\Phi_{\mathbb{T}}^{+} = \Phi_{\mathbb{T}}(\Phi_{\mathbb{T}}^{*}\Phi_{\mathbb{T}})^{-*}\mathbf{\Lambda}_{\mathbb{T}}^{-2\beta}(\Phi_{\mathbb{T}}^{*}\Phi_{\mathbb{T}})^{-1}\Phi_{\mathbb{T}}^{*}.$$

Therefore, using notations in Lemma A.1, we have that $\mathcal{P}_{\alpha,\beta}$ is simplified to

$$\mathcal{P}_{\alpha,\beta} = -\text{tr}\big(\mathbf{K}_{\mathbb{T}}\big),$$

while $\mathcal{Q}_{\alpha,\beta}$ is simplified to

$$
\begin{aligned}
\mathcal{Q}_{\alpha,\beta} &= \text{tr}\big(\Phi_{\mathbb{T}}(\Phi_{\mathbb{T}}^{*}\Phi_{\mathbb{T}})^{-*}\mathbf{\Lambda}_{\mathbb{T}}^{-2\beta}(\Phi_{\mathbb{T}}^{*}\Phi_{\mathbb{T}})^{-1}\Phi_{\mathbb{T}}^{*}\mathbf{\Lambda}_{[N]}^{-\alpha}\Psi_{\mathbb{T}^{c}}\mathbf{K}_{\mathbb{T}^{c}}\Psi_{\mathbb{T}^{c}}^{*}\mathbf{\Lambda}_{[N]}^{-\alpha}\big) \\
&= \text{tr}\big(\mathbf{\Lambda}_{[N]}^{-2\alpha}\Psi_{\mathbb{T}}(\Psi_{\mathbb{T}}^{*}\mathbf{\Lambda}_{[N]}^{-2\alpha}\Psi_{\mathbb{T}})^{-*}(\Psi_{\mathbb{T}}^{*}\mathbf{\Lambda}_{[N]}^{-2\alpha}\Psi_{\mathbb{T}})^{-1}\Psi_{\mathbb{T}}^{*}\mathbf{\Lambda}_{[N]}^{-2\alpha}\Psi_{\mathbb{T}^{c}}\mathbf{K}_{\mathbb{T}^{c}}\Psi_{\mathbb{T}^{c}}^{*}\big). \quad (47)
\end{aligned}
$$

In this regime, $p \leq N \leq P$. We therefore have $\mathbb{T}^{c} = \big([N]\backslash\mathbb{T}\big) \cup [N]^{c}$ ($[N]^{c} := [P]\backslash[N]$). Using the fact that $\mathbf{K}$ is diagonal, we obtain

$$\Psi_{\mathbb{T}^{c}}\mathbf{K}_{\mathbb{T}^{c}}\Psi_{\mathbb{T}^{c}}^{*} = \Psi_{[N]\backslash\mathbb{T}}\mathbf{K}_{[N]\backslash\mathbb{T}}\Psi_{[N]\backslash\mathbb{T}}^{*} + \Psi_{[N]^{c}}\mathbf{K}_{[N]^{c}}\Psi_{[N]^{c}}^{*}$$

Following the result of Lemma A.3, the second part of decomposition is simply the matrix $\mathbf{\Pi}_{2}$ with $\nu = 1$. To avoid confusion, we denote it by $\mathbf{\Pi}_{3}$ and use the result of Lemma A.3 to get

$$\Psi_{[N]^{c}}\mathbf{K}_{[N]^{c}}\Psi_{[N]^{c}}^{*} = \Psi_{[N]}\mathbf{\Lambda}_{\Pi_{3},2\gamma}\Psi_{[N]}^{*}$$

where the diagonal elements of $\mathbf{\Lambda}_{\Pi_{3},2\gamma}$ are

$$\lambda_{\Pi_{3},2\gamma}^{(m)} = \sum_{k=0}^{N-1}\Big(\sum_{\eta=1}^{\mu-1}t_{k+N\eta}^{-2\gamma}\Big)\Big(\frac{1}{N}\sum_{j=0}^{N-1}\omega_{N}^{(m-k)j}\Big), \ \ 0 \leq m \leq N-1. \quad (48)$$

Next we perform the decomposition

$$\Psi_{[N]}\mathbf{\Lambda}_{\Pi_{3},2\gamma}\Psi_{[N]}^{*} = \Psi_{\mathbb{T}}\mathbf{\Lambda}_{\Pi_{3},2\gamma\mathbb{T}}\Psi_{\mathbb{T}}^{*} + \Psi_{[N]\backslash\mathbb{T}}\mathbf{\Lambda}_{\Pi_{3},2\gamma[N]\backslash\mathbb{T}}\Psi_{[N]\backslash\mathbb{T}}^{*},$$

and define the diagonal matrices

$$\widehat{\mathbf{K}}_{1} := \mathbf{\Lambda}_{\Pi_{3},2\gamma\mathbb{T}} \quad \text{and} \quad \widehat{\mathbf{K}}_{2} := \mathbf{K}_{[N]\backslash\mathbb{T}} + \mathbf{\Lambda}_{\Pi_{3},2\gamma[N]\backslash\mathbb{T}}.$$

We can then have

$$\Psi_{\mathbb{T}^{c}}\mathbf{K}_{\mathbb{T}^{c}}\Psi_{\mathbb{T}^{c}}^{*} = \Psi_{\mathbb{T}}\widehat{\mathbf{K}}_{1}\Psi_{\mathbb{T}}^{*} + \Psi_{[N]\backslash\mathbb{T}}\widehat{\mathbf{K}}_{2}\Psi_{[N]\backslash\mathbb{T}}^{*}.$$

Plugging this into the expression for $\mathcal{Q}_{\alpha,\beta}$, we have

$$
\begin{aligned}
\mathcal{Q}_{\alpha,\beta} = &\text{tr}\big(\mathbf{\Lambda}_{[N]}^{-2\alpha}\Psi_{\mathbb{T}}(\Psi_{\mathbb{T}}^{*}\mathbf{\Lambda}_{[N]}^{-2\alpha}\Psi_{\mathbb{T}})^{-*}(\Psi_{\mathbb{T}}^{*}\mathbf{\Lambda}_{[N]}^{-2\alpha}\Psi_{\mathbb{T}})^{-1}\Psi_{\mathbb{T}}^{*}\mathbf{\Lambda}_{[N]}^{-2\alpha}\Psi_{\mathbb{T}}\widehat{\mathbf{K}}_{1}\Psi_{\mathbb{T}}^{*}\big) \\
&+ \text{tr}\big(\mathbf{\Lambda}_{[N]}^{-2\alpha}\Psi_{\mathbb{T}}(\Psi_{\mathbb{T}}^{*}\mathbf{\Lambda}_{[N]}^{-2\alpha}\Psi_{\mathbb{T}})^{-*}(\Psi_{\mathbb{T}}^{*}\mathbf{\Lambda}_{[N]}^{-2\alpha}\Psi_{\mathbb{T}})^{-1}\Psi_{\mathbb{T}}^{*}\mathbf{\Lambda}_{[N]}^{-2\alpha}\Psi_{[N]\backslash\mathbb{T}}\widehat{\mathbf{K}}_{2}\Psi_{[N]\backslash\mathbb{T}}^{*}\big). \quad (49)
\end{aligned}
$$

The first term simplifies to $\text{tr}\big(\widehat{\mathbf{K}}_{1}\big)$. The second term vanishes in the case of $\alpha = 0$ and in the case when the problem is formally determined; that is, when $p = N$, but is in general nonzero when $p < N$. Using the result in part (ii) of Lemma A.3, we have that

$$
\begin{aligned}
&\text{tr}\big(\mathbf{\Lambda}_{[N]}^{-2\alpha}\Psi_{\mathbb{T}}(\Psi_{\mathbb{T}}^{*}\mathbf{\Lambda}_{[N]}^{-2\alpha}\Psi_{\mathbb{T}})^{-*}(\Psi_{\mathbb{T}}^{*}\mathbf{\Lambda}_{[N]}^{-2\alpha}\Psi_{\mathbb{T}})^{-1}\Psi_{\mathbb{T}}^{*}\mathbf{\Lambda}_{[N]}^{-2\alpha}\Psi_{[N]\backslash\mathbb{T}}\widehat{\mathbf{K}}_{2}\Psi_{[N]\backslash\mathbb{T}}^{*}\big) \\
=\ &\text{tr}\big(\Psi_{[N]\backslash\mathbb{T}}^{*}\mathbf{\Lambda}_{[N]}^{-2\alpha}\Psi_{\mathbb{T}}(\Psi_{\mathbb{T}}^{*}\mathbf{\Lambda}_{[N]}^{-2\alpha}\Psi_{\mathbb{T}})^{-*}(\Psi_{\mathbb{T}}^{*}\mathbf{\Lambda}_{[N]}^{-2\alpha}\Psi_{\mathbb{T}})^{-1}\Psi_{\mathbb{T}}^{*}\mathbf{\Lambda}_{[N]}^{-2\alpha}\Psi_{[N]\backslash\mathbb{T}}\widehat{\mathbf{K}}_{2}\big) \\
=\ &N\text{tr}\big(\mathbf{X}^{-*}\Psi_{[N]\backslash\mathbb{T}}^{*}\mathbf{\Lambda}_{[N]}^{4\alpha}\Psi_{[N]\backslash\mathbb{T}}\mathbf{X}^{-1}\widehat{\mathbf{K}}_{2}\big) - \text{tr}\big(\widehat{\mathbf{K}}_{2}\big), \ \ \mathbf{X} := \Psi_{[N]\backslash\mathbb{T}}^{*}\mathbf{\Lambda}_{[N]}^{2\alpha}\Psi_{[N]\backslash\mathbb{T}}.
\end{aligned}
$$

Using this result and the singular value decomposition

$$\mathbf{\Lambda}_{[N]}^{\alpha}\Psi_{[N]\backslash\mathbb{T}} = \mathbf{U}\text{diag}([\Sigma_{00}, \cdots, \Sigma_{(N-p-1)(N-p-1)}])\mathbf{V}^{*}$$

introduced in Theorem 3.1, we have

$$
\begin{aligned}
\mathcal{Q}_{\alpha,\beta} &= \operatorname{tr}\big(\widehat{\mathbf{K}}_1 - \widehat{\mathbf{K}}_2\big) + N\operatorname{tr}\big(\boldsymbol{\Sigma}^{-1}\mathbf{U}^*\boldsymbol{\Lambda}_{[N]}^{2\alpha}\mathbf{U}\boldsymbol{\Sigma}^{-1}\mathbf{V}^*\widehat{\mathbf{K}}_2\mathbf{V}\big) \\
&= \operatorname{tr}\big(\boldsymbol{\Lambda}_{\Pi_3,2\gamma_{\mathbb{T}}} - \mathbf{K}_{[N]\backslash\mathbb{T}} - \boldsymbol{\Lambda}_{\Pi_3,2\gamma_{[N]\backslash\mathbb{T}}}\big) + N\sum_{i=0}^{N-p-1}\sum_{j=0}^{N-p-1}\frac{\widetilde{e}_{ij}^{(N)}\widehat{e}_{ji}^{(N)}}{\Sigma_{ii}\Sigma_{jj}}\,.
\end{aligned}
$$

where

$$
\widetilde{e}_{ij}^{(N)} = \sum_{k=0}^{N-1} t_k^{2\alpha}\overline{U}_{ki}U_{kj}, \quad \widehat{e}_{ij}^{(N)} = \sum_{k'=0}^{N-p-1}(c_\gamma t_{p+k'}^{-2\gamma} + \lambda_{\Pi_3,2\gamma}^{(p+k')})\overline{V}_{ik'}V_{jk'}, \quad 0 \leq i,j \leq N-p-1\,.
$$

The proof is complete when we insert the expressions of $\mathcal{P}_{\alpha,\beta}$ and $\mathcal{Q}_{\alpha,\beta}$ back to the general formula (28) and use the form of $\lambda_{\Pi_3,2\gamma}^{(m)}$ in (48). $\qquad\square$

When the problem is formally determined, i.e., in the case of $N = p$ (equivalent to $\nu = 1$), $\Psi_{\mathbb{T}}^*\boldsymbol{\Lambda}_{[N]}^{-2\alpha}\Psi_{\mathbb{T}} = \Psi_{\mathbb{T}}\boldsymbol{\Lambda}_{[N]}^{-2\alpha}\Psi_{\mathbb{T}}^*$. This allows us to find that

$$
\begin{aligned}
\mathcal{Q}_{\alpha,\beta} &= \operatorname{tr}\big(\frac{1}{N^2}\Psi_{[N]}\boldsymbol{\Lambda}_{\Pi_1,2\alpha}^{-1}\Psi_{[N]}^*\Psi_{[N]}\boldsymbol{\Lambda}_{\Pi_1,4\alpha}\Psi_{[N]}^*\Psi_{[N]}\boldsymbol{\Lambda}_{\Pi_1,2\alpha}^{-1}\Psi_{[N]}^*\frac{1}{N^2}\Psi_{[N]}\boldsymbol{\Lambda}_{\Pi_2,2\gamma}\Psi_{[N]}^*\big) \\
&= \operatorname{tr}\big(\boldsymbol{\Lambda}_{\Pi_1,2\alpha}^{-2}\boldsymbol{\Lambda}_{\Pi_1,4\alpha}\boldsymbol{\Lambda}_{\Pi_2,2\gamma}\big) = \sum_{k=0}^{N-1}\frac{\big(\sum_{\eta=0}^{\nu-1}t_{k+N\eta}^{-4\alpha}\big)\big(\sum_{\eta=\nu}^{\mu-1}t_{k+N\eta}^{-2\gamma}\big)}{\big[\sum_{\eta=0}^{\nu-1}t_{k+N\eta}^{-2\alpha}\big]^2}\,,
\end{aligned}
$$

which degenerates to its form (46) in the overparameterized regime with $\nu = 1$. The result is then independent of $\alpha$ since the terms that involve $\alpha$ cancel each other. Similarly, the simplification of $\mathcal{Q}_{\alpha,\beta}$ in (46) with $\nu = 1$ ($N = p$) also leads to the fact that $\beta$ disappears from the formula.

## A.3 PROOFS OF LEMMA 4.1 AND THEOREM 4.2

*Proof of Lemma 4.1.* By Lemma A.2, the main task is to estimate the size of the generalization error caused by the random noise in the two regimes. The error is, according to (36),

$$
\mathcal{E}_{\text{noise}}(P,p,N) = \operatorname{tr}\big(\boldsymbol{\Lambda}_{[N]}^{-\alpha}(\Phi_{\mathbb{T}}^+)^*\boldsymbol{\Lambda}_{\mathbb{T}}^{-2\beta}\Phi_{\mathbb{T}}^+\boldsymbol{\Lambda}_{[N]}^{-\alpha}\mathbb{E}_{\boldsymbol{\theta}}[\boldsymbol{\delta}\boldsymbol{\delta}^*]\big)\,.
$$

In the overparameterized regime, we have that

$$
\begin{aligned}
\boldsymbol{\Lambda}_{[N]}^{-\alpha}(\Phi_{\mathbb{T}}^+)^*\boldsymbol{\Lambda}_{\mathbb{T}}^{-2\beta}\Phi_{\mathbb{T}}^+\boldsymbol{\Lambda}_{[N]}^{-\alpha} &= \boldsymbol{\Lambda}_{[N]}^{-\alpha}(\Phi_{\mathbb{T}}\Phi_{\mathbb{T}}^*)^{-*}\Phi_{\mathbb{T}}\boldsymbol{\Lambda}_{\mathbb{T}}^{-2\beta}\Phi_{\mathbb{T}}^*(\Phi_{\mathbb{T}}\Phi_{\mathbb{T}}^*)^{-1}\boldsymbol{\Lambda}_{[N]}^{-\alpha} \\
&= (\Psi_{\mathbb{T}}\boldsymbol{\Lambda}_{\mathbb{T}}^{-2\beta}\Psi_{\mathbb{T}}^*)^{-*}\Psi_{\mathbb{T}}\boldsymbol{\Lambda}_{\mathbb{T}}^{-4\beta}\Psi_{\mathbb{T}}^*(\Psi_{\mathbb{T}}\boldsymbol{\Lambda}_{\mathbb{T}}^{-2\beta}\Psi_{\mathbb{T}}^*)^{-1}\,.
\end{aligned}
$$

Therefore we have, using the results in part (i) of Lemma A.3,

$$
\begin{aligned}
\mathcal{E}_{\text{noise}}(P,p,N) &= \sigma^2\operatorname{tr}\big(\frac{1}{N^2}\Psi_{[N]}\boldsymbol{\Lambda}_{\Pi_1,2\beta}^{-1}\Psi_{[N]}^*\Psi_{[N]}\boldsymbol{\Lambda}_{\Pi_1,4\beta}\Psi_{[N]}^*\frac{1}{N^2}\Psi_{[N]}\boldsymbol{\Lambda}_{\Pi_1,2\beta}^{-1}\Psi_{[N]}^*\big) \\
&= \frac{\sigma^2}{N}\operatorname{tr}\big(\boldsymbol{\Lambda}_{\Pi_1,2\beta}^{-1}\boldsymbol{\Lambda}_{\Pi_1,4\beta}\boldsymbol{\Lambda}_{\Pi_1,2\beta}^{-1}\big) = \frac{\sigma^2}{N}\sum_{k=0}^{N-1}\frac{\sum_{\eta=0}^{\nu-1}t_{k+N\eta}^{-4\beta}}{\big[\sum_{\eta=0}^{\nu-1}t_{k+N\eta}^{-2\beta}\big]^2}\,.
\end{aligned}
$$

To get the variance of the generalization error with respect to noise, we first compute

$$
\mathbb{E}_{\boldsymbol{\delta}}[\mathrm{tr}\big(\boldsymbol{\delta}^*\boldsymbol{\Lambda}_{[N]}^{-\alpha}(\Phi_{\mathbb{T}}^+)^*\boldsymbol{\Lambda}_{\mathbb{T}}^{-2\beta}\Phi_{\mathbb{T}}^+\boldsymbol{\Lambda}_{[N]}^{-\alpha}\boldsymbol{\delta}\boldsymbol{\delta}^*\boldsymbol{\Lambda}_{[N]}^{-\alpha}(\Phi_{\mathbb{T}}^+)^*\boldsymbol{\Lambda}_{\mathbb{T}}^{-2\beta}\Phi_{\mathbb{T}}^+\boldsymbol{\Lambda}_{[N]}^{-\alpha}\boldsymbol{\delta}\big)]
$$

$$
= \quad \sigma^4\mathrm{tr}\big(\boldsymbol{\Lambda}_{[N]}^{-\alpha}(\Phi_{\mathbb{T}}^+)^*\boldsymbol{\Lambda}_{\mathbb{T}}^{-2\beta}\Phi_{\mathbb{T}}^+\boldsymbol{\Lambda}_{[N]}^{-\alpha}\big)\mathrm{tr}\big(\boldsymbol{\Lambda}_{[N]}^{-\alpha}(\Phi_{\mathbb{T}}^+)^*\boldsymbol{\Lambda}_{\mathbb{T}}^{-2\beta}\Phi_{\mathbb{T}}^+\boldsymbol{\Lambda}_{[N]}^{-\alpha}\big)
$$

$$
+2\sigma^4\mathrm{tr}\big(\boldsymbol{\Lambda}_{[N]}^{-\alpha}(\Phi_{\mathbb{T}}^+)^*\boldsymbol{\Lambda}_{\mathbb{T}}^{-2\beta}\Phi_{\mathbb{T}}^+\boldsymbol{\Lambda}_{[N]}^{-\alpha}\boldsymbol{\Lambda}_{[N]}^{-\alpha}(\Phi_{\mathbb{T}}^+)^*\boldsymbol{\Lambda}_{\mathbb{T}}^{-2\beta}\Phi_{\mathbb{T}}^+\boldsymbol{\Lambda}_{[N]}^{-\alpha}\big)
$$

$$
= \quad \frac{\sigma^4}{N^4}\mathrm{tr}\big(\Psi_{[N]}\boldsymbol{\Lambda}_{\Pi_1,2\beta}^{-1}\boldsymbol{\Lambda}_{\Pi_1,4\beta}\boldsymbol{\Lambda}_{\Pi_1,2\beta}^{-1}\Psi_{[N]}^*\big)\mathrm{tr}\big(\Psi_{[N]}\boldsymbol{\Lambda}_{\Pi_1,2\beta}^{-1}\boldsymbol{\Lambda}_{\Pi_1,4\beta}\boldsymbol{\Lambda}_{\Pi_1,2\beta}^{-1}\Psi_{[N]}^*\big)
$$

$$
+\frac{2\sigma^4}{N^4}\mathrm{tr}\big(\Psi_{[N]}\boldsymbol{\Lambda}_{\Pi_1,2\beta}^{-1}\boldsymbol{\Lambda}_{\Pi_1,4\beta}\boldsymbol{\Lambda}_{\Pi_1,2\beta}^{-1}\Psi_{[N]}^*\Psi_{[N]}\boldsymbol{\Lambda}_{\Pi_1,2\beta}^{-1}\boldsymbol{\Lambda}_{\Pi_1,4\beta}\boldsymbol{\Lambda}_{\Pi_1,2\beta}^{-1}\Psi_{[N]}^*\big)
$$

$$
= \quad \frac{\sigma^4}{N^2}\mathrm{tr}\big(\boldsymbol{\Lambda}_{\Pi_1,2\beta}^{-1}\boldsymbol{\Lambda}_{\Pi_1,4\beta}\boldsymbol{\Lambda}_{\Pi_1,2\beta}^{-1}\big)\mathrm{tr}\big(\boldsymbol{\Lambda}_{\Pi_1,2\beta}^{-1}\boldsymbol{\Lambda}_{\Pi_1,4\beta}\boldsymbol{\Lambda}_{\Pi_1,2\beta}^{-1}\big)
$$

$$
+\frac{2\sigma^4}{N^2}\mathrm{tr}\big(\boldsymbol{\Lambda}_{\Pi_1,2\beta}^{-1}\boldsymbol{\Lambda}_{\Pi_1,4\beta}\boldsymbol{\Lambda}_{\Pi_1,2\beta}^{-1}\boldsymbol{\Lambda}_{\Pi_1,2\beta}^{-1}\boldsymbol{\Lambda}_{\Pi_1,4\beta}\boldsymbol{\Lambda}_{\Pi_1,2\beta}^{-1}\big)
$$

$$
= \quad \frac{\sigma^4}{N^2}\Big[\sum_{k=0}^{N-1}\frac{\sum_{\eta=0}^{\nu-1}t_{k+N\eta}^{-4\beta}}{\big[\sum_{\eta=0}^{\nu-1}t_{k+N\eta}^{-2\beta}\big]^2}\Big]^2 + \frac{2\sigma^4}{N^2}\sum_{k=0}^{N-1}\frac{\big[\sum_{\eta=0}^{\nu-1}t_{k+N\eta}^{-4\beta}\big]^2}{\big[\sum_{\eta=0}^{\nu-1}t_{k+N\eta}^{-2\beta}\big]^4}.
$$

This, together with the general form of the variance in (37), gives the result in (18).

In the underparameterized regime, we have that

$$
\boldsymbol{\Lambda}_{[N]}^{-\alpha}(\Phi_{\mathbb{T}}^+)^*\boldsymbol{\Lambda}_{\mathbb{T}}^{-2\beta}\Phi_{\mathbb{T}}^+\boldsymbol{\Lambda}_{[N]}^{-\alpha} = \boldsymbol{\Lambda}_{[N]}^{-2\alpha}\Psi_{\mathbb{T}}(\Psi_{\mathbb{T}}^*\boldsymbol{\Lambda}_{[N]}^{-2\alpha}\Psi_{\mathbb{T}})^{-*}(\Psi_{\mathbb{T}}^*\boldsymbol{\Lambda}_{[N]}^{-2\alpha}\Psi_{\mathbb{T}})^{-1}\Psi_{\mathbb{T}}^*\boldsymbol{\Lambda}_{[N]}^{-2\alpha}.
$$

Therefore we have, using the fact that $\Psi_{\mathbb{T}}\Psi_{\mathbb{T}}^* + \Psi_{[N]\backslash\mathbb{T}}\Psi_{[N]\backslash\mathbb{T}}^* = N\mathbf{I}_{[N]}$, that

$$
\boldsymbol{\Lambda}_{[N]}^{-\alpha}(\Phi_{\mathbb{T}}^+)^*\boldsymbol{\Lambda}_{\mathbb{T}}^{-2\beta}\Phi_{\mathbb{T}}^+\boldsymbol{\Lambda}_{[N]}^{-\alpha} \tag{50}
$$

$$
= \quad \frac{1}{N^2}\Big(\Psi_{\mathbb{T}}\Psi_{\mathbb{T}}^* + \Psi_{[N]\backslash\mathbb{T}}\Psi_{[N]\backslash\mathbb{T}}^*\Big)\boldsymbol{\Lambda}_{[N]}^{-\alpha}(\Phi_{\mathbb{T}}^+)^*\boldsymbol{\Lambda}_{\mathbb{T}}^{-2\beta}\Phi_{\mathbb{T}}^+\boldsymbol{\Lambda}_{[N]}^{-\alpha}\Big(\Psi_{\mathbb{T}}\Psi_{\mathbb{T}}^* + \Psi_{[N]\backslash\mathbb{T}}\Psi_{[N]\backslash\mathbb{T}}^*\Big)
$$

$$
= \quad \frac{1}{N^2}\Psi_{\mathbb{T}}\Psi_{\mathbb{T}}^* + \frac{1}{N^2}\Psi_{\mathbb{T}}(\Psi_{\mathbb{T}}^*\boldsymbol{\Lambda}_{[N]}^{-2\alpha}\Psi_{\mathbb{T}})^{-1}\Psi_{\mathbb{T}}^*\boldsymbol{\Lambda}_{[N]}^{-2\alpha}\Psi_{[N]\backslash\mathbb{T}}\Psi_{[N]\backslash\mathbb{T}}^*
$$

$$
+\frac{1}{N^2}\Psi_{[N]\backslash\mathbb{T}}\Psi_{[N]\backslash\mathbb{T}}^*\boldsymbol{\Lambda}_{[N]}^{-2\alpha}\Psi_{\mathbb{T}}(\Psi_{\mathbb{T}}^*\boldsymbol{\Lambda}_{[N]}^{-2\alpha}\Psi_{\mathbb{T}})^{-*}\Psi_{\mathbb{T}}^*
$$

$$
+\frac{1}{N^2}\Psi_{[N]\backslash\mathbb{T}}\Psi_{[N]\backslash\mathbb{T}}^*\boldsymbol{\Lambda}_{[N]}^{-2\alpha}\Psi_{\mathbb{T}}(\Psi_{\mathbb{T}}^*\boldsymbol{\Lambda}_{[N]}^{-2\alpha}\Psi_{\mathbb{T}})^{-*}(\Psi_{\mathbb{T}}^*\boldsymbol{\Lambda}_{[N]}^{-2\alpha}\Psi_{\mathbb{T}})^{-1}\Psi_{\mathbb{T}}^*\boldsymbol{\Lambda}_{[N]}^{-2\alpha}\Psi_{[N]\backslash\mathbb{T}}\Psi_{[N]\backslash\mathbb{T}}^*.
$$

This leads to, using the fact that $\Psi_{\mathbb{T}}^*\Psi_{\mathbb{T}} = N\mathbf{I}_{[p]}$ and properties of traces,

$$
\mathcal{E}_{\mathrm{noise}}(P,p,N) = \sigma^2\mathrm{tr}\big(\boldsymbol{\Lambda}_{[N]}^{-\alpha}(\Phi_{\mathbb{T}}^+)^*\boldsymbol{\Lambda}_{\mathbb{T}}^{-2\beta}\Phi_{\mathbb{T}}^+\boldsymbol{\Lambda}_{[N]}^{-\alpha}\big)
$$

$$
= \frac{p}{N}\sigma^2 + \frac{\sigma^2}{N}\mathrm{tr}\big(\Psi_{[N]\backslash\mathbb{T}}^*\boldsymbol{\Lambda}_{[N]}^{-2\alpha}\Psi_{\mathbb{T}}(\Psi_{\mathbb{T}}^*\boldsymbol{\Lambda}_{[N]}^{-2\alpha}\Psi_{\mathbb{T}})^{-*}(\Psi_{\mathbb{T}}^*\boldsymbol{\Lambda}_{[N]}^{-2\alpha}\Psi_{\mathbb{T}})^{-1}\Psi_{\mathbb{T}}^*\boldsymbol{\Lambda}_{[N]}^{-2\alpha}\Psi_{[N]\backslash\mathbb{T}}\big)
$$

Using the result in part (ii) of Lemma A.3, we have that the second term simplifies to

$$
\mathrm{tr}\big(\Psi_{[N]\backslash\mathbb{T}}^*\boldsymbol{\Lambda}_{[N]}^{-2\alpha}\Psi_{\mathbb{T}}(\Psi_{\mathbb{T}}^*\boldsymbol{\Lambda}_{[N]}^{-2\alpha}\Psi_{\mathbb{T}})^{-*}(\Psi_{\mathbb{T}}^*\boldsymbol{\Lambda}_{[N]}^{-2\alpha}\Psi_{\mathbb{T}})^{-1}\Psi_{\mathbb{T}}^*\boldsymbol{\Lambda}_{[N]}^{-2\alpha}\Psi_{[N]\backslash\mathbb{T}}\big)
$$

$$
= N\mathrm{tr}\big(\mathbf{X}^{-*}\Psi_{[N]\backslash\mathbb{T}}^*\boldsymbol{\Lambda}_{[N]}^{4\alpha}\Psi_{[N]\backslash\mathbb{T}}\mathbf{X}^{-1}\big) - (N-p), \quad \mathbf{X} := \Psi_{[N]\backslash\mathbb{T}}^*\boldsymbol{\Lambda}_{[N]}^{2\alpha}\Psi_{[N]\backslash\mathbb{T}}.
$$

Using this result and the singular value decomposition

$$
\boldsymbol{\Lambda}_{[N]}^{\alpha}\Psi_{[N]\backslash\mathbb{T}} = \mathbf{U}\mathrm{diag}([\Sigma_{00},\cdots,\Sigma_{(N-p-1)(N-p-1)}])\mathbf{V}^*
$$

introduced in Theorem 3.1, we have

$$
\mathcal{E}_{\mathrm{noise}}(P,p,N) = \frac{p}{N}\sigma^2 - \frac{N-p}{N}\sigma^2 + \sigma^2\mathrm{tr}\big(\boldsymbol{\Sigma}^{-1}\mathbf{U}^*\boldsymbol{\Lambda}_{[N]}^{2\alpha}\mathbf{U}\boldsymbol{\Sigma}^{-1}\big) = \sigma^2\Big(\frac{2p-N}{N} + \sum_{j=0}^{N-p-1}\frac{\widetilde{e}_{jj}^{(N)}}{\Sigma_{jj}^2}\Big).
$$

Following the same procedure as in the overparameterized regime, the variance of the generalization error with respect to noise in this case is

$$
\begin{aligned}
&\mathbb{E}_{\boldsymbol{\delta}}[\mathrm{tr}\big(\boldsymbol{\delta}^*\boldsymbol{\Lambda}_{[N]}^{-\alpha}(\Phi_{\mathbb{T}}^+)^*\boldsymbol{\Lambda}_{\mathbb{T}}^{-2\beta}\Phi_{\mathbb{T}}^+\boldsymbol{\Lambda}_{[N]}^{-\alpha}\boldsymbol{\delta}\boldsymbol{\delta}^*\boldsymbol{\Lambda}_{[N]}^{-\alpha}(\Phi_{\mathbb{T}}^+)^*\boldsymbol{\Lambda}_{\mathbb{T}}^{-2\beta}\Phi_{\mathbb{T}}^+\boldsymbol{\Lambda}_{[N]}^{-\alpha}\boldsymbol{\delta})] \\
=\ & \sigma^4\mathrm{tr}\big(\boldsymbol{\Lambda}_{[N]}^{-\alpha}(\Phi_{\mathbb{T}}^+)^*\boldsymbol{\Lambda}_{\mathbb{T}}^{-2\beta}\Phi_{\mathbb{T}}^+\boldsymbol{\Lambda}_{[N]}^{-\alpha}\big)\mathrm{tr}\big(\boldsymbol{\Lambda}_{[N]}^{-\alpha}(\Phi_{\mathbb{T}}^+)^*\boldsymbol{\Lambda}_{\mathbb{T}}^{-2\beta}\Phi_{\mathbb{T}}^+\boldsymbol{\Lambda}_{[N]}^{-\alpha}\big) \\
& +2\sigma^4\mathrm{tr}\big(\boldsymbol{\Lambda}_{[N]}^{-\alpha}(\Phi_{\mathbb{T}}^+)^*\boldsymbol{\Lambda}_{\mathbb{T}}^{-2\beta}\Phi_{\mathbb{T}}^+\boldsymbol{\Lambda}_{[N]}^{-\alpha}\boldsymbol{\Lambda}_{[N]}^{-\alpha}(\Phi_{\mathbb{T}}^+)^*\boldsymbol{\Lambda}_{\mathbb{T}}^{-2\beta}\Phi_{\mathbb{T}}^+\boldsymbol{\Lambda}_{[N]}^{-\alpha}\big)
\end{aligned}
$$

The first term on the right hand side is simply $\big(\mathcal{E}_{\mathrm{noise}}(P,p,N)\big)^2$. To evaluate the second term, we use the formula (50). It is straightforward to obtain, after some algebra, that

$$
\begin{aligned}
&\mathrm{tr}\big(\boldsymbol{\Lambda}_{[N]}^{-\alpha}(\Phi_{\mathbb{T}}^+)^*\boldsymbol{\Lambda}_{\mathbb{T}}^{-2\beta}\Phi_{\mathbb{T}}^+\boldsymbol{\Lambda}_{[N]}^{-\alpha}\boldsymbol{\Lambda}_{[N]}^{-\alpha}(\Phi_{\mathbb{T}}^+)^*\boldsymbol{\Lambda}_{\mathbb{T}}^{-2\beta}\Phi_{\mathbb{T}}^+\boldsymbol{\Lambda}_{[N]}^{-\alpha}\big) \\
=\ & \frac{1}{N^3}\mathrm{tr}\big(\Psi_{\mathbb{T}}\Psi_{\mathbb{T}}^*\big)+\frac{2}{N^2}\mathrm{tr}\big(\Psi_{[N]\backslash\mathbb{T}}^*\boldsymbol{\Lambda}_{[N]}^{-2\alpha}\Psi_{\mathbb{T}}(\Psi_{\mathbb{T}}^*\boldsymbol{\Lambda}_{[N]}^{-2\alpha}\Psi_{\mathbb{T}})^{-*}(\Psi_{\mathbb{T}}^*\boldsymbol{\Lambda}_{[N]}^{-2\alpha}\Psi_{\mathbb{T}})^{-1}\Psi_{\mathbb{T}}^*\boldsymbol{\Lambda}_{[N]}^{-2\alpha}\Psi_{[N]\backslash\mathbb{T}}\big) \\
& +\frac{1}{N^2}\mathrm{tr}\big(\big[\Psi_{[N]\backslash\mathbb{T}}^*\boldsymbol{\Lambda}_{[N]}^{-2\alpha}\Psi_{\mathbb{T}}(\Psi_{\mathbb{T}}^*\boldsymbol{\Lambda}_{[N]}^{-2\alpha}\Psi_{\mathbb{T}})^{-*}(\Psi_{\mathbb{T}}^*\boldsymbol{\Lambda}_{[N]}^{-2\alpha}\Psi_{\mathbb{T}})^{-1}\Psi_{\mathbb{T}}^*\boldsymbol{\Lambda}_{[N]}^{-2\alpha}\Psi_{[N]\backslash\mathbb{T}}\big]^2\big) \\
=\ & \frac{p}{N^2}+\frac{2}{N^2}\Big(N\mathrm{tr}\big(\mathbf{X}^{-*}\Psi_{[N]\backslash\mathbb{T}}^*\boldsymbol{\Lambda}_{[N]}^{4\alpha}\Psi_{[N]\backslash\mathbb{T}}\mathbf{X}^{-1}\big)-(N-p)\Big) \\
& +\frac{1}{N^2}\Big(N^2\mathrm{tr}\big(\mathbf{X}^{-*}\Psi_{[N]\backslash\mathbb{T}}^*\boldsymbol{\Lambda}_{[N]}^{4\alpha}\Psi_{[N]\backslash\mathbb{T}}\mathbf{X}^{-1}\mathbf{X}^{-*}\Psi_{[N]\backslash\mathbb{T}}^*\boldsymbol{\Lambda}_{[N]}^{4\alpha}\Psi_{[N]\backslash\mathbb{T}}\mathbf{X}^{-1}\big) \\
& -2N\mathrm{tr}\big(\mathbf{X}^{-*}\Psi_{[N]\backslash\mathbb{T}}^*\boldsymbol{\Lambda}_{[N]}^{4\alpha}\Psi_{[N]\backslash\mathbb{T}}\mathbf{X}^{-1}\big)+\mathrm{tr}\big(\mathbf{I}_{[N-p]}\big)\Big) \\
=\ & \frac{2p-N}{N^2}+\mathrm{tr}\big(\mathbf{X}^{-*}\Psi_{[N]\backslash\mathbb{T}}^*\boldsymbol{\Lambda}_{[N]}^{4\alpha}\Psi_{[N]\backslash\mathbb{T}}\mathbf{X}^{-1}\mathbf{X}^{-*}\Psi_{[N]\backslash\mathbb{T}}^*\boldsymbol{\Lambda}_{[N]}^{4\alpha}\Psi_{[N]\backslash\mathbb{T}}\mathbf{X}^{-1}\big) \\
=\ & \frac{2p-N}{N^2}+\mathrm{tr}\big(\mathbf{X}^{-*}\Psi_{[N]\backslash\mathbb{T}}^*\boldsymbol{\Lambda}_{[N]}^{4\alpha}\Psi_{[N]\backslash\mathbb{T}}\mathbf{X}^{-1}\mathbf{X}^{-*}\Psi_{[N]\backslash\mathbb{T}}^*\boldsymbol{\Lambda}_{[N]}^{4\alpha}\Psi_{[N]\backslash\mathbb{T}}\mathbf{X}^{-1}\big)
\end{aligned}
$$

This leads to

$$
\begin{aligned}
&\mathbb{E}_{\boldsymbol{\delta}}[\mathrm{tr}\big(\boldsymbol{\delta}^*\boldsymbol{\Lambda}_{[N]}^{-\alpha}(\Phi_{\mathbb{T}}^+)^*\boldsymbol{\Lambda}_{\mathbb{T}}^{-2\beta}\Phi_{\mathbb{T}}^+\boldsymbol{\Lambda}_{[N]}^{-\alpha}\boldsymbol{\delta}\boldsymbol{\delta}^*\boldsymbol{\Lambda}_{[N]}^{-\alpha}(\Phi_{\mathbb{T}}^+)^*\boldsymbol{\Lambda}_{\mathbb{T}}^{-2\beta}\Phi_{\mathbb{T}}^+\boldsymbol{\Lambda}_{[N]}^{-\alpha}\boldsymbol{\delta})] \\
=\ & \big(\mathcal{E}_{\mathrm{noise}}(P,p,N)\big)^2+2\sigma^4\Big(\frac{2p-N}{N^2}+\mathrm{tr}\big(\boldsymbol{\Sigma}^{-2}\mathbf{U}_{\mathbb{T}}^*\boldsymbol{\Lambda}_{[N]}^{2\alpha}\mathbf{U}_{\mathbb{T}}\boldsymbol{\Sigma}^{-2}\mathbf{U}_{\mathbb{T}}^*\boldsymbol{\Lambda}_{[N]}^{2\alpha}\mathbf{U}_{\mathbb{T}}\big)\Big) \\
=\ & \big(\mathcal{E}_{\mathrm{noise}}(P,p,N)\big)^2+2\sigma^4\Big(\frac{2p-N}{N^2}+\sum_{i=0}^{N-p-1}\sum_{j=0}^{N-p-1}\frac{\widetilde{e}_{ij}^{(N)}\widetilde{e}_{ji}^{(N)}}{\Sigma_{ii}^2\Sigma_{jj}^2}\Big).
\end{aligned}
$$

Inserting this into the general form of the variance, i.e. (37), will lead to (19) for the underparameterized regime. The proof is complete. $\qquad\square$

*Proof of Theorem 4.2.* By standard decomposition, we have that

$$
\begin{aligned}
\|\widehat{\boldsymbol{\theta}}^{\delta}-\boldsymbol{\theta}\|_2^2 &= \|\widehat{\boldsymbol{\theta}}^{\delta}-\widehat{\boldsymbol{\theta}}+\widehat{\boldsymbol{\theta}}-\boldsymbol{\theta}\|_2^2 \\
&= \left\|\begin{pmatrix}\Psi_{\mathbb{T}}^+ \\ \mathbf{O}_{(P-p)\times N}\end{pmatrix}(\mathbf{y}^{\delta}-\mathbf{y})+\big(\mathbf{I}_{[P]}-\begin{pmatrix}\Psi_{\mathbb{T}}^+\Psi \\ \mathbf{O}_{(P-p)\times P}\end{pmatrix}\big)\boldsymbol{\theta}\right\|_2^2 \\
&\leq 2\|\Psi_{\mathbb{T}}^+\boldsymbol{\delta}\|_2^2+2\left\|\big(\mathbf{I}_{[P]}-\begin{pmatrix}\Psi_{\mathbb{T}}^+\Psi \\ \mathbf{O}_{(P-p)\times P}\end{pmatrix}\big)\boldsymbol{\theta}\right\|_2^2.
\end{aligned}
\tag{51}
$$

From the calculations in the previous sections and the normalization of $\Psi$ we used, it is straightforward to verify that, for any fixed $N$,

$$
\|\Psi_{\mathbb{T}}^+\|_{2,\boldsymbol{\Lambda}_{[N]}^{-\alpha}\mapsto 2}\sim\begin{cases}\frac{1}{\sqrt{\nu N}}p^{-\alpha}=p^{-\frac{1}{2}-\alpha}, & p=\nu N \\ p^{-\alpha}, & p<N\end{cases}
$$

and

$$
\left\|\mathbf{I}_{[P]}-\begin{pmatrix}\Psi_{\mathbb{T}}^+\Psi \\ \mathbf{O}_{(P-p)\times P}\end{pmatrix}\right\|_{2,\boldsymbol{\Lambda}_{[P]}^{-\beta}\mapsto 2}\sim p^{-\beta},
$$

where $\|\mathbf{X}\|_{2,\boldsymbol{\Lambda}_{[N]}^{-\alpha}\mapsto 2}$ denotes the norm of $\mathbf{X}$ as an operator from the $\boldsymbol{\Lambda}_{[N]}^{-\alpha}$-weighted $\mathbb{C}^N$ to $\mathbb{C}^p$.

This allows us to conclude, after taking expectation with respect to $\boldsymbol{\theta}$ and then $\boldsymbol{\delta}$, that

$$\mathbb{E}_{\boldsymbol{\delta},\boldsymbol{\theta}}[\|\widehat{\boldsymbol{\theta}}^{\boldsymbol{\delta}} - \boldsymbol{\theta}\|_2^2] \leq 2\|\Psi^+\|_{2,\boldsymbol{\Lambda}_{[N]}^{-\alpha}\mapsto 2}^2 \mathbb{E}_{\boldsymbol{\delta}}[\|\boldsymbol{\Lambda}_{[N]}^{-\alpha}(\mathbf{y}^{\boldsymbol{\delta}} - \mathbf{y})\|_2^2]$$

$$+ 2\|(\mathbf{I}_{[P]} - \Psi^+\Psi)\|_{2,\boldsymbol{\Lambda}_{[P]}^{-\beta}\mapsto 2}^2 \mathbb{E}_{\boldsymbol{\theta}}[\|\boldsymbol{\theta}\|_{2,\boldsymbol{\Lambda}_{[P]}^{-\beta}}^2] \lesssim \rho p^{-2\alpha}\mathbb{E}_{\boldsymbol{\delta}}[\|\boldsymbol{\delta}\|_{2,\boldsymbol{\Lambda}_{[N]}^{-\alpha}}^2] + p^{-2\beta}\mathbb{E}_{\boldsymbol{\theta}}[\|\boldsymbol{\theta}\|_{2,\boldsymbol{\Lambda}_{[P]}^{-\beta}}^2],$$

where $\rho = \min(1, N/p)$. For any fixed $N$, when $\alpha > -1/2$ in the over-parameterized regime (resp. $\alpha > 0$ in the under-parameterized regime), the error decrease monotonically with respect to $p$. When $\alpha < -1/2$ in the over-parameterized regime (resp. $\alpha < 0$ in the under-parameterized regime), the first term increase with $p$ while the second term descreases with $p$. To minimize the right-hand side, we take

$$p \sim (\mathbb{E}_{\boldsymbol{\delta}}[\|\boldsymbol{\delta}\|_{2,\boldsymbol{\Lambda}_{[N]}^{-\alpha}}^2]^{-1}\mathbb{E}_{\boldsymbol{\theta}}[\|\boldsymbol{\theta}\|_{2,\boldsymbol{\Lambda}_{[P]}^{-\beta}}^2])^{\frac{1}{2(\beta-\widehat{\alpha})}}, \tag{52}$$

where $\widehat{\alpha} := \alpha + \frac{1}{2}$ in the over-parameterized regime and $\widehat{\alpha} = \alpha$ in the under-parameterized regime. This leads to

$$\mathbb{E}_{\boldsymbol{\theta},\boldsymbol{\delta}}[\|\widehat{\boldsymbol{\theta}}^{\boldsymbol{\delta}} - \boldsymbol{\theta}\|_2^2] \lesssim \mathbb{E}_{\boldsymbol{\theta}}[\|\boldsymbol{\theta}\|_{2,\boldsymbol{\Lambda}_{[P]}^{-\beta}}^2]^{\frac{-2\widehat{\alpha}}{2(\beta-\widehat{\alpha})}} \mathbb{E}_{\boldsymbol{\delta}}[\|\boldsymbol{\delta}\|_{2,\boldsymbol{\Lambda}_{[N]}^{-\alpha}}^2]^{\frac{2\beta}{2(\beta-\widehat{\alpha})}}. \tag{53}$$

The proof is now complete. $\qquad\qquad\square$

## A.4 PROOF OF THEOREM 5.1

*Proof of Theorem 5.1.* Without loss of generality, we assume that $\Psi$ is diagonal with diagonal elements $\Psi_{kk} \sim k^{-\zeta}$. If not, we can rescale the weight matrix $\boldsymbol{\Lambda}_{[p]}^{-\beta}$ by $\mathbf{V}$ and weight matrix $\boldsymbol{\Lambda}_{[N]}^{-\alpha}$ by $\mathbf{U}$ as given in the theorem.

Let $\Psi^+$ be the pseudoinverse of $\Psi$ that consists of the first $p$ features such that

$$(\Psi^+)_{qq} \sim \begin{cases} q^{\zeta}, & q \leq p \\ 0, & q > p \end{cases}$$

where $\zeta$ is the exponent in the SVD of $\Psi$ assumed in Theorem 5.1.

We can then check that the operators $\mathbf{I} - \Psi^+\Psi : \ell_{\boldsymbol{\Lambda}_{[P]}^{-\beta}}^2 \mapsto \ell^2$ and $\Psi^+ : \ell_{\boldsymbol{\Lambda}_{[N]}^{-\alpha}}^2 \mapsto \ell^2$ have the following norms respectively

$$\|\Psi^+\|_{2,\boldsymbol{\Lambda}_{[N]}^{-\alpha}\mapsto 2} \sim p^{\zeta-\alpha} \qquad \text{and} \qquad \|(\mathbf{I} - \Psi^+\Psi)\|_{2,\boldsymbol{\Lambda}_{[P]}^{-\beta}\mapsto 2} \sim p^{-\beta}.$$

By the error decomposition (51), we then conclude, after taking expectation with respect to $\boldsymbol{\theta}$ and then $\boldsymbol{\delta}$, that

$$\mathbb{E}_{\boldsymbol{\delta},\boldsymbol{\theta}}[\|\widehat{\boldsymbol{\theta}}_c^{\boldsymbol{\delta}} - \boldsymbol{\theta}\|_2^2] \lesssim \|\Psi^+\|_{2,\boldsymbol{\Lambda}_{[N]}^{-\alpha}\mapsto 2}^2 \mathbb{E}_{\boldsymbol{\delta}}[\|\boldsymbol{\Lambda}_{[N]}^{-\alpha}\boldsymbol{\delta}\|_2^2]$$

$$+ \|(\mathbf{I} - \Psi^+\Psi)\|_{2,\boldsymbol{\Lambda}_{[P]}^{-\beta}\mapsto 2}^2 \mathbb{E}_{\boldsymbol{\theta}}[\|\boldsymbol{\theta}\|_{2,\boldsymbol{\Lambda}_{[P]}^{-\beta}}^2] \lesssim p^{2(\zeta-\alpha)}\mathbb{E}_{\boldsymbol{\delta}}[\|\boldsymbol{\delta}\|_{2,\boldsymbol{\Lambda}_{[N]}^{-\alpha}}^2] + p^{-2\beta}\mathbb{E}_{\boldsymbol{\theta}}[\|\boldsymbol{\theta}\|_{2,\boldsymbol{\Lambda}_{[P]}^{-\beta}}^2].$$

We can now select

$$p \sim (\mathbb{E}_{\boldsymbol{\delta}}[\|\boldsymbol{\delta}\|_{2,\boldsymbol{\Lambda}_{[N]}^{-\alpha}}^2]^{-1}\mathbb{E}_{\boldsymbol{\theta}}[\|\boldsymbol{\theta}\|_{2,\boldsymbol{\Lambda}_{[P]}^{-\beta}}^2])^{\frac{1}{2(\zeta+\beta-\alpha)}} \tag{54}$$

to minimize the error. This leads to

$$\mathbb{E}_{\boldsymbol{\delta},\boldsymbol{\theta}}[\|\widehat{\boldsymbol{\theta}}^{\boldsymbol{\delta}} - \boldsymbol{\theta}\|_2^2] \lesssim \mathbb{E}_{\boldsymbol{\theta}}[\|\boldsymbol{\theta}\|_{2,\boldsymbol{\Lambda}_{[P]}^{-\beta}}^2]^{\frac{\zeta-\alpha}{(\zeta-\alpha+\beta)}} \mathbb{E}_{\boldsymbol{\delta}}[\|\boldsymbol{\delta}\|_{2,\boldsymbol{\Lambda}_{[N]}^{-\alpha}}^2]^{\frac{\beta}{(\zeta+\beta-\alpha)}}. \tag{55}$$

The proof is now complete. $\qquad\qquad\square$

## B ON THE APPLICABILITY OF THEOREM 5.1

In order for the result of Theorem 5.1 to hold, we need the model to be learned to have the smoothing property. In the case of feature regression, this requires that the feature matrices (or sampling matrices) correspond to kernels whose singular values decay algebraically. This turns out to be true for many kernel regression models in practical applications. In the case of solving inverse problems, this is extremely common as most inverse problems based on physical models have smoothing operators; see, for instance, Isakov (2006); Kirsch (2011) for a more in-depth discussion on this issue.

**General kernel regression.** Let $\mathcal{H}$ be a reproducing kernel Hilbert space (RKHS) over $X$, and $\mathcal{K} : \mathcal{H} \times \mathcal{H} \to \mathbb{R}$ be the corresponding (symmetric and positive semidefinite) reproducing kernel. We are interested in learning a function $f^*$ from given data $\{x_j, y_j^\delta\}_{j=1}^N$. The learning process can be formulated as

$$\min_{f \in \mathcal{H}} \sum_{j=1}^N \left( f(x_j) - y_j^\delta \right)^2 + \beta \|f\|_{\mathcal{H}}^2 . \tag{56}$$

Let $\{\varphi_k\}_{k \geq 0}$ be the eigenfunctions of the kernel $\mathcal{K}$ such that

$$\int \mathcal{K}(x, \tilde{x}) \varphi_k(\tilde{x}) \mu(\tilde{x}) d\tilde{x} = \lambda_k \varphi_k(x) , \tag{57}$$

where $\mu$ is the probability measure that generates the data. Following Mercer's Theorem (Rasmussen & Williams, 2006), we have that $\mathcal{K}$ admits a representation in terms of its kernel eigenfunctions; that is, $\mathcal{K}(x, \tilde{x}) = \sum_{k \geq 0} \lambda_k \varphi_k(x) \varphi_k(\tilde{x})$. Moreover, the solution to the learning problem as well as the target function can be approximated respectively as

$$f_{\boldsymbol{\theta}}^*(x) = \sum_{k=0}^{P-1} \theta_k \varphi_k(x), \quad \text{and} \quad f_{\widehat{\boldsymbol{\theta}}_p}(x) = \sum_{k=0}^{p-1} \widehat{\theta}_k \varphi_k(x) , \tag{58}$$

where to be consistent with the setup in the previous sections, we have used $P$ and $p$ to denote the numbers of modes in the target function and the learning solution respectively. We can now define $\Psi$ to be the feature matrix (with components $(\Psi)_{kj} = \varphi_k(x_j)$) so that the kernel regression problem can be recast as the optimization problem

$$\widehat{\boldsymbol{\theta}}_p^\delta = \arg\min_{\boldsymbol{\theta}} \|\Psi \boldsymbol{\theta}_p - \mathbf{y}^\delta\|_2^2 + \beta \|\boldsymbol{\theta}_p\|_2^2 . \tag{59}$$

For a given training data set consisting of $N$ data points, the generalization error for this learning problem can then be written in the same form as (6); that is,

$$\mathcal{E}_{\alpha,\beta}^\delta(P, p, N) = \mathbb{E}_{\boldsymbol{\theta},\delta} \left[ \|f_{\boldsymbol{\theta}}^*(x) - f_{\widehat{\boldsymbol{\theta}}_p}(x)\|_{L^2(X)}^2 \right] = \mathbb{E}_{\boldsymbol{\theta},\delta} \left[ \|\widehat{\boldsymbol{\theta}}_p - \boldsymbol{\theta}\|_2^2 \right] , \tag{60}$$

using the generalized Parseval's identity (also Mercer's Theorem) in the corresponding reproducing kernel Hilbert space.

Popular kernels (Rasmussen & Williams, 2006, Chapter 4) in applications include the polynomial kernel

$$K_{Polynomial}(x, \tilde{x}) = (\alpha \langle x, \tilde{x} \rangle + 1)^d, \tag{61}$$

where $d$ is the degree of the polynomial, and the Gaussian RBF (Radial Basis Function) kernel

$$K_{Gaussian}(x, \tilde{x}) = \exp\left( -\frac{\|x - \tilde{x}\|^2}{2\sigma^2} \right) . \tag{62}$$

where $\sigma$ is the standard deviation. For appropriate datasets, such as those normalized ones that live on the unit sphere $\mathbb{S}^{d-1}$, there is theoretical as well as numerical evidence to show that the feature matrix $\Psi$ has eigenvalues decay fast (for instance, algebraically). This means that for such problems, we also have that the low-frequency modes dominate the high-frequency modes in the target function. This means that the weighted optimization framework we analyzed in the previous sections should also apply here. We refer interested readers to Rasmussen & Williams (2006) and references therein for more technical details and summarize the main theoretical results here.

**Neural tangent kernel.** It turns out that a similar technique can be used to understand some aspects of learning with neural networks. It is particularly related to the frequency bias of neural networks that has been extensively studied (Ronen et al., 2019; Wang et al., 2020). It is also closely related to the regularization properties of neural networks (Martin & Mahoney, 2018). To make the connection, we consider the training of a simple two-layer neural network following the work of Yang & Salman (2019) and Ronen et al. (2019). We refer interested readers to Daniely et al. (2016) where kernel formulation of the initialization of deep neural nets was first introduced. In their setting, the neural network is a concentration of a computation skeleton, and the initialization of the neural network is done by sampling a Gaussian random variable.

We denote by $f$ a two-layer neural network that takes input vector $\mathbf{x} \in \mathbb{R}^d$ to output a scalar value. We assume that the hidden layer has $J$ neurons. We can then write the network, with activation function $\sigma$, as

$$f(\mathbf{x}; \Theta, \boldsymbol{\alpha}) = \frac{1}{\sqrt{M}} \sum_{m=1}^{M} \alpha_m \sigma(\boldsymbol{\theta}_m^{\mathfrak{T}} \mathbf{x}), \tag{63}$$

where $\Theta = [\boldsymbol{\theta}_1, \cdots, \boldsymbol{\theta}_M] \in \mathbb{R}^{d \times M}$ and $\boldsymbol{\alpha} = [\alpha_1, \cdots, \alpha_M]^{\mathfrak{T}} \in \mathbb{R}^J$ are respectively the weights of the hidden layer and the output layer of the network. We omit bias in the model only for simplicity.

In the analysis of Ronen et al. (2019); Yang & Salman (2019), it is assumed that the weight of the output layer $\boldsymbol{\alpha}$ is known and one is therefore only interested in fitting the data to get $\Theta$. This training process is done by minimizing the $L^2$ loss over the data set $\{\mathbf{x}_j, y_j\}_{j=1}^N$:

$$\Phi(\Theta) = \frac{1}{2} \sum_{j=1}^{N} \left( y_j - f(\mathbf{x}_j; \Theta, \boldsymbol{\alpha}) \right)^2.$$

When the activation function $\sigma$ is taken as the ReLU function $\sigma(x) = \max(x, 0)$, we can define the matrix, which depends on $\Theta$,

$$\Psi(\Theta) = \frac{1}{\sqrt{M}} \begin{pmatrix} \alpha_1 \chi_{11} \mathbf{x}_1 & \alpha_2 \chi_{12} \mathbf{x}_1 & \cdots & \alpha_M \chi_{1M} \mathbf{x}_1 \\ \alpha_1 \chi_{21} \mathbf{x}_2 & \alpha_2 \chi_{22} \mathbf{x}_2 & \cdots & \alpha_M \chi_{2M} \mathbf{x}_2 \\ \vdots & \vdots & \ddots & \vdots \\ \alpha_1 \chi_{N1} \mathbf{x}_N & \alpha_2 \chi_{N2} \mathbf{x}_N & \cdots & \alpha_M \chi_{NM} \mathbf{x}_N \end{pmatrix},$$

where $\chi_{jm} = 1$ if $\boldsymbol{\theta}_m^{\mathfrak{T}} \mathbf{x}_j \geq 0$ and $\chi_{jm} = 0$ if $\boldsymbol{\theta}_m^{\mathfrak{T}} \mathbf{x}_j < 0$. The least-squares training loss can then be written as $\frac{1}{2} \|\Psi(\Theta)\Theta - y\|_2^2$. A linearization of the least-squares problem around $\Theta_0$ can then be formulated as

$$\Delta\Theta = \arg\min_{\Delta\Theta} \|\Psi(\Theta_0)\Delta\Theta - \Delta y\|_2^2, \tag{64}$$

where $\Delta y := y - \Psi(\Theta_0)\Theta_0$ is the perturbed data.

Under the assumption that the input data are normalized such that $\|\mathbf{x}\| = 1$ and the weight $\alpha_k \sim \mathcal{U}(-1, 1)$ ($1 \leq k \leq M$), it was shown in Ronen et al. (2019) that the singular values of the matrix $\Psi(\Theta_0)$ decays algebraically. In fact, starting from initialization $\boldsymbol{\theta}_m \sim \mathcal{N}(0, \kappa^2 \mathbf{I})$, this result holds during the whole training process under some mild assumptions.

We summarize the main result on the Gaussian kernel and the linear kernel in the following theorem.

**Theorem B.1** (Theorem 2 and Theorem 3 of Minh et al. (2006)). *Let $X = \mathbb{S}^{n-1}$, $n \in \mathbb{N}$ and $n \geq 2$. Let $\mu$ be the uniform probability distribution on $\mathbb{S}^{n-1}$. Then eigenvalues and eigenfunctions for the Gaussian kernel (62) are respectively*

$$\lambda_k = e^{-2/\sigma^2} \sigma^{n-2} I_{k+n/2-1} \left( \frac{2}{\sigma^2} \right) \Gamma \left( \frac{n}{2} \right) \tag{65}$$

*for all $k \geq 0$, where $I$ denotes the modified Bessel function of the first kind. Each $\lambda_k$ occurs with multiplicity $N(n, k)$ with the corresponding eigenfunctions being spherical harmonics of order $k$ on $\mathbb{S}^{d-1}$. The $\lambda_k$'s are decreasing if $\sigma \geq \sqrt{2/n}$ for the Gaussian kernel (Rasmussen & Williams, 2006). Furthermore, $\lambda_k$ forms a descreasing sequence and*

$$\left( \frac{2e}{\sigma^2} \right)^k \frac{A_1}{(2k+n-2)^{k+\frac{n-1}{2}}} < \lambda_k \left( \frac{2e}{\sigma^2} \right)^k \frac{A_2}{(2k+n-2)^{k+\frac{n-1}{2}}}$$

*The nonzero eigenvalues are*

$$\lambda_k = 2^{d+n-2} \frac{d!}{(d-k)!} \frac{\Gamma(d + \frac{n-1}{2})\Gamma(\frac{n}{2})}{\sqrt{\pi}\Gamma(d+k+n-1)} \tag{66}$$

*for the polynommial kernel (61) with $0 \leq k \leq d$. Each $\lambda_k$ occurs with multiplicity $N(n, k)$, with the correspoding eigenfunctions being spherical harmonics of order $k$ on $\mathbb{S}^{n-1}$. Furthermore, $\lambda_k$ forms a descreasing sequence and*

$$\frac{B_1}{(k+d+n-2)^{2d+n-\frac{3}{2}}} < \lambda_k < \frac{B_2}{(k+d+n-2)^{d+n-\frac{3}{2}}}.$$

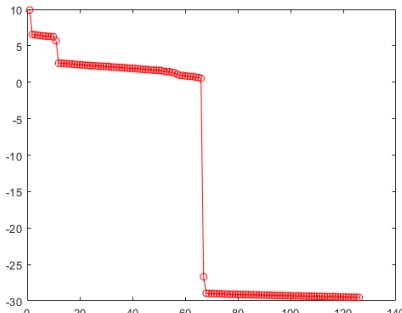 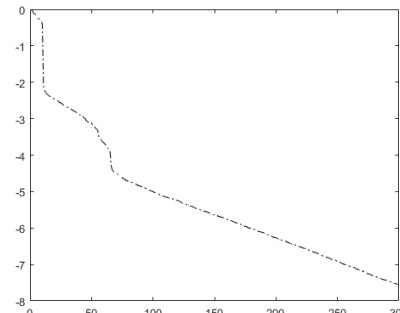

Figure 2: Decay of singular values of sampling matrices for the polynomial and Gaussian kernels respectively for the (normalized) MNIST data set. The $x$-axis represents the singular value index.

The results have been proved in different settings. When the samples are not on $\mathbb{S}^{n-1}$ but in the cube $[-1, 1]^n$ or the unit ball $\mathcal{B}^n$, or the underlying distrubion of the data is not uniform, there are similar results. We refer interested readers to Minh et al. (2006); Rasmussen & Williams (2006) and references therein for more details. In Figure 2, we plot the singular values of the sampling matrix for the polynomial kernel and the Gaussian RBF kernel for the MINST data set.

For the theorem to work for learning with neural networks as we discussed in Section 5, it has been shown that the neural tangent kernel also satisfies the required property in different settings. The main argument is that neural tangent kernel is equivalent to kernel regression with the Laplace kernel as proved in Chen & Xu (2020). We cite the following result for the two-layer neural network model.

**Theorem B.2** (Proposition 5 of Bietti & Mairal (2019)). *For any $x$, $\widetilde{x} \in \mathbb{S}^{n-1}$, the eigenvalues of the neural tangent kernel $\mathcal{K}$ are non-negative, satisfying $\mu_0, \mu_1 > 0$, $\mu_k = 0$ if $k = 2j + 1$ with $j \geq 1$, and otherwise $\mu_k \sim C(n)k^{-n}$ as $k \to \infty$, with $C(n)$ a constant depending only on $n$. The eigenfunction corresponding to $\mu_k$ is the spherical harmonic polynomials of degree $k$.*

The proof of this result can be found in Bietti & Mairal (2019). The similarity between neural tangent kernel and the Laplace kernel was first documented in Geifman et al. (2020), and a rigorous theory was developed in Chen & Xu (2020).

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
