# OpenReview forum: "A Generalized Weighted Optimization Method for Computational Learning and Inversion"
_ICLR.cc/2022/Conference — ICLR 2022 Poster_

### Official Review · Reviewer_WRNZ · 2021-11-03

**Correctness:** 4
**Technical Novelty And Significance:** 4
**Empirical Novelty And Significance:** 4
**Recommendation:** 6
**Confidence:** 4

**Main Review:**

strengths: solid analysis

my concerns:
1. I feel difficult to imagine how the introduction of weight to least squares loss helps to "deal with noise in the training data" or introduce a priori information. I hope that the author(s) could provide some motivating application scenarios.
2. It seems to me (for example, if one studies Theorem 3.1 carefully) that in this paper, the weights for both parameter vector and data points are the same (except that they are raised to different powers, -alpha and -beta, respectively). Prior information in these two spaces could be different. Is there some reason why we use the same sequence of weights?
3. For the sake of application, are there any guidelines on selecting the weight sequence?

**Summary Of The Paper:**

This paper starts from the random Fourier model, generalizes the recent results on weighting the coefficients to adopt weighted least squares loss. Under different settings of noise levels and over-/under- parameterized regimes, error analysis has been provided.

**Summary Of The Review:**

Better if the author(s) could provide more motivations and guidelines on how the weight could be selected, and why the same sequence of weights is used for both the sample space and the parameter space.

---

> ### Author Response · Authors · 2021-11-17
> **Response to Reviewer WRNZ**
>
> We thank the reviewer for the thorough review and constructive suggestions. Comments and suggestions made by the reviewer are addressed below.
>
> ### Q1. ... how the introduction of weight to least squares loss helps to ``deal with noise in the training data'' or introduce a priori information. I hope that the author(s) could provide some motivating application scenarios.
> ---
> Thanks for the comments and suggestions. We have now included more discussions on this issue as pointed out by the referee in the paper where we introduce the weighting matrices; see Pages 3-4.
>
> One scenario is to consider the random Fourier feature (RFF) model in the physical space.  In this case, if we have a weight matrix with decaying diagonal elements, we have a smoothing operator applied to the data mismatch,  which corresponds to an a priori assumption that the data contain high-frequency noise.  In this case, we essentially smooth out the high-frequency noise in the data and then learn with the smoothed version of the data. In our case, we kept the weight matrix invertible, so we did not remove noise but instead suppressed the noise (in the frequency domain). If we choose a weight matrix to be non-invertible, we would be effectively changing the data by applying the weight matrices before learning from it.  What we showed here is that even if the weight matrix is invertible, when noise is present, the weight matrix can still have a large impact on the learning error.
>
> Indeed, one can also choose weighting matrices with growing diagonal elements in the RFF model if the a priori assumption on the noise is of low frequency (which is the case for some applications) or assumption (3) holds for some $\gamma < 0$ in contrast to the current assumption that $\gamma$ is a positive constant.
>
> ### Q2. ... the weights for both parameter vector and data points are the same... Prior information in these two spaces could be different. Is there some reason why we use the same sequence of weights?
> ---
> The referee raised good points. The reason we use the same type of weights for both the parameter domain and the data domain is to make the calculation explicit to show the effective impact of the weighting scheme quantitatively.  This choice makes it easy to characterize the dependence of the generalization error on the weighting schemes.  However, we remark that the selection is by no means necessary in terms of applications,  while here it is for the convenience of mathematical derivations. We now put a short remark on this issue on Page 3 of the revision.
>
> On the other hand,  it is also evident that different weighting schemes may have different impacts on the generalization error.  In the specific setup of the RFF model, our selection mimics the effect that signals we recover are usually smooth, and thus have decaying Fourier modes.  There are two scenarios.
> 1. First, these two weight matrices matter differently in different regimes when the problem we work on is noise-free, and the optimization can be solved perfectly, and the weighting scheme is invertible. This is the result in Section 3.
> 2. When we are away from this ideally clean regime and move to the practical settings, both weight matrices matter, and their impacts can be seen in both the overparameterized and the underparameterized regimes (as part of Section 4 and Section 5). We remark that the roles of these two weight matrices are still different, even if they both matter in this scenario.
>
> ### Q3. ... are there any guidelines on selecting the weight sequence?
> ------
> One general principle to select the parameter-domain weighting matrix (the one with exponent $-\beta$ in (7) of our paper) is that if we want to highlight certain features in the object to be learned, we should choose weighting schemes that emphasize these features. When the features are indeed the correct features to be emphasized, we get better generalization also (otherwise, we emphasize the wrong features, and therefore get worse generalization). The exact format of the weight matrix depends on the form of the features. The weight matrix we selected in the RFF model corresponds to a smoothing convolution operator (whose kernel has Fourier coefficients given in the diagonal of the weight matrix) in the physical space.
>
> One general principle to select the data-domain weighting matrix (the one with exponent $-\alpha$ in (7) of our paper) is to suppress the noise effect in the data.  For example, it is preferable to select weight matrices with smoothing features if the noise in the data is high-frequency and to select anti-smoothing weight matrices if the noise in the data is of low frequency. This is a straightforward way to reduce the impact of the noise propagated to the learned coefficient ${\widehat{\theta}}^\delta_p$.
>
> We have added such remarks in the revision; please see the last paragraph of Section 3 and Section 4, respectively, and the last two paragraphs of Section 5.

---

### Official Review · Reviewer_muho · 2021-11-04

**Correctness:** 3
**Technical Novelty And Significance:** 3
**Empirical Novelty And Significance:** Not applicable
**Recommendation:** 6
**Confidence:** 2

**Main Review:**

The technical contribution is solid and generally of interest to the machine learning community. However, there is only a loose match with the topics of interest according to the CfP of ICLR22, so it may be very different (topically) compared to the bulk of submissions. Not sure how the PC sees that.

Some parts could be underpinned with more information, e.g., I am wondering what implication follows from the statement that "in the case of α = 0, when β ̸= 0, our result in (14) is slightly different from its equivalence in Xie et al. (2020)"?

**Summary Of The Paper:**

The authors study generalization errors in Fourier regression scenarios with over and under parameterized models. They also present generalizations to vanilla feature regression.

**Summary Of The Review:**

Strong technical part, perhaps somewhat off topic for ICLR22 but nevertheless still a very good paper.

---

> ### Author Response · Authors · 2021-11-17
> **Response to Reviewer muho**
>
> We thank the reviewer for the thorough review and constructive suggestions. Comments and suggestions made by the reviewer are addressed below.
>
> ### Q1. Match with the topics of interest of ICLR
> --------
> One of the areas mentioned on https://iclr.cc/Conferences/2022/CallForPapers is feature learning, which is the focus of our paper as the main objective of our work is to analyze the impact of weighting optimization schemes on the generalization error of linear feature learning problems.
>
> Here, weighting is relative to the usual $L^2$-based least-squares method. The work is motivated by the fact that while weighting is used ubiquitously in learning explicitly (for instance least-squares weighted by the covariance matrix of the data) or implicitly (for instance learning with objective functions using metrics different from the $L^2$ metric), there is little detailed analysis on the impact of weighting on the result of learning. In our work, we first analyze a toy random Fourier feature model in detail which allows us to compute the generalization error exactly, to demonstrate the impact of weighting in different clean settings. We then generalize the results to more realistic settings (in Section 5 where we consider general features in the multi-dimensional setting). While we were not able to obtain exact formulas for generalization error in practically relevant cases, error bounds in these situations are obtained which demonstrate the same type of impact we observe in the toy model. Our framework offers an additional degree of freedom in the optimization process for users to select weight schemes that are adapted to the problem they are studying. Our analytical results provide guidelines on selecting such weighting schemes to emphasize the features of the unknowns to be learned and understand the impact of the selection on the generalization error of the learning results.
>
> ### Q2. Some parts could be underpinned with more information, e.g., I am wondering what implication follows from the statement that "in the case of $\alpha = 0$, when $\beta\neq 0$, our result in (14) is slightly different from its equivalence in Xie et al. (2020)"?
> --------
> What we derived, in the setup of $\alpha=0$ and $\beta\neq 0$, is a minor correction of the corresponding result in Xie et al. (2020). We have now clarified this statement in the new version of the paper; please see Page 5. Thanks again for the comments.

---

### Official Review · Reviewer_vfX6 · 2021-11-04

**Correctness:** 4
**Technical Novelty And Significance:** 3
**Empirical Novelty And Significance:** 3
**Recommendation:** 5
**Confidence:** 4

**Main Review:**

Pros:

This paper is well-written and easy to follow. The findings of this paper mainly focus on the relationship between generalization and noise under various weighted matrices. For least squares model, the weighted matrices play different roles in under-/over-parameterized regimes. For random features model, the weighted setting is able to minimize the impact of noise.

Cons:

One significant issue I concerned is the one-dimensional data setting, i.e., x_j = 2\pi j/N,
 which could significantly decrease the value of this work and restrict its findings/observations for a general real-world case. The considered data distribution is quite simple and specific.

Regarding to Eq. (3), this is actually an assumption. It requires that the second-order moment (matrix) of the parameter theta is diagonal and decay fast. Normally, it’s fair to assume the data admitting this, e.g., E(xx’)=\Sigma for isotopic/covariate data. I do not find any justification for the diagonality assumption on the parameter. Besides, it also requires this matrix decays fast with \gamma > 0, I understand this setting as the generalization error requires this decay, as given in Eq. (28). Nevertheless, the problem setting is quite specific on a toy model.

Another issue is that, in Theorem 5.1 for random features model, the generalization error (LHS) is a random variable as it does not take the expectation on the random features (implicitly included in Eq. (22)) ? The RHS is deterministic? This error bound appears a little strange if my understanding was right. If both LHS and RHS are random, a probability error bound is needed.

Besides, it would be best to compare Theorem 5.1 with the following refs that focuses on RFF under noisy data in the interpolation regime.

Li, Zhu, Zhi-Hua Zhou, and Arthur Gretton. "Towards an Understanding of Benign Overfitting in Neural Networks." arXiv preprint arXiv:2106.03212 (2021).


**Summary Of The Paper:**

This paper studies the weighted least squares, random features model under noise one-dimensional data setting in under-/over-parameterized regime. The derived error bounds demonstrate the impact of noise on the generalization error. Besides, the extension to kernel regression shows that, the selected weighted matrix is helpful to generalization when the RKHS is small (i.e., singular values of \Psi decay fast).

**Summary Of The Review:**

In sum, the problem setting is quite simple, specific, and appears far away from practice settings. The derived results bring in new message and findings but could be not enough to overlook the drawback of problem setting.

---

> ### Author Response · Authors · 2021-11-17
> **Response to Reviewer vfX6**
>
> We thank the reviewer for the thorough review and constructive suggestions. Comments and suggestions made by the reviewer are addressed below.
>
> ### Q1. ... the one-dimensional data setting, i.e., $x_j = 2\pi j/N$, which could significantly decrease the value of this work and restrict its findings/observations for a general real-world case. The considered data distribution is quite simple and specific.
> The reviewer raised a similar issue with Q1 from Referee "Koia". First, there are two main reasons why we used the 1D input for the random Fourier feature (RFF) model.
> 1. We started with the toy RFF model in the earlier sections (majorly Section 3 and Section 4) is that it allows us to explicitly compute the generalization error exactly so that we could see the impact of every factor in the weighting scheme precisely.
> 2. We use the 1D assumption at the beginning of the paper is to compare with Xie et al. (2020) and to illustrate the additional impact of the data domain weighting in our proposed generalized weighting framework,  which was not considered in Xie et al. (2020).
>
> The main results that we are interested in are the ones in the asymptotic regimes in Section 5 and also in Appendix B. Those are for general feature models (not restricted to the RFF assumption) with multi-dimensional input.
>
> Moreover, the input can come from uniform random sampling points rather than a uniform grid,  following the same lines of calculation in Belkin et al.~(2020) as we referenced and remarked in the paper.  The deterministic results we have will have to be changed to hold with high probability. We added a remark on this issue in the revision on Page 1.
>
> Again, we remark that some assumptions we made here are to achieve **precise analytical generalization errors**. The effects of weighting schemes will remain when such assumptions are relaxed, but mathematically, we can only see the effects from error bounds (Section 5) instead of equalities (Sections 3 and 4).
>
> ------
>
> ### Q2. Regarding the assumption in Equation (3).
> Thanks for the comments. We want to emphasize that neither the decay nor the diagonal structure is necessary for the main asymptotic results in Section 5 to hold. In fact, the decay with $\gamma > 0$ and the diagonal structure are only taken to simplify the calculations so that we could have an exact result on the generalization error for the random Fourier feature model. (We have revised the paper to make this point more clear; please see Page 9.) The only required assumption is that the parameter and the noise in the data are independent.
>
> The main assumption in (3) says that statistically, the signal to be recovered has algebraically decaying Fourier coefficients. This is equivalent to assuming that the target function we are learning is relatively smooth, which is undoubtedly true for many functions to be learned from kernel regression and functions as physical forward models in practical applications such as inverse coefficient problems to PDEs (optical tomography, electrical impedance tomography, seismic imaging, etc.), since the major ill-posedness of such inverse problems is due to the smoothness of the forward model.
>
> In such physical applications, the smoothness of the forward model, i.e., the $\gamma$ in our (3), is a quantitative characterization of how difficult the inverse problem is; see, for instance, the reference Engl et al. (1996).  We have included more discussions on this aspect in the revised version of the paper under Eq. (3).
>
> ----
>
> ### Q3. Regarding the LHS and RHS of the main results in Theorem 5.1.
>
> Thanks for pointing out the issue. The LHS has included the expectation over $\boldsymbol\theta$ (in fact both $\boldsymbol\theta$ and $\boldsymbol \delta$). We have clarified the notation in the revised version; please see the paragraph above Theorem 5.1.
>
> ----
>
> ### Q4. ... it would be best to compare Theorem 5.1 with the following Li et al. (2021) that focuses on RFF under noisy data in the interpolation regime.
>
> Thanks for pointing out the reference. The study in the reference is very interesting and is certainly highly related. We have cited this in the revised version of our paper.
>
> The objective of the reference and that of our paper are different, however. Our goal is to analyze the impact of two different weighting matrices (data weighting and model parameter weighting, respectively) on the generalization error while the reference analyzed generalization error in the regular optimization setting (without weightings).  Our study tries to understand phenomenon such as "optimization based on weak mismatch metrics yield results with better generalization". We believe we could perform a similar analysis in our setting analog to some of the results within the reference in future work.  We do not have this at the moment.
>
> ----
>
> We hope our responses clarify some issues that the reviewer has raised. We will be more than happy to continue the discussions.

---

> > ### Comment · Reviewer_vfX6 · 2021-11-29
> > **increase the score but concern the motivation**
> >
> > Thanks for your update.
> >
> > Regarding Q1 and Q2, the authors claim that these two issues can be fixed in RFF in Section 5, and thus I increase my score to 5. The reasons why I tend to reject this paper are:
> >
> > 1. If RFF setting holds for multiple dimensional data and does not need the assumption in Eq. (3), there appears no need to focus on least squares. It sounds natural and nice to begin with the RFF setting.
> >
> > 2. The motivation on anlyzing the impact of two different weighting matrices (data weighting and model parameter weighting, respectively) on the generalization error is relatively weak as the weights are directly given or assumed.

---

> > > ### Author Response · Authors · 2021-11-30
> > > **Reply to further comments from Reviewer vfX6**
> > >
> > > We really appreciate the reviewer for raising the score and the further comments. Please see below for our replies that address some of the concerns.
> > >
> > > 1. One of the main reasons we consider the least-squares formulation is its connection to objective functions in optimization used in the literature based on other mathematical metrics (such as those based on the quadratic Wasserstein metric from Optimal Transport and the ones related to the $L^2$-based Sobolev norms). Many of such objective functions can be viewed as weighted least-squares with special weighting matrices after discretization. Our analysis through the weighted least-squares is related to studying the impact of different objective functions (particularly, the class of $L^2$-based Sobolev norms) on the generalization error of machine learning problems.
> > >
> > > 2. We take the weight matrices in our analysis in specific forms for the sake of concreteness. The assumption in Eq. (3) is to provide a priori knowledge about this model, based on which we select proper weighting schemes. Eq. (3) corresponds to a smooth forward operator in the physical domain. This is the type of problem where focusing on low-frequency modes of the target function (**model-parameter-space weighting**) is beneficial (of course, if the prior knowledge about the model changes, the optimal weighting scheme needs to change accordingly). An equally important consideration is, with noise in data, one should try to suppress the impacts of noise in the inversion/learning process to avoid overfitting. Taking a "smoothing" **data-space weighting** operator is a natural choice to suppress the high-frequency oscillatory noise in the data. The weight matrices in data space and the model parameter space **do not generally commute** unless the feature matrix is diagonal. This is the main motivation to separate the two spaces and study the impact of the two weight matrices separately.
> > >
> > > It is true that with the length limit of the paper, we could not have a more thorough analysis of different types of weighting strategies for various applications. We do hope that our work can motivate more analytical and computational work in this direction (which we believe is important for optimization-based learning and inversion algorithms).

---

### Official Review · Reviewer_Koia · 2021-11-06

**Correctness:** 4
**Technical Novelty And Significance:** 3
**Empirical Novelty And Significance:** Not applicable
**Recommendation:** 6
**Confidence:** 2

**Main Review:**

The presented weighted least-squares formulation is of interest. Theorem 3.1 gives the generalization error bounds for random Fourier feature model in both under- and over-parameterized regrimes for noise-free setting. Theses results are extended to the noisy setting in Theorem 4.2 and general feature regression in Theorem 5.1. I have some concerns as follows.

1. Is it possible to consider higher dimensional input space (instead of 1) and random sampling points?

2. The weight matrix $\Lambda_{[N]}^{-\alpha}$ only matters in the underparameterized regime. Does this mean we should not use it in the overparameterization regime, which should be of more interest?

3. In the overparameterization regime, is it possible to observe the double descent phenomenon?


**Summary Of The Paper:**

The paper considers a novel generalized weighted least-squares optimization method for the random Fourier feature model and conducts its generalization error analysis in both under- and over-parameterized regimes. The impact of the proposed weight matrices are also discussed.

**Summary Of The Review:**

Overall, the paper is well written and technically sound. The results should be of interest to the community.

---

> ### Author Response · Authors · 2021-11-17
> **Response to Reviewer Koia**
>
> We thank the reviewer for the thorough review and constructive suggestions. Comments and suggestions made by the reviewer are addressed below.
>
> -----------
> ### Q1. Is it possible to consider higher dimensional input space (instead of 1) and random sampling points?
> Yes, it is possible.  Firstly, there are two reasons why we used the 1D input for the random Fourier feature (RFF) model.
> 1. The first main reason that we started with the toy RFF model in the earlier sections (majorly Section 3 and Section 4) is that it allows us to explicitly compute the generalization error exactly so that we could see the impact of every factor the weighting scheme precisely.
> 2. The second main reason that we use the 1D assumption at the beginning of the paper is to compare with Xie et al. (2020)  and to illustrate the additional impact of the data domain weighting in our proposed generalized weighting framework, which was not considered in Xie et al. (2020).
>
> The main results that we are interested in are the ones in the asymptotic regimes in Section 5,  and also in Appendix B.
> Those results are for general feature models (and are not restricted to the random Fourier feature assumption) with multi-dimensional input.   Moreover, the input can come from uniform random sampling points rather than a uniform grid,  following the same lines of calculation in Belkin et al. (2020) as we referenced.  The deterministic results we have here in the paper will need to be modified to hold with high probability (w.h.p.).
> -----------
> ### Q2. The weight matrix only matters in the underparameterized regime. Does this mean we should not use it in the overparameterization regime, which should be of more interest?
>
> We thank the referee for pointing out the issue. There are two scenarios.
> 1. First, these two weight matrices matter differently in different regimes when the problem we work on is noise-free, and the optimization can be solved perfectly, and the weighting scheme is invertible. This is the result in Section 3.
> 2. When we are away from this ideally clean regime and move to the practical settings, both weight matrices matter, and their impacts can be seen in both the overparameterized and the underparameterized regimes (as part of Section 4 and Section 5).
>
> We have now included more discussions to emphasize this point in the revised version of the paper; please see Page 5.
>
> -----------
> ### Q3.  In the overparameterization regime, is it possible to observe the double descent phenomenon?
> We did not particularly study this phenomenon in our paper. However, double descent should still hold in general. We have now included a demonstration of the double-descent phenomenon in Figure 1b on Page 6 to illustrate the impact of the weight matrix in the under-parameterized regime.

---

### Official Review · Reviewer_XgS8 · 2021-11-14

**Correctness:** 4
**Technical Novelty And Significance:** 3
**Empirical Novelty And Significance:** Not applicable
**Recommendation:** 6
**Confidence:** 4

**Main Review:**


 The paper is a solid work that provides some substantial extension of the work (Xie et al. (2020)) for a more general weighted scheme and the noise case study.   The main results are given in Theorem 3.1

Comments:

1. It would be useful to provide some practical/theoretical motivations on why the model
assumption given by (3) is reasonable.
2.	On page 5, could you provide a clear and explicit explanation based on the results of Theorem 3.1 for the stated observation there:
“the weight matrix $\Lambda_{[p]}^{-\beta}$ only matters in the over parametrized regime while …”
3.	Would it be possible to provide a figure to describe the double-descent curve or the effect of different weight matrices which is similar to Figure 1 in Xie et al. (2020).
4.	It would be more helpful if there are some explanations about the technical difference/novelty from the previous work.


**Summary Of The Paper:**

 The paper follows and extends the work by Belkin, Hsu, and Xu (2020) and  Liang and Rakhlin (2018) which studies the bias-variance trade-off of the regression/interpolation problem in the under/over-parametrization regions.  In particular, this paper follows the random Fourier model setting in Xie et al. (2020) and analyzed a generalized weighted least-square optimization method that allows the weighting in both the parametrization and data space. The authors derived the generalization error of such weighted least-square framework for the over parametrized and under parameterized regimes and compare them in these two cases.  The general conclusion is that emphasizing low-frequency features provide better generalization ability.  The paper also studies both noise-free and noise cases.

**Summary Of The Review:**


 This solid theoretical work is a substantial extension of Xie et al. (2020) by considering weighting in both parameter space and data space. The main contribution compared to the related work is well illustrated.  The paper is also well written. I have no time to read the proofs step by step, but it would be helpful if the authors can highlight the novelty of the proof techniques.   I support its acceptance of this solid theoretical work after the authors can address some of my comments mentioned above.

---

> ### Author Response · Authors · 2021-11-17
> **Response to Reviewer XgS8**
>
> We thank the reviewer for the thorough review and constructive suggestions. Comments and suggestions made by the reviewer are addressed below.
>
> -----------
> ### Q1. It would be useful to provide some practical/theoretical motivations on why the model assumption given by (3) is reasonable.
>
> We thank the referee for the suggestion.  First, assumption (3) is not necessary for the theory of the paper to hold. Changing (3) will only change the generalization error quantitatively but will not change the paper's main conclusion.  Second, the mean zero assumption can be easily removed. The primary assumption in (3) says that statistically, the signal to be recovered has algebraically decaying Fourier coefficients. This is simply saying that the target function we are learning is relatively smooth, which is undoubtedly true for many functions in kernel regression (as detailed in Appendix B) and as physical models in practical applications, such as inverse radon transform,  inverse scattering, and electrical impedance tomography problems.  If the Fourier coefficients do not decay,  then the signal in the physical space behaves like random noise.  As a result, there is no coherent structure in the signal to be learned.  Moreover, the covariance matrix does not need to be diagonal either, which we assume simply for convenience in terms of explicit calculation in Sections 3 and 4.  We now have added more discussions in the revised version under (3).
>
> -----------
> ### Q2. On page 5, could you provide a clear and explicit explanation based on the results of Theorem 3.1 for the stated observation there: "the weight matrix only matters in the over parametrized regime while ..."
>
> There are two scenarios. First, the two weight matrices matter differently when the problem we work on is noise-free, and the optimization problem can be solved perfectly, and the weighting schemes are invertible.  Second, when we are away from the formerly ideal and clean regime to move to the practical settings without such strong assumptions, both weight matrices matter and their impacts can be seen in both the overparameterized and the underparameterized regimes. We have now added more discussions to emphasize these points in the revised version of the paper on Page 5.
>
> -----------
> ### Q3. Would it be possible to provide a figure to describe the double-descent curve or the effect of different weight matrices which is similar to Figure 1 in Xie et al. (2020).
>
> We did not systematically study this double descent phenomenon for the more complicated settings that we have analyzed, but we have included now a plot of the double descent curves for the case of the random Fourier feature model with noise-free data obtained in Theorem 1 to demonstrate the impact of the weight matrix in the underparameterized regime; please see Figure 1b on Page 6.
>
> -----------
> ### Q4.  It would be more helpful if there are some explanations about the technical difference/novelty from the previous work.
>
> Thanks a lot for the suggestion.  We have now added more discussion on this in the summary of our contributions from this paper; see Page 3.   Our contributions are mainly on two aspects.
>
> 1. From the mathematical understanding perspective: our analysis provides explanations on the impact of two different weight schemes, one on the model parameter domain and another on the data domain (whether it is done explicitly by using weight matrices or implicitly by using different metrics to measure the data mismatches),  in terms of  the generalization error of the learning result;
> 2. From the application perspective: our analysis provides some guidelines on selecting weighting schemes through either the parameter domain weighting or the data domain weighting, or both, to emphasize the features of the unknowns to be learned based on a priori knowledge; see some applications in some references we cited, for instance, Bal et al. (2009), Engquist et al. (2020), and Yang et al. (2021).

---

### Decision · Program_Chairs · 2022-01-20

**Decision:**

Accept (Poster)

**Comment:**

This paper considers a generalized weighted least-squares optimization method for the random Fourier feature model. Generalization error analysis is carried out under both the over-parametrized and under-parametrized schemes, and under both noise-free and noisy scenarios.

Reviewers generally agree that this is a solid theoretical work that considerably extends previous work (Belkin et al. (2020), Xie et al. (2020)) in the literature and that the paper is ready for publication.

While some reviewers express reservation about the relevance of the work to ICLR, I believe this is a nice work that would be of interest to the machine learning community at large.